# Transmission of West Nile and five other temperate mosquito-borne viruses peaks at temperatures between 23°C and 26°C

Marta S Shocket[1,2]*, Anna B Verwillow[1], Mailo G Numazu[1], Hani Slamani[3], Jeremy M Cohen[4,5], Fadoua El Moustaid[6], Jason Rohr[4,7], Leah R Johnson[3,6], Erin A Mordecai[1]

[1]Department of Biology, Stanford University, Stanford, United States; [2]Department of Ecology and Evolutionary Biology, University of California Los Angeles, Los Angeles, United States; [3]Department of Statistics, Virginia Polytechnic Institute and State University (Virginia Tech), Blacksburg, United States; [4]Department of Integrative Biology, University of South Florida, Tampa, United States; [5]Department of Forest and Wildlife Ecology, University of Wisconsin, Madison, United States; [6]Department of Biological Sciences, Virginia Polytechnic Institute and State University (Virginia Tech), Blacksburg, United States; [7]Department of Biological Sciences, Eck Institute of Global Health, Environmental Change Initiative, University of Notre Dame, South Bend, United States

**Abstract** The temperature-dependence of many important mosquito-borne diseases has never been quantified. These relationships are critical for understanding current distributions and predicting future shifts from climate change. We used trait-based models to characterize temperature-dependent transmission of 10 vector–pathogen pairs of mosquitoes (*Culex pipiens*, *Cx. quinquefasciatus*, *Cx. tarsalis*, and others) and viruses (West Nile, Eastern and Western Equine Encephalitis, St. Louis Encephalitis, Sindbis, and Rift Valley Fever viruses), most with substantial transmission in temperate regions. Transmission is optimized at intermediate temperatures (23–26°C) and often has wider thermal breadths (due to cooler lower thermal limits) compared to pathogens with predominately tropical distributions (in previous studies). The incidence of human West Nile virus cases across US counties responded unimodally to average summer temperature and peaked at 24°C, matching model-predicted optima (24–25°C). Climate warming will likely shift transmission of these diseases, increasing it in cooler locations while decreasing it in warmer locations.

*For correspondence:
marta.shocket@gmail.com

Competing interests: The authors declare that no competing interests exist.

## Introduction

Temperature is a key driver of transmission of mosquito-borne diseases. Both mosquitoes and the pathogens they transmit are ectotherms whose physiology and life histories depend strongly on environmental temperature (*Johnson et al., 2015*; *Mordecai et al., 2019*; *Mordecai et al., 2017*; *Mordecai et al., 2013*; *Paull et al., 2017*; *Rogers and Randolph, 2006*; *Shocket et al., 2018*; *Tesla et al., 2018*). These temperature-dependent traits drive the biological processes required for transmission. For example, temperature-dependent fecundity, development, and mortality of mosquitoes determine whether vectors are present in sufficient numbers for transmission. Temperature also affects the mosquito biting rate on hosts and probability of becoming infectious.

Mechanistic models based on these traits and guided by principles of thermal biology predict that the thermal response of transmission is unimodal: transmission peaks at intermediate

temperatures and declines at extreme cold and hot temperatures (*Johnson et al., 2015*; *Liu-Helmersson et al., 2014*; *Martens et al., 1997*; *Mordecai et al., 2019*; *Mordecai et al., 2017*; *Mordecai et al., 2013*; *Parham and Michael, 2010*; *Paull et al., 2017*; *Shocket et al., 2018*; *Tesla et al., 2018*; *Wesolowski et al., 2015*). This unimodal response is predicted consistently across mosquito-borne diseases (*Johnson et al., 2015*; *Mordecai et al., 2019*; *Mordecai et al., 2017*; *Mordecai et al., 2013*; *Paull et al., 2017*; *Shocket et al., 2018*; *Tesla et al., 2018*) and supported by independent empirical evidence for positive relationships between temperature and human cases in many settings (*Stewart-Ibarra and Lowe, 2013*; *Paull et al., 2017*; *Peña-García et al., 2017*; *Siraj et al., 2015*; *Werner et al., 2012*), but negative relationships at high temperatures in other studies (*Gatton et al., 2005*; *Mordecai et al., 2013*; *Peña-García et al., 2017*; *Perkins et al., 2015*; *Shah et al., 2019*). Accordingly, we expect increasing temperatures due to climate change to shift disease distributions geographically and seasonally, as warming increases transmission in cooler settings but decreases it in settings near or above the optimal temperature for transmission (*Lafferty, 2009*; *Lafferty and Mordecai, 2016*; *Rohr et al., 2011*; *Ryan et al., 2015*). Thus, mechanistic models have provided a powerful and general rule describing how temperature affects the transmission of mosquito-borne disease. However, thermal responses vary among mosquito and pathogen species and drive important differences in how predicted transmission responds to temperature, including the specific temperatures of the optimum and thermal limits for each vector–pathogen pair (*Johnson et al., 2015*; *Mordecai et al., 2019*; *Mordecai et al., 2017*; *Mordecai et al., 2013*; *Paull et al., 2017*; *Shocket et al., 2018*; *Tesla et al., 2018*). We currently lack a framework to describe or predict this variation among vectors and pathogens.

Filling this gap requires comparing mechanistic, temperature-dependent transmission models for many vector–pathogen pairs. However, models that incorporate all relevant traits are not yet available for many important pairs for several reasons. First, the number of relevant vector–pathogen pairs is large because many mosquitoes transmit multiple pathogens and many pathogens are transmitted by multiple vectors. Second, empirical data are costly to produce, and existing data are often insufficient because experiments or data reporting were not designed for this purpose. Here, we address these challenges by systematically compiling data and building models for understudied mosquito-borne disease systems, including important pathogens with substantial transmission in temperate areas like West Nile virus (WNV) and Eastern Equine Encephalitis virus (EEEV). Accurately characterizing the thermal limits and optima for these systems is critical for understanding where and when temperature currently promotes or suppresses transmission and where and when climate change will increase, decrease, or have minimal effects on transmission.

In this study, we model the effects of temperature on an overlapping suite of widespread, important mosquito vectors and viruses that currently lack complete temperature-dependent models. These viruses include: West Nile virus (WNV), St. Louis Encephalitis virus (SLEV), Eastern and Western Equine Encephalitis viruses (EEEV and WEEV), Sindbis virus (SINV), and Rift Valley fever virus (RVFV) (*Adouchief et al., 2016*; *Go et al., 2014*; *Kilpatrick, 2011*; *Linthicum et al., 2016*; *Weaver and Barrett, 2004*; summarized in *Table 1*). All but RVFV sustain substantial transmission in temperate regions (*Adouchief et al., 2016*; *Go et al., 2014*; *Kilpatrick, 2011*; *Linthicum et al., 2016*; *Weaver and Barrett, 2004*). We selected this group because many of the viruses share common vector species and several vector species transmit multiple viruses (*Table 1*, *Figure 1*). All the viruses cause febrile illness and severe disease symptoms, including long-term arthralgia and neuroinvasive syndromes with a substantial risk of mortality in severe cases (*Adouchief et al., 2016*; *Go et al., 2014*; *Kilpatrick, 2011*; *Linthicum et al., 2016*; *Weaver and Barrett, 2004*). Since invading North America in 1999, WNV is now distributed worldwide (*Kilpatrick, 2011*; *Rohr et al., 2011*) and is the most common mosquito-borne disease in the US, Canada, and Europe. SLEV, EEEV, and WEEV occur in the Western hemisphere (*Table 1*), with cases in North, Central, and South America (*Centers for Disease Control and Prevention, 2018a*; *Centers for Disease Control and Prevention, 2018b*; *Go et al., 2014*). For EEEV, North American strains are genetically distinct and more virulent than the Central and South American strains (*Go et al., 2014*). An unusually large outbreak of EEEV in the United States in 2019 has yielded incidence four times higher than average (31 cases, resulting in nine fatalities) and brought renewed attention to this disease (*Bates, 2019*). SINV occurs across Europe, Africa, Asia, and Australia, with substantial transmission in northern Europe and southern Africa (*Adouchief et al., 2016*; *Go et al., 2014*). RVFV originated in eastern Africa and now also occurs across Africa and the Middle East (*Linthicum et al., 2016*). These pathogens

**Table 1.** Properties of six viruses transmitted by an overlapping network of mosquito vectors.

| Virus (*genus*) | Primary vector spp. | Geographic range | Presentation and mortality | Epidemiology and Ecology |
|---|---|---|---|---|
| West Nile virus (WNV, *Flavivirus*) | *Cx. modestus, Cx. pipiens, Cx. quinquefasciatus, Cx. tarsalis* | Globally distributed | Febrile illness and encephalitis. 10% mortality in neuro-invasive cases. Long-term physical and cognitive disabilities. | The most common mosquito-borne disease in North America. Since invading in 1999, 7 million estimated infections, 22,999 neuroinvasive cases, and 2163 deaths in US; 5614 reported cases in Canada. Typically 100–300 cases annually in Europe, but over 1500 in 2018. Poor surveillance in Africa, but seroprevalence ~ 80% in some areas. Birds are main reservoir/amplification hosts. |
| St. Louis Encephalitis virus (SLEV, *Flavivirus*) | *Cx. quinquefasciatus, Cx. tarsalis* | Western hemisphere; western, midwestern, and southern US | Encephalitis. 5–15% mortality in diagnosed cases. | 92 cases and six deaths recorded in US from 2009 to 2018. Birds are main reservoir/amplification hosts. |
| Eastern Equine Encephalitis virus (EEEV, *Alphavirus*) | *Ae. triseriatus, Cs. melanura* | Western hemisphere; eastern and midwestern US | Febrile illness and encephalitis. 33% mortality in diagnosed cases. Long-term cognitive disabilities. | 73 cases and 30 deaths recorded in US from 2009 to 2018. Birds are main reservoir/amplification hosts. |
| Western Equine Encephalitis virus (WEEV, *Alphavirus*) | *Cx. tarsalis* | Western hemisphere; western and midwestern US | Febrile illness and encephalitis. Low mortality, except in infants. | 640 cases recorded in US from 1964 to 2010. Birds are main reservoir/amplification hosts. WEEV is derived from a recombinant event between the ancestors of EEEV and SINV. |
| Sindbis virus (SINV, *Alphavirus*), also called Pogosta, Ockelbo, and Karelian Fever | *Cx. torrentium, Cx. pipiens, Cx. univittatus* | Europe, Africa, Asia and Australia, primarily northern Europe and southern Africa | Febrile illness, rash, and joint pain. No mortality, but long-term disability. | Poor surveillance except in Finland, where annual incidence is 2–26 per 100,000 people and seroprevalence can reach ~40%. Birds are main reservoir/amplification hosts. Long-distance migratory birds may spread the virus between temperate zones in Northern and Southern hemispheres. |
| Rift Valley Fever virus (RVFV, *Phlebovirus*) | *Ae. mcintoshi, Ae. ochraceus, Ae. vexans, Cx. pipiens, Cx. poicilipes, Cx. theileri* and many more | Africa and the Middle East | Febrile illness and encephalitis. < 1% mortality in total cases. 50% mortality in hemorrhagic cases, permanent blindness in 50% of ocular cases (<2% of cases). | Livestock are main reservoir/amplification hosts, and suffer mortality and abortion after being infected by mosquitoes. Most transmission to humans occurs via direct contact with infected livestock. Vertical transmission in vectors (via dormant eggs) can initiate epidemics. In eastern and southern Africa, there are large epidemics every 5–15 years driven by rainfall and blooms of *Ae. spp.* from low-lying flooded areas known as *dambos*. |

Sources: WNV (**Centers for Disease Control and Prevention, 2018c**; **European Centre for Disease Prevention and Control, 2018**; **Golding et al., 2012**; **Government of Canada, 2018**; **Kilpatrick, 2011**; **Petersen et al., 2013**; **Ronca et al., 2019**; **Weaver and Barrett, 2004**); SLEV (**Centers for Disease Control and Prevention, 2018a**; **Weaver and Barrett, 2004**); EEEV (**Centers for Disease Control and Prevention, 2018b**; **Weaver and Barrett, 2004**); WEEV (**Ronca et al., 2016**; **Weaver and Barrett, 2004**); SINV (**Adouchief et al., 2016**); RVFV (**Braack et al., 2018**; **Linthicum et al., 2016**; **Sang et al., 2017**; **World Health Organization, 2018**).

primarily circulate and amplify in wild bird reservoir hosts (except RVFV, which primarily circulates in livestock). For all six viruses, humans are dead-end or unimportant reservoir hosts (**Go et al., 2014**; **Sang et al., 2017**), in contrast to pathogens like malaria, dengue virus, yellow fever virus, and Ross River virus, which sustain infection cycles between humans and mosquitoes (**Go et al., 2014**; **Gonçalves et al., 2017**; **Harley et al., 2001**). Most transmission of RVFV to humans occurs through direct contact with infected livestock (that are infected by mosquitoes), and to a lesser extent via the mosquito-borne transmission from infected vectors (**Sang et al., 2017**).

We primarily focus on *Culex pipiens*, *Cx. quinquefasciatus*, and *Cx. tarsalis*, well-studied species that are important vectors for many of the viruses and for which appropriate temperature-dependent data exist for nearly all traits relevant to transmission. Although the closely-related *Cx. pipiens* and

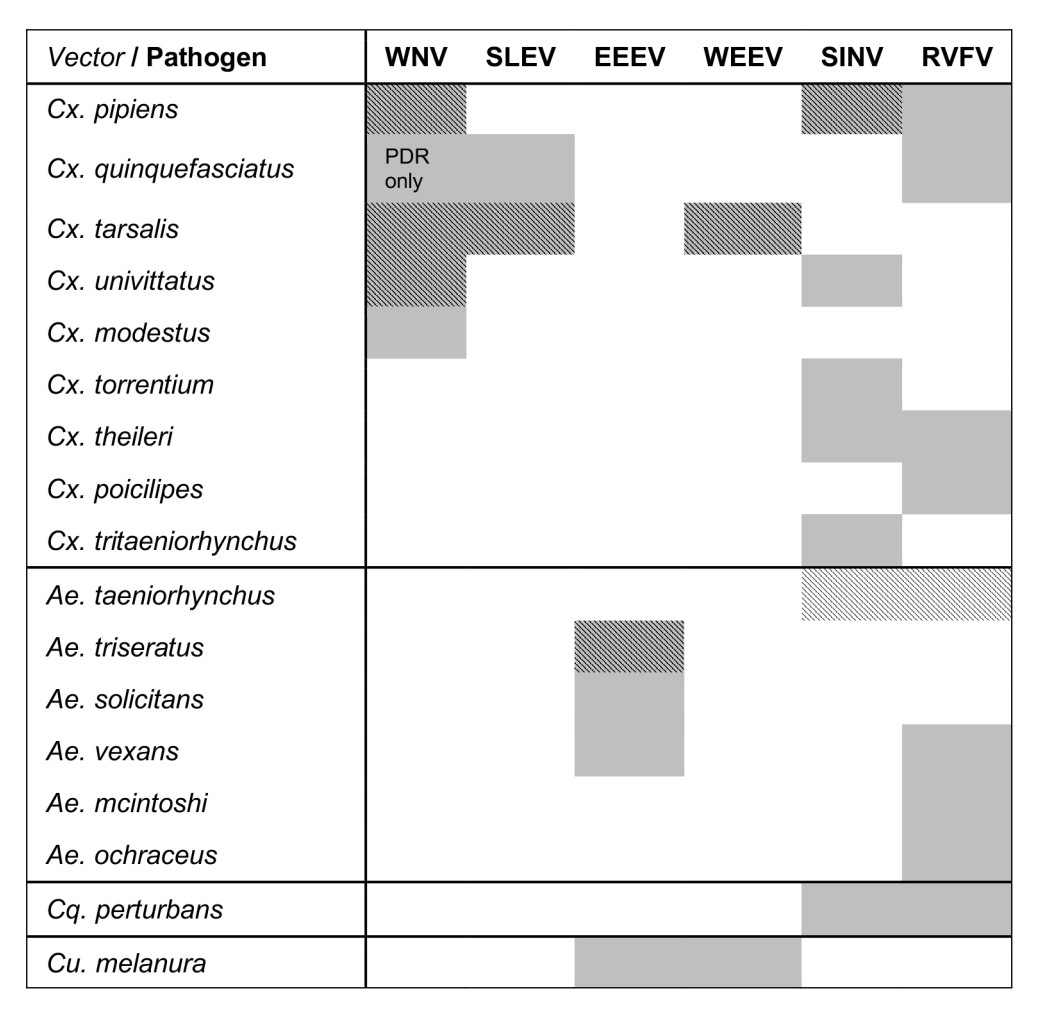

**Figure 1.** Viruses are transmitted by multiple vectors and vectors transmit multiple viruses; infection data are only available for a subset of important vector species. The six viruses in this study (WNV = West Nile virus, SLEV = St. Louis Encephalitis virus, EEEV = Eastern Equine Encephalitis virus, WEEV = Western Equine Encephalitis virus, SINV = Sindbis virus, RVFV = Rift Valley Fever virus) and the *Culex* (*Cx*.), *Aedes* (*Ae*.), *Coquillettidia* (*Cq*.), and *Culiseta* (*Cs*.) vectors that are most important for sustaining transmission to humans according to the following sources: (***Adouchief et al., 2016***; ***Braack et al., 2018***; ***Golding et al., 2012***; ***Linthicum et al., 2016***; ***Sang et al., 2017***; ***Weaver and Barrett, 2004***). The importance of each vector for transmission varies over the geographic range of the virus, and this list of vectors is not exhaustive for any virus (see sources for more complete lists of confirmed and potential vectors). Grey shading indicates an important vector-virus pair; hatching indicates available temperature-dependent data for infection traits (pathogen development rate [*PDR*] and vector competence [*bc*], which is comprised of infection efficiency [*c*] and transmission efficiency [*b*]). Infection data were available for SINV and RVFV in *Ae. taeniorhynchus*, although this North American mosquito does not occur in the endemic range of these pathogens.

*Cx. quinquefasciatus* overlap in their home ranges in Africa, they have expanded into distinct regions globally (*Figure 2*; *Farajollahi et al., 2011*). *Cx. pipiens* occurs in higher latitude temperate areas in the Northern and Southern hemisphere, while *Cx. quinquefasciatus* occurs in lower latitude temperate and tropical areas (*Figure 2A*). By contrast, *Cx. tarsalis* is limited to North America but spans the tropical-temperate gradient (*Figure 2B*). In this system of shared pathogens and vectors with distinct geographical distributions, we also test the hypothesis that differences in thermal performance underlie variation in vector and pathogen geographic distributions, since temperate environments have cooler temperatures and a broader range of temperatures than tropical environments. We also include thermal responses from other relevant vector or laboratory model species in some models:

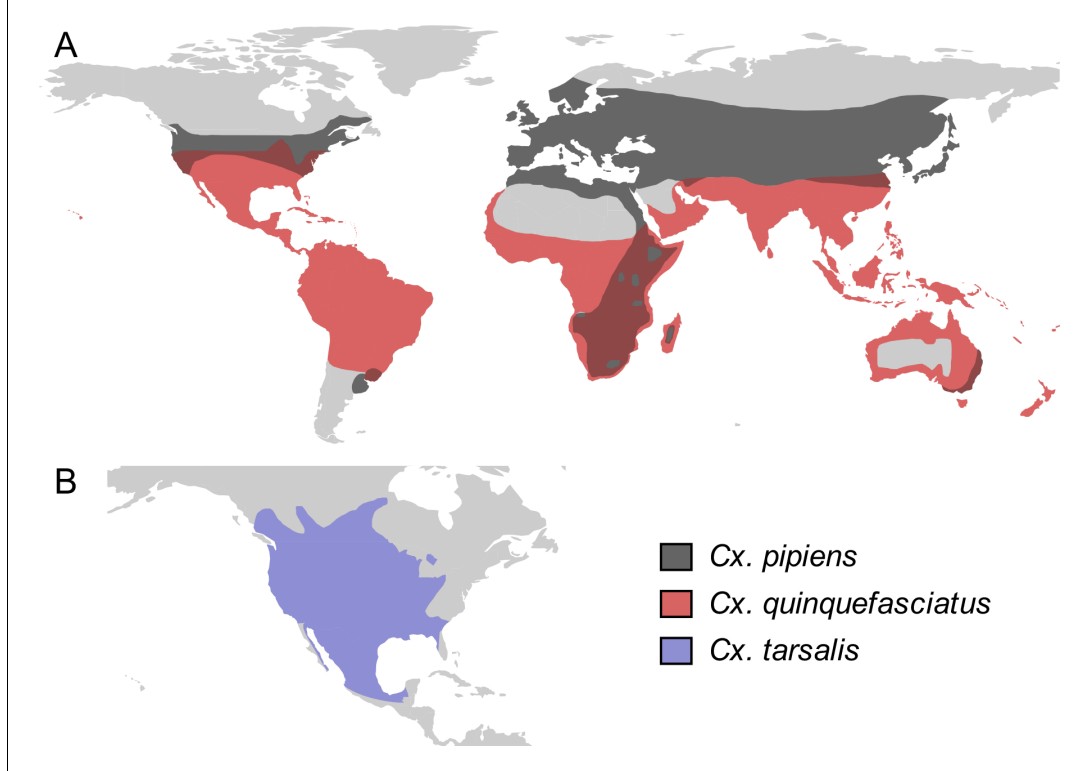

**Figure 2.** *Culex* spp. vectors of West Nile and other viruses have distinct but overlapping geographic distributions. The geographic distribution of the primary vectors of West Nile virus: (**A**) *Culex pipiens* (dark grey) and *Cx. quinquefasciatus* (red), adapted from *Farajollahi et al., 2011*; *Smith and Fonseca, 2004*; (**B**) *Cx. tarsalis* (blue), northern boundary from *Darsie and Ward, 2016*, southern boundary based on data from the Global Biodiversity Information Facility. Figure created by Michelle Evans for this paper.

*Aedes taeniorhynchus* (SINV and RVFV), *Ae. triseriatus* (EEEV), *Ae. vexans* (RVFV), *Cx. theileri* (RVFV), and *Culiseta melanura* (EEEV). Additionally, we compare our results to previously published models (*Johnson et al., 2015*; *Mordecai et al., 2017*; *Mordecai et al., 2013*; *Shocket et al., 2018*; *Tesla et al., 2018*) for transmission of more tropical diseases by the following vectors: *Ae. aegypti*, *Ae. albopictus*, *Anopheles* spp., and *Cx. annulirostris*.

We use a mechanistic approach to characterize the effects of temperature on vector–virus pairs in this network using the thermal responses of traits that drive transmission. Specifically, we use experimental data to measure the thermal responses of the following traits: vector survival, biting rate, fecundity, development rate, competence for acquiring and transmitting each virus, and the extrinsic incubation rate of the virus within the vector. We ask: (1) Do these vectors have qualitatively similar trait thermal responses to each other, and to vectors from previous studies? (2) Is transmission of disease by these vectors predicted to be optimized and limited at similar temperatures, compared to each other and to other mosquito-borne diseases in previous studies? (3) How do the thermal responses of transmission vary across vectors that transmit the same virus and across viruses that share a vector? (4) Which traits limit transmission at low, intermediate, and high temperatures? Broadly, we hypothesize that variation in thermal responses is predictable based on vectors' and viruses' geographic ranges.

Mechanistic models allow us to incorporate nonlinear effects of temperature on multiple traits, measured in controlled laboratory experiments across a wide thermal gradient, to understand their combined effect on disease transmission. This approach is critical when making predictions for future climate regimes because thermal responses are almost always nonlinear, and therefore current temperature–transmission relationships may not extend into temperatures beyond those currently observed in the field. We use Bayesian inference to quantify uncertainty and to rigorously incorporate prior knowledge of mosquito thermal physiology to constrain uncertainty when data are sparse

(*Johnson et al., 2015*). The mechanistic modeling approach also provides an independently generated, a priori prediction for the relationship between temperature and transmission to test with observational field data on human cases, allowing us to connect data across scales, from individual-level laboratory experiments, to population-level patterns of disease transmission, to climate-driven geographic variation across populations. Using this approach, we build mechanistic models for 10 vector–virus pairs by estimating thermal responses of the traits that drive transmission. We validate the models using observations of human cases in the US over space (county-level) and time (month-of-onset). The validation focuses on WNV because it is the most common of the diseases we investigated and has the most complete temperature-dependent trait data. Preliminary results of this study—the thermal responses for traits and relative $R_0$ models—were included in a review and synthesis article that was published last year (*Mordecai et al., 2019*). The present publication presents the complete methods and results, describes the vector and pathogen ecology in more detail, and provides original analyses of human case data.

## Model overview

To understand the effect of temperature on transmission and to compare the responses across vector and virus species, we used $R_0$—the basic reproduction number (*Diekmann et al., 2010*). We use $R_0$ as a static, relative metric of temperature suitability for transmission that incorporates the nonlinear effects of constant temperature on multiple traits (*Dietz, 1993*; *Mordecai et al., 2019*; *Rogers and Randolph, 2006*) and is comparable across systems, rather than focusing on its more traditional interpretation as a threshold for disease invasion into a susceptible population. Temperature variation creates additional nonlinear effects on transmission (*Huber et al., 2018*; *Lambrechts et al., 2011*; *Murdock et al., 2017*; *Paaijmans et al., 2010*) that are not well-captured by $R_0$, (*Bacaër, 2007*; *Bacaër and Ait Dads, 2012*; *Bacaër and Guernaoui, 2006*; *Diekmann et al., 2010*; *Parham and Michael, 2010*) but could be incorporated in future work by integrating the thermal performance curves fit here over the observed temperature regime.

The basic $R_0$ model for mosquito-borne diseases, originally developed for malaria (*Equation 1*; *Dietz, 1993*), includes the following traits that depend on temperature (*T*): adult mosquito mortality rate (μ, the inverse of lifespan [*lf*]), biting rate (*a*, the inverse of the gonotrophic [oviposition] cycle duration), pathogen development rate (*PDR*, the inverse of the extrinsic incubation period: the time required for exposed mosquitoes to become infectious), and vector competence (*bc*, the proportion of exposed mosquitoes that become infectious), where all rates are measured in inverse days. Vector competence is the product of infection efficiency (*c*, the proportion of exposed mosquitoes that develop a disseminated infection) and transmission efficiency (*b*, the proportion of infected mosquitoes that become infectious, with virus present in saliva). Mosquito density (*M*) also depends on temperature but is not an organism-level trait that can be measured in a laboratory setting. Two parameters do not depend on temperature: host density (*N*) and the rate at which infected hosts recover and are no longer infectious (*r*).

$$\text{Basic } R_0 : R_0(T) = \left( \frac{a(T)^2 bc(T) e^{-\frac{\mu(T)}{PDR(T)}} M(T)}{N\, r\, \mu(T)} \right)^{1/2} \tag{1}$$

Because host density (*N*) and recovery rate (*r*) are not temperature-dependent, we omit them from our model (*Equation 2*), which isolates the effect of temperature on transmission (see explanation of 'relative $R_0$' versus absolute $R_0$ below). As in previous work (*Johnson et al., 2015*; *Mordecai et al., 2019*; *Mordecai et al., 2017*; *Mordecai et al., 2013*; *Parham and Michael, 2010*; *Shocket et al., 2018*; *Tesla et al., 2018*), we extend the basic $R_0$ model to account for the effects of temperature on mosquito density (*M*) via additional temperature-sensitive life history traits (*Equation 2*): fecundity (as eggs per female per day, *EFD*), egg viability (proportion of eggs hatching into larvae, *EV*), proportion of larvae surviving to adulthood (*pLA*), and mosquito development rate (*MDR*, the inverse of the development period).

$$\text{Full } R_0 : R_0(T) = \left( \frac{a(T)^2 bc(T) e^{-\frac{\mu(T)}{PDR(T)}} EFD(T) EV(T) pLA(T) MDR(T)}{\mu(T)^3} \right)^{1/2} \tag{2}$$

Fecundity data were only available as eggs per female per gonotrophic cycle (*EFGC*; for Cx. *pipiens*) or eggs per raft (*ER*; for Cx. *quinquefasciatus*). Thus, we further modified the model to obtain the appropriate units for fecundity: we added an additional biting rate (*a*) term to the numerator (to divide by the length of the gonotrophic cycle, *Equations A1 and A2*) and for *Cx. quinquefasciatus* we also added a term for the proportion of females ovipositing (*pO*; *Equation A2*).

We parameterized the full temperature-dependent $R_0$ model (*Equation 2*) for each relevant vector–virus pair using previously published data. We conducted a literature survey to identify studies that measured the focal traits at three or more constant temperatures in a controlled laboratory experiment. From these data, we fit thermal responses for each trait using Bayesian inference. This approach allowed us to quantify uncertainty and formally incorporate prior data (*Johnson et al., 2015*) to constrain fits when data for the focal species were sparse or only measured on a limited portion of the temperature range (see *Material and Methods* for details).

For each combination of trait and species, we selected the most appropriate of three functional forms for the thermal response. As in previous work (*Johnson et al., 2015*; *Mordecai et al., 2019*; *Mordecai et al., 2017*; *Mordecai et al., 2013*; *Shocket et al., 2018*; *Tesla et al., 2018*), we fit traits with a symmetrical unimodal thermal response with a quadratic function (*Equation 3*) and traits with an asymmetrical unimodal thermal response with a Briére function (*Briere et al., 1999*; *Equation 4*). For some asymmetrical responses (e.g. pathogen development rate [*PDR*] for most vector–virus pairs), we did not directly observe a decrease in trait values at high temperatures due to a limited temperature range. In these cases, we chose to fit a Briére function based on previous studies with wider temperature ranges (*Mordecai et al., 2017*; *Mordecai et al., 2013*; *Paull et al., 2017*; *Shocket et al., 2018*) and thermal biology theory (*Amarasekare and Savage, 2012*); the upper thermal limit for these fits did not limit transmission in the $R_0$ models, and therefore did not impact the results. Unlike in previous work, lifespan data for all vectors here exhibited a monotonically decreasing thermal response over the range of experimental temperatures available. We fit these data using a piecewise linear function (*Equation 5*) that plateaued at the coldest observed data point. By assuming a plateau, rather than extrapolating that lifespan continues to increase at temperatures below those measured in the laboratory, this approach is conservative, ensuring that lifespan was not a major driver of the temperature-dependence of $R_0$ at temperatures where it was not measured and that the $R_0$ models were instead constrained at reasonable temperatures by other traits. It is also consistent with the observed natural history of two of the vector species. To overwinter, *Cx. pipiens* and *Cx. tarsalis* enter reproductive diapause and hibernate (*Nelms et al., 2013*; *Vinogradova, 2000*), and *Cx. pipiens* can survive temperatures at or near freezing (0˚C) for several months (*Vinogradova, 2000*). *Cx. quinquefasciatus* enters a non-diapause quiescent state (*Diniz et al., 2017*; *Nelms et al., 2013*) and is likely less tolerant of cold stress, but we wanted a consistent approach across models and other traits constrained the lower thermal limit of the *Cx. quinquefasciatus* $R_0$ model to realistic temperatures. All vectors are likely to exhibit decreased lifespans at extremely low temperatures (near or below 0˚C), limiting the accuracy of our inferred lifespan thermal performance curve at these temperatures.

$$\text{Quadratic function}: f(T) = -q(T - T_{min})(T - T_{max}) \tag{3}$$

$$\text{Briére function}: f(T) = q \cdot T(T - T_{min})\sqrt{(T_{max} - T)} \tag{4}$$

$$\text{Linear function}: f(T) = -mT + z \tag{5}$$

In the quadratic and Briére functions of temperature (*T*), the trait values depend on a lower thermal limit ($T_{min}$), an upper thermal limit ($T_{max}$), and a scaling coefficient (*q*). In the linear function, the trait values depend on a slope (*m*) and intercept (*z*).

The fitting via Bayesian inference produced posterior distributions for each parameter in the thermal response functions (*Equations 3, 4, 5*) for each trait–species combination. These posterior distributions represent the estimated uncertainty in the parameters. We used these parameter distributions to calculate distributions of expected mean thermal performance functions for each trait over a temperature gradient (from 1˚C to 45˚C by 0.1˚C increments). Then we substituted these samples from the distributions of the thermal responses for each trait into *Equation 2* to calculate

the posterior distributions of predicted $R_0$ over this same temperature gradient for each vector–virus pair (see Material and methods and Appendix 1 for details). Thus, the estimated uncertainty in the thermal response of each trait is propagated through to $R_0$ and combined to produce the estimated response of $R_0$ to temperature, including the uncertainty in $R_0(T)$.

Because the magnitude of realized $R_0$ depends on system-specific factors like breeding habitat availability, reservoir and human host availability, vector control, species interactions, and additional climate factors, we focused on the relative relationship between $R_0$ and temperature (*Mordecai et al., 2019*). We rescaled the $R_0$ model results to range from 0 to 1 (i.e. 'relative $R_0$'), preserving the temperature-dependence (including the absolute thermal limits and thermal optima) while making each model span the same scale. To compare trait responses and $R_0$ models, we quantify three key temperature values: the optimal temperature for transmission ($T_{opt}$) and the lower and upper thermal limits ($T_{min}$ and $T_{max}$, respectively) where temperature is predicted to prohibit transmission ($R_0 = 0$).

## Results

### Trait thermal responses

We fit thermal response functions from empirical data for all of the vector and virus traits that affect transmission for which data were available (*Figure 1* and *Appendix 1—table 1*). All mosquito traits were temperature-sensitive (three main *Culex* species: *Figure 3*, *Figure 4*; *Ae. taeniorhynchus*, *Ae. triseriatus*, *Ae. vexans*, *Cx. theileri*, and *Cs. melanura*: *Appendix 1—figure 1*). For most species, the extensive data for larval traits (mosquito development rate [MDR] and survival [pLA]) produced clear unimodal thermal responses with relatively low uncertainty (*Figure 3A,B*, *Appendix 1—figure 1A, B*). For biting rate (*a*) and fecundity traits (proportion ovipositing [pO], eggs per female per gonotrophic cycle [EFGC], or per raft [ER], and egg viability [EV]), trait data were often more limited and fits were more uncertain, but still consistent with the expected unimodal thermal responses based on previous studies (*Mordecai et al., 2017*; *Mordecai et al., 2013*; *Paull et al., 2017*; *Shocket et al., 2018*) and theory (*Amarasekare and Savage, 2012*; *Figure 3C*, *Figure 4*, *Appendix 1—figure 1C-F*). However, adult lifespan (*lf*) data clearly contrasted with expectations from previous studies of more tropical mosquitoes. Lifespan (*lf*) decreased linearly over the entire temperature range of available data (coldest treatments: 14–16°C, *Figure 3D*; 22°C, *Appendix 1—figure 1D*) instead of peaking at intermediate temperatures (e.g. previously published optima for more tropical species: 22.2–23.4°C) (*Johnson et al., 2015*; *Mordecai et al., 2017*; *Mordecai et al., 2013*; *Shocket et al., 2018*; *Tesla et al., 2018*).

The thermal responses for pathogen development rate (*PDR*) were similar among most vector–virus pairs (*Figure 5*), with a few notable exceptions: WNV in *Cx. quinquefasciatus* had a warmer lower thermal limit (*Figure 5A*); WNV in *Cx. univittatus* had a cooler optimum and upper thermal limit (*Figure 5A*); and SINV in *Ae. taeniorhynchus* had limited data that indicated very little response to temperature (*Figure 5C*). By contrast, the thermal response of vector competence (*bc*) and its component traits varied substantially across vectors and viruses (*Figure 6*). For example, infection efficiency (*c*) of *Cx. pipiens* peaked at warmer temperatures for WNV than for SINV (*Figure 6A,G*; 95% CIs: SINV = 14.1–30.5°C, WNV = 31.9–36.1°C), transmission efficiency (*b*) of *Cx. tarsalis* peaked at warmer temperatures for WNV and SLEV than for WEEV (*Figure 6B,E,H*; CIs: WEEV = 19.2–23.2°C, SLEV = 23.5–29.7°C, WNV = 23.9–29.3°C), and the lower thermal limit for vector competence (*bc*) for WNV was much warmer in *Cx. pipiens* than in *Cx. univittatus* (*Figure 6C*; CIs: *Cx. univittatus* = 1.5–7.1°C, *Cx. pipiens* = 15.0–17.9°C). Infection data (used to calculate pathogen development rate [PDR] and vector competence [bc]) for RVFV and SINV were only available in *Ae. taeniorhynchus*, a New World species that is not a known vector for these viruses in nature.

### Temperature-dependent R$_0$ models

Relative $R_0$ responded unimodally to temperature for all the vector–virus pairs, with many peaking at fairly cool temperatures (medians: 22.7–26.0°C, see *Table 2* for CIs; *Figure 7*). The lower thermal limits (medians: 8.7–19.0°C, see *Table 2* for CIs; *Figure 7*) were more variable than the optima or the upper thermal limits (medians: 31.9–37.8°C, see *Table 2* for CIs; *Figure 7*), although confidence intervals overlapped in most cases because lower thermal limits also had higher uncertainty

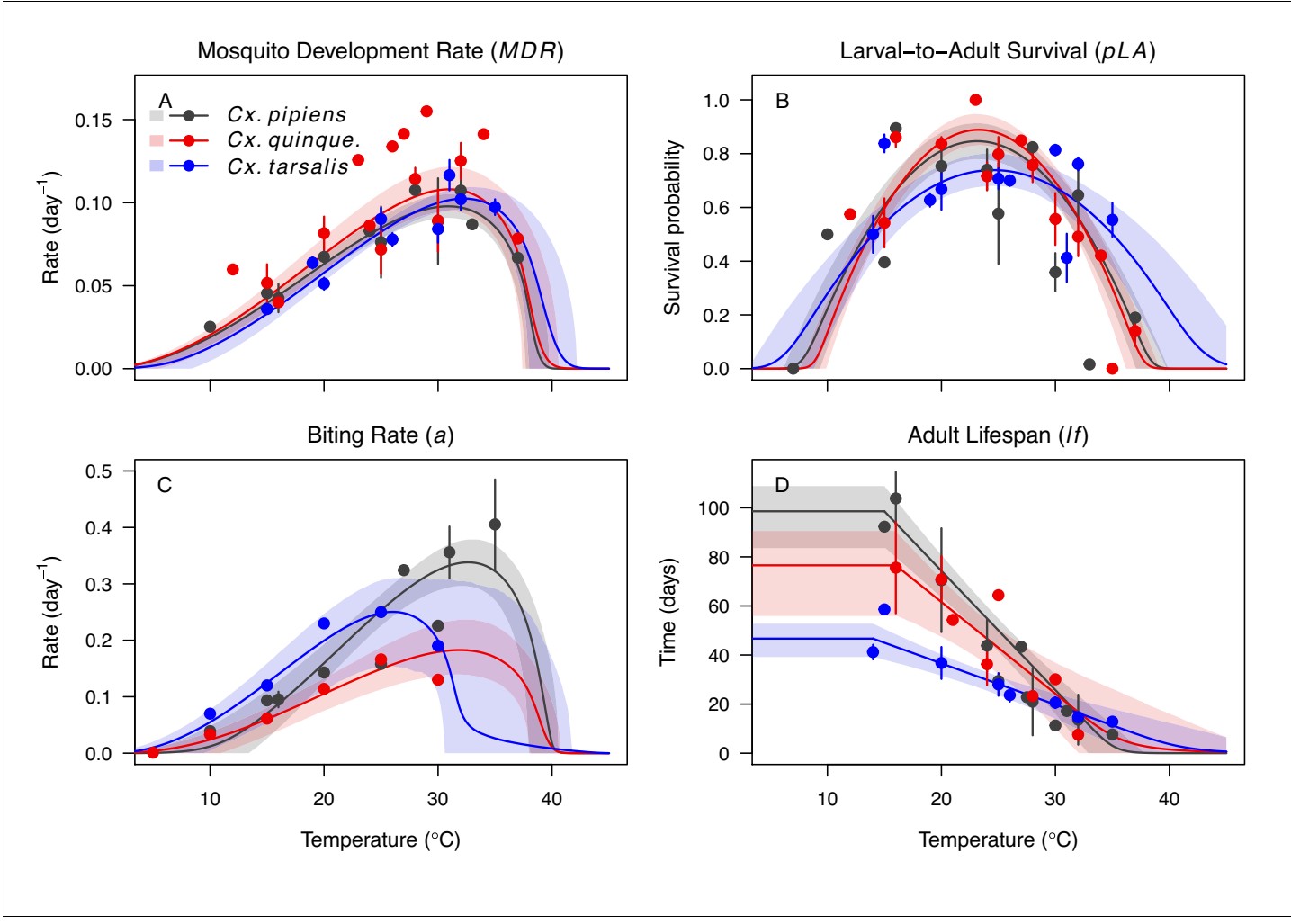

**Figure 3.** *Culex* spp. mosquito traits respond strongly and consistently to temperature. The thermal responses of mosquito traits for the North American vectors of West Nile virus: *Culex pipiens* (dark grey), *Cx. quinquefasciatus* (red), and *Cx. tarsalis* (blue). (A) Mosquito development rate (*MDR*), (B) larval-to-adult survival (*pLA*), (C) biting rate (*a*), and (D) adult lifespan (*lf*). Points without error bars are reported means from single studies; points with error bars are averages of means from multiple studies (+ / - standard error, for visual clarity only; thermal responses were fit to reported means, see *Appendix 1—figures 2*, *3*, *4*, *5*). Solid lines are posterior means; shaded areas are 95% credible intervals of the trait mean. See *Appendix 1—figure 1* for thermal responses for *Aedes taeniorhynchus*, *Ae. triseriatus*, *Ae. vexans*, and *Culiseta melanura*. The mean thermal responses for these traits were printed in *Mordecai et al., 2019* (as part of Figure 3 in that paper) without the trait data and 95% CIs, and along with thermal responses for six other vectors. See *Appendix 1—tables 2*, *3*, *6* for data sources and *Appendix 1—tables 7*, *8*, *9* for priors.

(*Figure 7*). The *Ae. taeniorhynchus* models (both unnatural vector-pathogen pairs) were clear outliers, with much warmer distributions for the upper thermal limits, and optima that trended warmer as well.

Differences in relative $R_0$ stemmed from variation both in vector traits (e.g. in *Figure 7A*, with WNV in different vector species) and in virus infection traits (e.g. in *Figure 7B*, with different viruses in *Cx. tarsalis*). The upper thermal limit was warmer for WNV transmitted by *Cx. pipiens* (34.9°C [CI: 32.9–37.5°C]) than by *Cx. quinquefasciatus* (31.8°C [CI: 31.1–32.2°C]), counter to the a priori prediction based on the higher-latitude range of *Cx. pipiens* in North America, South America, and Europe (*Figure 2*). This result implies that warming from climate change may differentially impact transmission by these two vectors. Additionally, the lower thermal limit for WNV varied widely (but with slightly overlapping 95% CIs) across different vector species (*Figure 7D*), from 19.0°C (14.2–21.0°C) in *Cx. quinquefasciatus* to 16.8°C (14.9–17.8°C) in *Cx. pipiens* to 12.2°C (9.7–15.3°C) in *Cx. tarsalis* to 11.1°C (8.1–15.4°C) in *Cx. univittatus* (an African and Eurasian vector; *Table 2*). Based on these trends

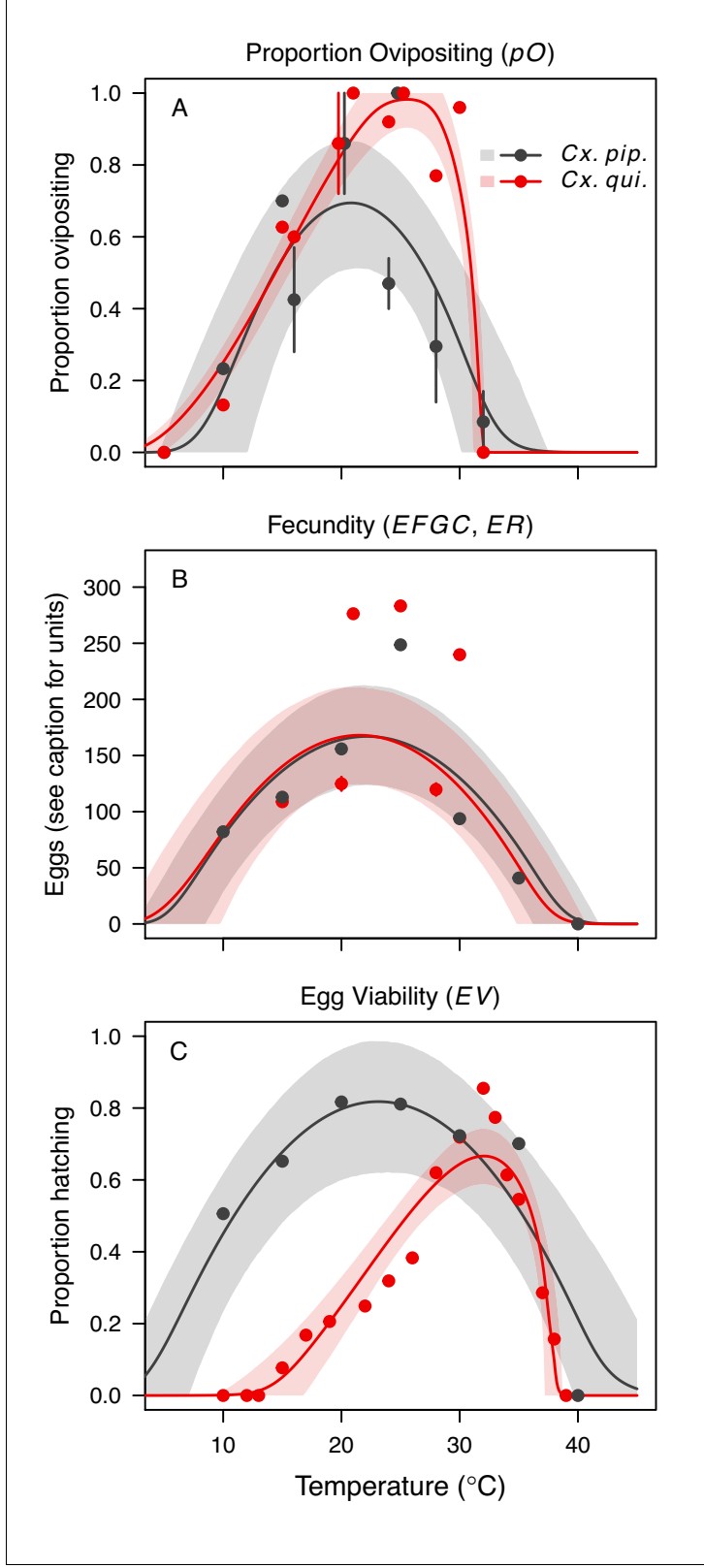

**Figure 4.** *Culex pipiens* and *Cx. quinquefasciatus* reproductive traits respond strongly to temperature but with different functional forms. The thermal responses of mosquito traits for the primary vectors of West Nile virus: *Culex pipiens* (dark grey) and *Cx. quinquefasciatus* (red). (A) Proportion ovipositing (*pO*), (B) fecundity (as eggs per female per gonotrophic cycle [*EFGC*] in *Cx. pipiens,* and eggs per raft, [*ER*] in *Cx. quinequefasciatus*), and (C) egg
*Figure 4 continued on next page*

*Figure 4 continued*

viability (*EV*). Points without error bars are reported means from single studies; points with error bars are averages of means from multiple studies (+ / - standard error, for visual clarity only; thermal responses were fit to reported means, see *Appendix 1—figure 6*). Solid lines are posterior distribution means; shaded areas are 95% credible intervals of the trait mean. See *Appendix 1—figure 1* for thermal responses for *Ae. vexans*, *Cx. theileri*, and *Culiseta melanura*. The mean thermal responses for these traits were printed in *Mordecai et al., 2019* (as part of Figure 3 in that paper) without the trait data and 95% CIs, and along with thermal responses for six other vectors. See *Appendix 1—table 2* for data sources and *Appendix 1—table 7* for priors.

in the thermal limits of $R_0$, the seasonality of transmission and the upper latitudinal and elevational limits could vary for WNV transmitted by these different species.

Different traits determined the lower and upper thermal limits and optimum for transmission across vector–virus pairs. The lower thermal limit for transmission was most often determined by pathogen development rate (*PDR*; WNV and SLEV in *Cx. tarsalis*, WNV in *Cx. quinquefasciatus*) or biting rate (*a*; WNV in *Cx. univitattus*, WEEV in *Cx. tarsalis*, EEEV in *Ae. triseriatus,* RVFV and SINV in *Ae. taeniorhynchus*, SINV in *Cx. pipiens*; *Appendix 1—figures 12–20*), which tend to respond asymmetrically to temperature, with high optima and low performance at low temperatures. However, vector competence (*bc*) determined the lower limit for WNV in *Cx. pipiens* (*Appendix 1—figure 11*). The upper thermal limit was determined by biting rate (*a*) for the three *Cx. tarsalis* models and by adult lifespan (*lf*) for all others, although proportion ovipositing (*pO*) was also important for WNV in *Cx. quinquefasciatus* (*Appendix 1—figures 11–20*). In all models, lifespan (*lf*) and biting rate (*a*) had the strongest impact on the optimal temperature for transmission, with biting rate increasing transmission at low temperatures and lifespan decreasing transmission at high temperatures (*Appendix 1—figures 11–20*). This result is consistent with previous mechanistic models of tropical mosquito-borne diseases, despite the qualitative difference in the shape of the lifespan thermal response between those tropical mosquitoes and the more temperate mosquitoes investigated here (*Johnson et al., 2015*; *Mordecai et al., 2017*; *Mordecai et al., 2013*; *Shocket et al., 2018*; *Tesla et al., 2018*).

## Model validation with human case data

We validated the $R_0$ models for WNV with independent data on human cases because the temperature-dependent trait data for those models were relatively high quality and because human case data were available from the Centers for Disease Control and Prevention across a wide climatic gradient in the contiguous United States. We averaged county-level incidence and mean summer temperatures across the entire period from 2001 to 2016 to estimate the impact of temperature over space, while ignoring interannual variation in disease that is largely driven by changes in host immunity and drought (*Paull et al., 2017*). We used generalized additive models (GAMs, which produce flexible, smoothed responses) to ask: does average incidence respond unimodally to mean summer temperature? If so, what is the estimated optimal temperature for transmission? Can we detect upper or lower thermal limits for transmission? Incidence of human neuroinvasive West Nile disease responded unimodally to average summer temperature and peaked at 24°C (23.5–24.2°C depending on the spline settings; *Figure 8*, *Appendix 1—figure 24*), closely matching the optima from the mechanistic models for the three North American *Culex* species (23.9–25.2°C; *Table 2*). However, the human disease data did not show evidence for lower or upper thermal limits: mean incidence remained positive and with relatively flat slopes below ~19°C and above ~28°C, although sample size was very low above 28°C and below 15°C resulting in wide confidence intervals (*Figure 8*, *Appendix 1—figure 24*).

We used national month-of-onset data for WNV, EEEV, and SLEV to ask: is the seasonality of incidence consistent with our models for temperature-dependent transmission? The month-of-onset for cases of WNV was consistent with predicted transmission, $R_0(T)$ (*Figure 9*). As expected (based on previous studies and the time required for mosquito populations to increase, become infectious, and bite humans, and for humans to present symptoms and seek medical care [*Mordecai et al., 2017*; *Shocket et al., 2018*]), there was a 2-month lag between initial increases in $R_0(T)$ and incidence: cases began rising in June to the peak in August. The dramatic decline in transmission between September and October corresponds closely to the predicted decline in relative $R_0$, but

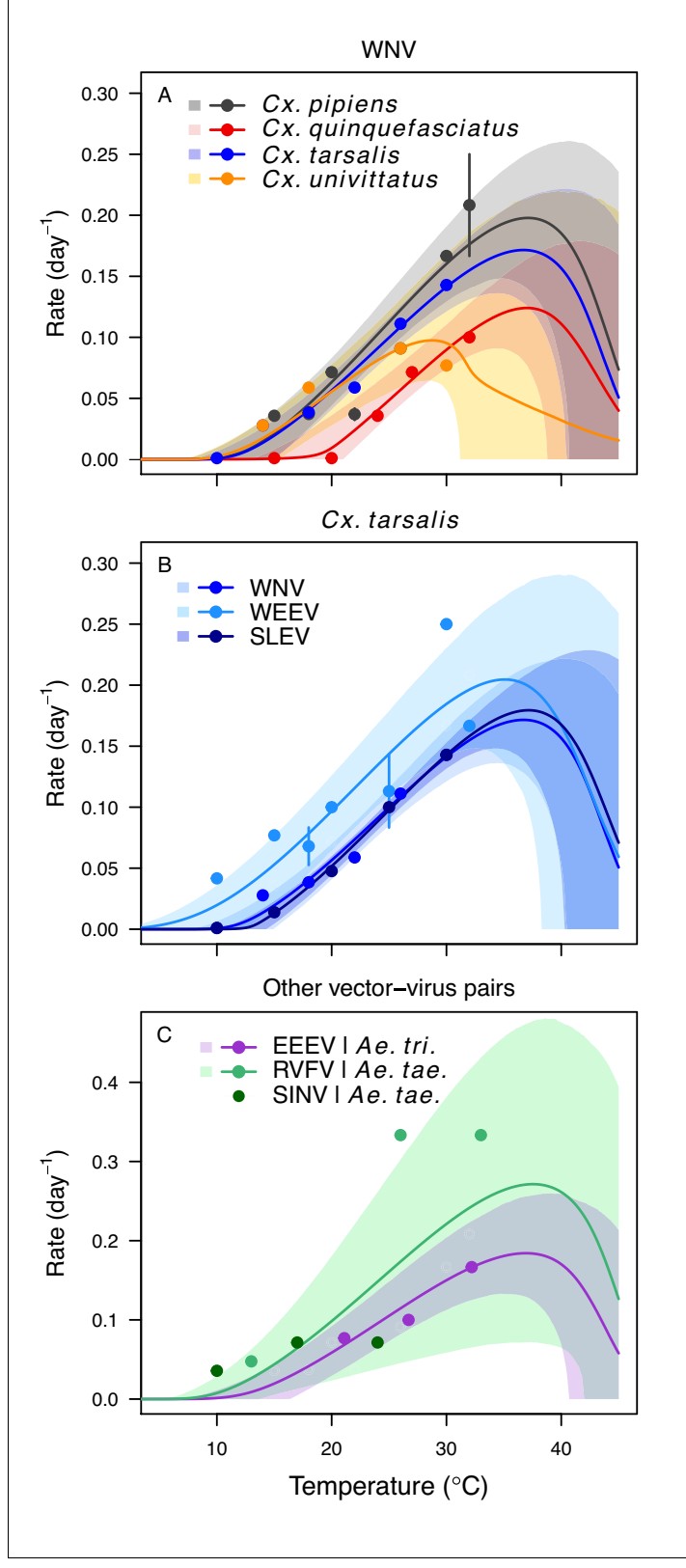

**Figure 5.** Pathogen development rates (PDR) have high thermal optima. Thermal responses of pathogen development rate (*PDR*). (A) West Nile virus in *Culex pipiens* (dark grey), *Cx. quinquefasciatus* (red), *Cx. tarsalis* (blue), and *Cx. univitattus* (orange). (B) Three viruses in *Cx. tarsalis*: West Nile virus (same as in A, blue), Western Equine Encephalitis virus (light blue), and St. Louis Encephalitis virus (dark blue). (C) Eastern Equine Encephalitis
*Figure 5 continued on next page*

*Figure 5 continued*

virus in *Aedes triseriatus* (violet), Rift Valley Fever virus in *Ae. taeniorhynchus* (light green), Sindbis virus in *Ae. taeniorhynchus* (dark green). We did not fit a thermal response for Sindbis virus in *Ae. taeniorhynchus* because the limited data responded weakly to temperature and did not match our priors. Points without error bars are reported means from single studies; points with error bars are averages of means from multiple studies (+ / - standard error, for visual clarity only; thermal responses were fit to reported means, see *Appendix 1—figure 7*). Solid lines are posterior distribution means; shaded areas are 95% credible intervals of the trait mean. The mean thermal responses for this trait were printed in *Mordecai et al., 2019* (as part of Figure 4 in that paper) without the trait data and 95% CIs, combined into a single panel, and along with thermal responses for six other vector-pathogen pairs. See *Appendix 1—table 5* for data sources and *Appendix 1—table 8* for priors.

without the expected two-month lag. In general, the seasonal patterns of SLEV and EEEV incidence were similar to WNV, but differed by three orders of magnitude from ~20,000 cases of WNV to ~40–50 cases of EEEV and SLEV during the peak month (*Figure 9*). However, transmission of SLEV and EEEV are predicted to begin increasing 1 month earlier than WNV (March versus April, *Figure 9*), because the mechanistic models predict that the lower thermal limits for SLEV and EEEV are cooler than those for WNV in two of the three North American vectors (*Cx. pipiens* and *Cx. quinquefasciatus*, *Figure 7*). The month-of-onset data partially support this prediction, as cases of SLEV (but not EEEV) disease begin to increase earlier in the year than WNV, relative to the summer peak.

## Discussion

As the climate changes, it is critical to understand how changes in temperature will affect the transmission of mosquito-borne diseases. Toward this goal, we developed temperature-dependent, mechanistic transmission models for 10 vector–virus pairs. The viruses—West Nile virus (WNV), St. Louis Encephalitis virus (SLEV), Eastern and Western Equine Encephalitis viruses (EEEV and WEEV), Sindbis virus (SINV), and Rift Valley fever virus (RVFV)—sustain substantial transmission in temperate areas (except RVFV), and are transmitted by shared vector species, including *Cx. pipiens*, *Cx. quinquefasciatus*, and *Cx. tarsalis* (except EEEV; *Figure 1*). Although most traits responded unimodally to temperature, as expected (*Johnson et al., 2015*; *Mordecai et al., 2019*; *Mordecai et al., 2017*; *Mordecai et al., 2013*; *Shocket et al., 2018*; *Tesla et al., 2018*), lifespan decreased linearly with temperature over the entire temperature range of available data (>14°C) for these *Culex* vectors (*Figure 3*). Transmission responded unimodally to temperature, with the thermal limits and optima for transmission varying among some of the focal mosquito and virus species (*Figure 7*, *Table 2*), largely due to differences in the thermal responses of mosquito biting rate, lifespan, vector competence, and pathogen development rate. Human case data for WNV disease across the US exhibited a strong unimodal thermal response (*Figure 8*), and month-of-onset data for WNV, SLEV, and EEEV were largely consistent with the predicted seasonality of transmission (*Figure 9*). Thus, the mechanistic models captured geographical and seasonal patterns of human incidence, despite the complexity of the enzootic cycles and spillover into humans. Our analysis was somewhat limited by the lack of data for several trait-species combinations, or by data that were sparse, particularly at high temperatures. However, our key results—maximal transmission at intermediate temperatures—are unlikely to change, and underscore the importance of considering unimodal thermal responses when predicting how climate change will impact mosquito-borne disease transmission.

The monotonically decreasing thermal responses of lifespan (*lf*) within the range of the available experimental data for these more temperate mosquitoes (*Figure 3D*) contrast with the clearly unimodal responses of more tropical species (*Mordecai et al., 2019*; *Mordecai et al., 2017*; *Mordecai et al., 2013*; *Shocket et al., 2018*). This contrast may reflect differing thermal physiology between species that use diapause or quiescence, two forms of dormancy, to persist over winter and those that do not (*Diniz et al., 2017*; *Nelms et al., 2013*; *Vinogradova, 2000*). Both *Cx. pipiens* and *Cx. tarsalis* diapause (*Nelms et al., 2013*; *Vinogradova, 2000*), and *Cx. pipiens* can survive temperatures at or near freezing (0°C) for several months (*Vinogradova, 2000*). *Cx. quinquefasciatus* enters a non-diapause quiescent state (*Diniz et al., 2017*; *Nelms et al., 2013*). *Ae. albopictus*, a species that occurs in both tropical and temperate zones, exhibits a latitudinal gradient in the United States in which more temperate populations diapause while sub-tropical populations do not

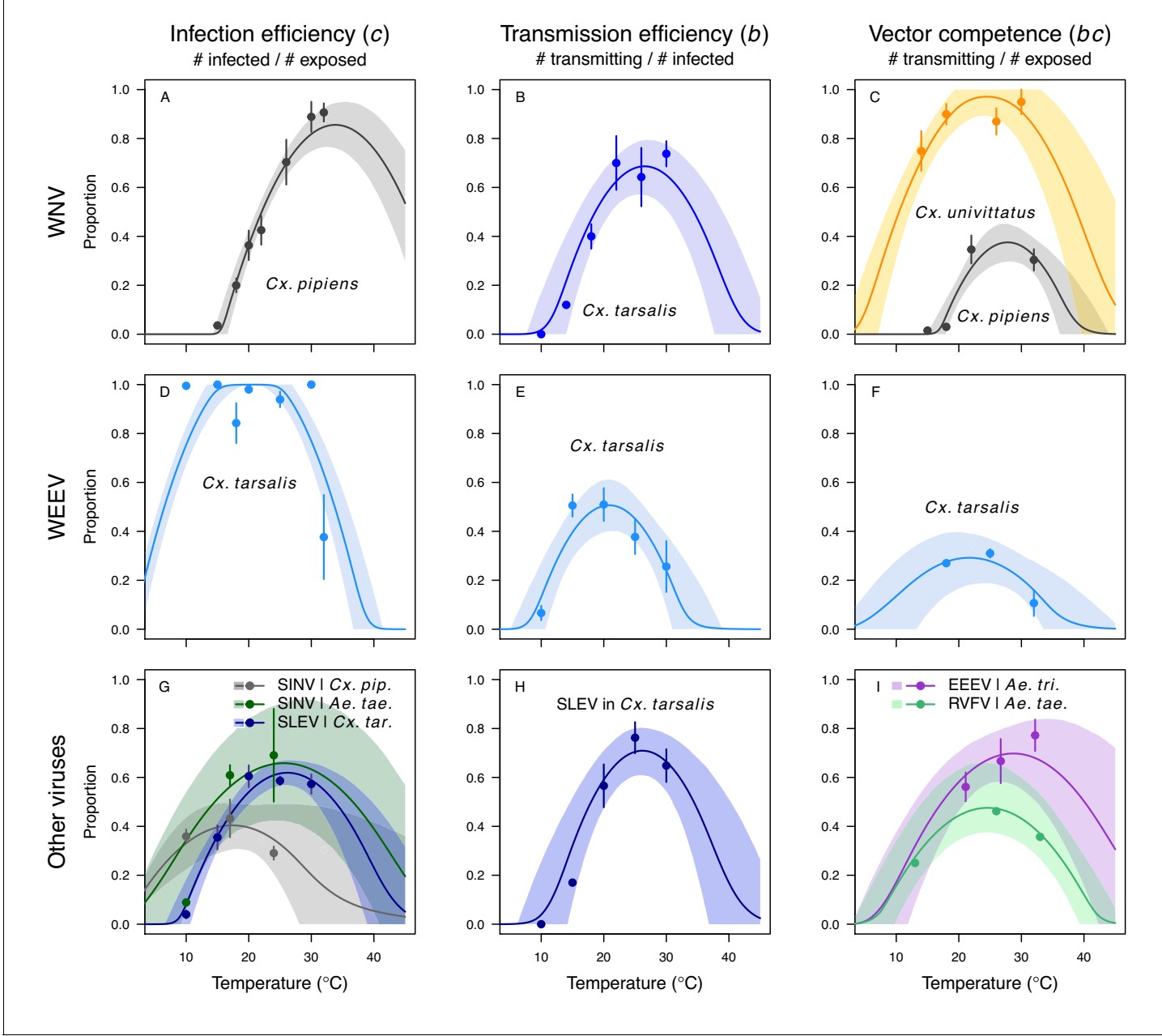

**Figure 6.** Vector competence (bc) and its component traits—infection efficiency (c) and transmission efficiency (b) respond strongly to temperature and vary across vector and virus species. Thermal responses of infection efficiency (c, # infected / # exposed; first column), transmission efficiency (b, # transmitting / # infected; second column) or vector competence (bc, # infected / # exposed; third column) for vector–virus pairs. First row (**A,B,C**): West Nile virus in *Culex pipiens* (dark grey), *Cx. tarsalis* (blue), and *Cx. univittatus* (yellow/orange). Second row: (**D,E,F**) Western Equine Encephalitis virus in *Cx. tarsalis* (light blue). Third row (**G,H,I**): Sindbis virus in *Aedes taeniorhynchus* (dark green), Sindbis virus in *Cx. pipiens* (light gray), St. Louis Encephalitis virus in *Cx. tarsalis* (dark blue), Eastern Equine Encephalitis virus in *Ae. triseriatus* (violet), and Rift Valley Fever virus in *Ae. taeniorhynchus* (light green). Points are means of replicates from single or multiple studies (+ / - standard error, for visual clarity only; thermal responses were fit to replicate-level data, see ***Appendix 1—figures 8*** and ***9***). Solid lines are posterior distribution means; shaded areas are 95% credible intervals of the trait mean. The mean thermal responses for these traits were printed in ***Mordecai et al., 2019*** (as part of Figure 4 in that paper) without the trait data and 95% CIs, combined into fewer panels, and along with thermal responses for six other vector-pathogen pairs. See ***Appendix 1—table 4*** for data sources and ***Appendix 1—table 8*** for priors.

**Table 2.** Thermal optima and limits for transmission of mosquito-borne pathogens.
Median temperature of the lower thermal limit ($T_{min}$), optimum, and upper thermal limit ($T_{max}$), with 95% credible intervals in parentheses. A version of this table (without thermal breadth, different order of $R_0$ models) was published in *Mordecai et al., 2019* (Table 2 in that paper).

| $R_0$ Model | $T_{min}$ (°C) | Optimum (°C) | $T_{max}$ (°C) | Thermal breadth (°C) |
|---|---|---|---|---|
| *From this study:* | | | | |
| EEEV in *Ae. triseriatus* | 11.7 (8.8–16.3) | 22.7 (22.0–23.6) | 31.9 (31.1–33.0) | 20.0 (15.4–23.0) |
| WEEV in *Cx. tarsalis* | 8.6 (6.3–13.0) | 23.0 (22.0–24.7) | 31.9 (30.3–35.2) | 23.3 (18.2–27.0) |
| SINV in *Cx. pipiens* | 9.4 (6.9–13.3) | 23.2 (21.7–24.6) | 33.8 (28.2–37.0) | 23.8 (17.3–28.6) |
| WNV in *Cx. univittatus* | 11.0 (8.0–15.3) | 23.8 (22.7–25.0) | 33.6 (31.2–36.9) | 22.5 (18.2–26.3) |
| WNV in *Cx. tarsalis* | 12.1 (9.6–15.2) | 23.9 (22.9–25.9) | 32.0 (30.6–38.6) | 20.1 (16.3–26.7) |
| SLEV in *Cx. tarsalis* | 12.9 (11.0–14.8) | 24.1 (23.1–26.0) | 32.0 (30.6–38.5) | 19.2 (16.5–25.6) |
| WNV in *Cx. pipiens* | 16.8 (14.9–17.8) | 24.5 (23.6–25.5) | 34.9 (32.9–37.6) | 18.2 (15.8–21.2) |
| WNV in *Cx. quinquefasciatus* | 19.0 (14.1–20.9) | 25.2 (23.9–27.1) | 31.8 (31.1–32.2) | 12.7 (10.6–17.6) |
| RVFV in *Ae. taeniorhynchus* | 10.6 (8.6–14.4) | 25.9 (23.8–27.1) | 37.8 (34.4–39.1) | 27.0 (21.8–29.7) |
| SINV in *Ae. taeniorhynchus* | 9.7 (8.3–13.6) | 26.0 (23.9–27.3) | 37.8 (34.4–39.2) | 27.7 (22.6–30.0) |
| *From previous studies:* | | | | |
| Falciparum malaria (*Johnson et al., 2015*) | 19.1 (16.0–23.2) | 25.4 (23.9–27.0) | 32.6 (29.4–34.3) | 13.2 (8.3–17.1) |
| DENV in *Ae. albopictus* (*Mordecai et al., 2017*) | 16.2 (13.0–19.8) | 26.4 (25.4–27.6) | 31.4 (29.5–34.0) | 15.2 (11.2–19.3) |
| Ross River virus (*Shocket et al., 2018*) | 17.0 (15.8–18.0) | 26.4 (26.0–26.6) | 31.4 (30.4–33.0) | 14.2 (12.8–16.2) |
| ZIKV in *Ae. aegypti* (*Tesla et al., 2018*) | 22.8 (20.5–23.8) | 28.9 (28.2–29.6) | 34.5 (34.1–36.2) | 11.7 (10.4–14.5) |
| DENV in *Ae. aegypti* (*Mordecai et al., 2017*) | 17.8 (14.6–21.2) | 29.1 (28.4–29.8) | 34.5 (34.1–35.8) | 16.7 (13.2–20.2) |

(*Urbanski et al., 2010*). Experiments could test this hypothesis by measuring whether the functional form of the thermal response for lifespan differs between northern (diapausing) and southern (non-diapausing) US *Ae. albopictus* populations. Despite the difference in the shape of the thermal response, lifespan played a similarly important role here as in previous studies of mosquito-borne pathogens, strongly limiting transmission at high temperatures (*Appendix 1—figures 11–*

**Table 3.** Predicted optima for transmission of West Nile virus.
Predicted optima for transmission from this study and previous models. A version of this table (with $R_0$ models for additional viruses and including thermal limits) was published in *Mordecai et al., 2019* (as Table 3 in that paper).

| $R_0$ Model | Optimum (°C) |
|---|---|
| *From this study:* | |
| WNV in *Cx. pipiens* | 24.5 |
| WNV in *Cx. quinquefasciatus* | 25.2 |
| WNV in *Cx. tarsalis* | 23.9 |
| WNV in *Cx. univittatus* | 23.8 |
| *From previous studies:* | |
| WNV in *Cx. pipiens* (*Paull et al., 2017*) | 24.9 |
| WNV in *Cx. quinquefasciatus* (*Paull et al., 2017*) | 24.3 |
| WNV in *Cx. tarsalis* (*Paull et al., 2017*) | 24.9 |
| WNV in *Cx. pipiens* (*Vogels et al., 2017*) | 28 |
| WNV in *Cx. pipiens molestus* (*Vogels et al., 2017*) | 28 |
| WNV in *Cx.* and *Ae. spp.* (*Kushmaro et al., 2015*) | 35 |

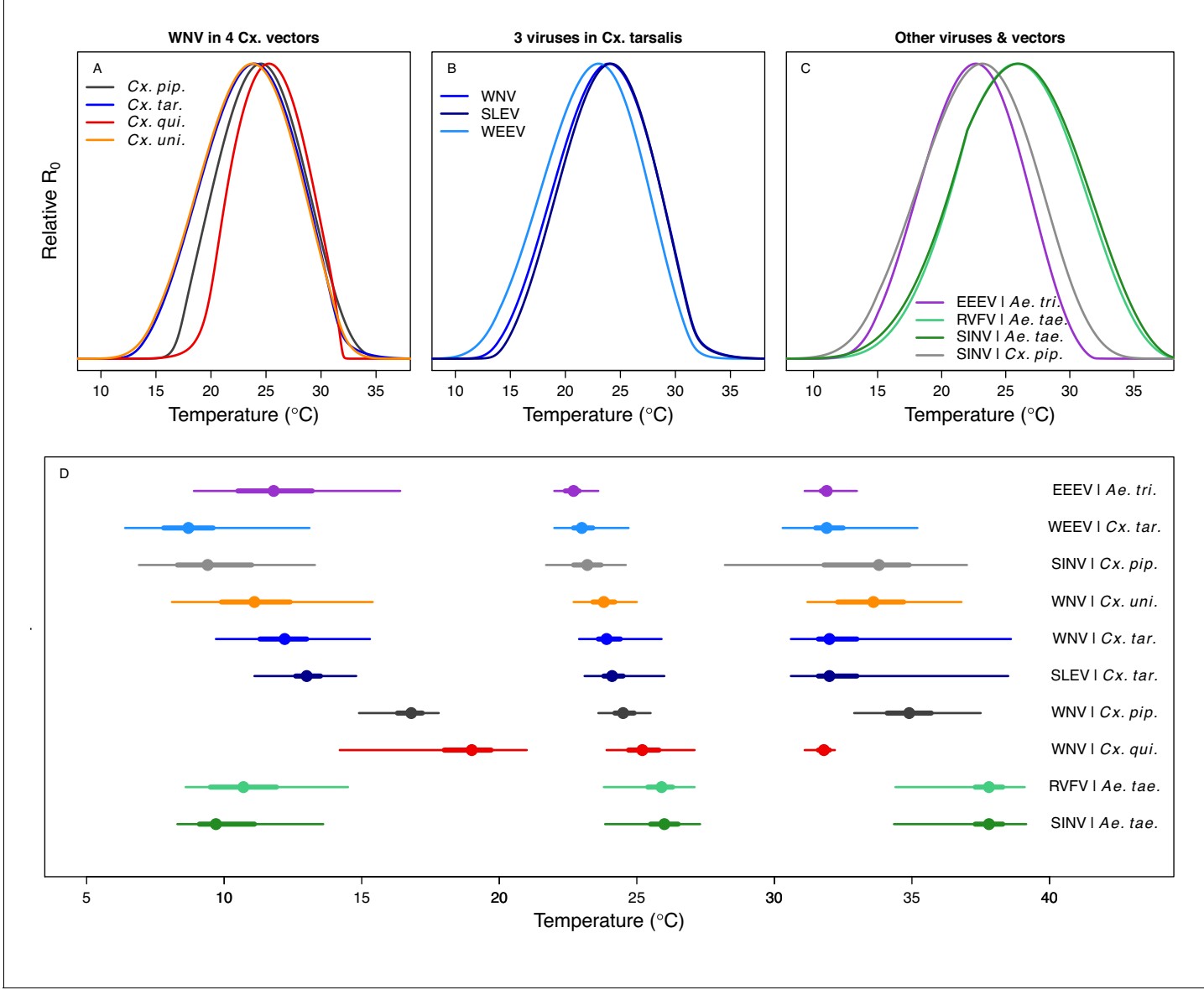

**Figure 7.** Unimodal thermal responses of transmission (relative $R_0$) for ten vector-virus pairs. Posterior mean relative $R_0$ for (**A**) West Nile virus (WNV) in *Culex pipiens* (dark grey), *Cx. tarsalis* (blue), *Cx. quinquefasciatus* (red), and *Cx. univitattus* (orange); (**B**) three viruses in *Cx. tarsalis*: WNV (same as in A, blue), Western Equine Encephalitis virus (WEEV, light blue), and St. Louis Encephalitis virus (SLEV, dark blue); (**C**) Sindbis virus (SINV) in *Aedes taeniorhynchus* (dark green) and *Cx. pipiens* (light grey), Rift Valley Fever virus (RVFV) in *Ae. taeniorhynchus* (light green), and Eastern Equine Encephalitis virus (EEEV) in *Ae. triseriatus* (violet). (**D**) Posterior median and uncertainty estimates for the lower thermal limit, optimum, and upper thermal limit. Points show medians, thick lines show middle 50% density, thin lines show 95% credible intervals. Models are ordered by increasing median optimal temperature. The thermal responses for $R_0$ were printed in *Mordecai et al., 2019* (as Figure 2 in that paper, reproduced here as *Appendix 1—table 10*), combined into two total panels and along with six other vector-pathogen pairs. See *Appendix 1—figure 21* for histograms of lower thermal limit, optimum, and upper thermal limit for each model.

20). Nonetheless, the thermal responses for lifespan here ultimately promote higher transmission at relatively cool temperatures because unlike in more tropical species, lifespan did not decline at cool temperatures within the range measured (>14˚C). Given the lack of rigorous trait data, we cannot be certain of the shape of the thermal response of lifespan (*lf*) below 14˚C, although it is almost certainly unimodal, especially at more extreme temperatures expected to be fatal even for diapausing mosquitoes (i.e. below 0˚C). Our decision to assume lifespan (*lf*) plateaued at temperatures below the observed data was based on vector natural history (*Vinogradova, 2000*) and intended to be conservative. This approach ensured that lifespan was not a major driver of the temperature-dependence

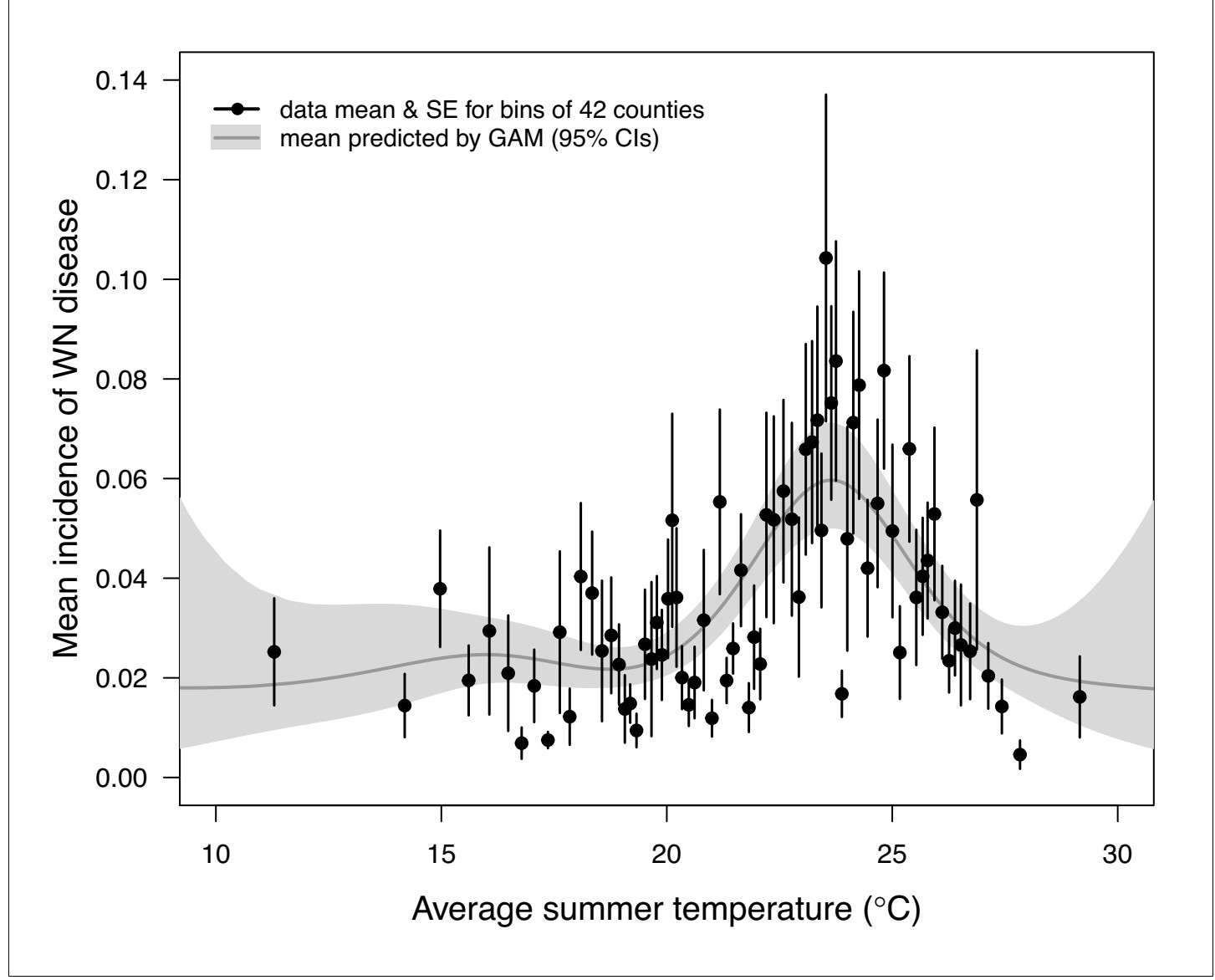

**Figure 8.** Incidence of human neuroinvasive West Nile disease across US counties responds unimodally to temperature, peaking at 24°C. Grey line: predicted mean incidence from a generalized additive model (GAM) fit to county-level data (n = 3109) of mean temperature from May-September and incidence of neuroinvasive West Nile disease per 1000 people, both averaged from 2001 to 2016. Black points: mean incidence (with standard error bars) for bins of 42 counties (for visual clarity). See *Appendix 1—figure 24* for fits across a range of smoothing parameters. See *Appendix 1—figure 25* for LOESS (moving average) fits of the data. A version of the LOESS analysis was published as Figure S3 in *Mordecai et al., 2019*.

of $R_0$ at temperatures where it was not measured and that $R_0$ was instead constrained by other traits. Accordingly, our functions for lifespan (*lf*) do not represent the real quantitative thermal responses below the coldest observations, which limits their utility for other applications, such as predicting survival at cold temperatures and lower thermal limits on survival.

Predicted transmission for many of the diseases in this study peaked at and extended to cooler temperatures than for previously studied diseases with more tropical distributions (see *Figure 7* and *Table 2* for 95% credible intervals)(*Mordecai et al., 2019*). Here, the optimal temperatures for transmission varied from 22.7–25.2°C (excluding *Ae. taeniorhynchus* models, *Figure 7*). By contrast, models predict that transmission peaks at 25.4°C for malaria (*Johnson et al., 2015*; *Mordecai et al., 2013*), 26.4°C for Ross River virus (*Shocket et al., 2018*) and dengue in *Ae. albopictus* (*Mordecai et al., 2017*), 28.9°C for Zika in *Ae. aegypti* (*Tesla et al., 2018*), and 29.1°C for dengue in

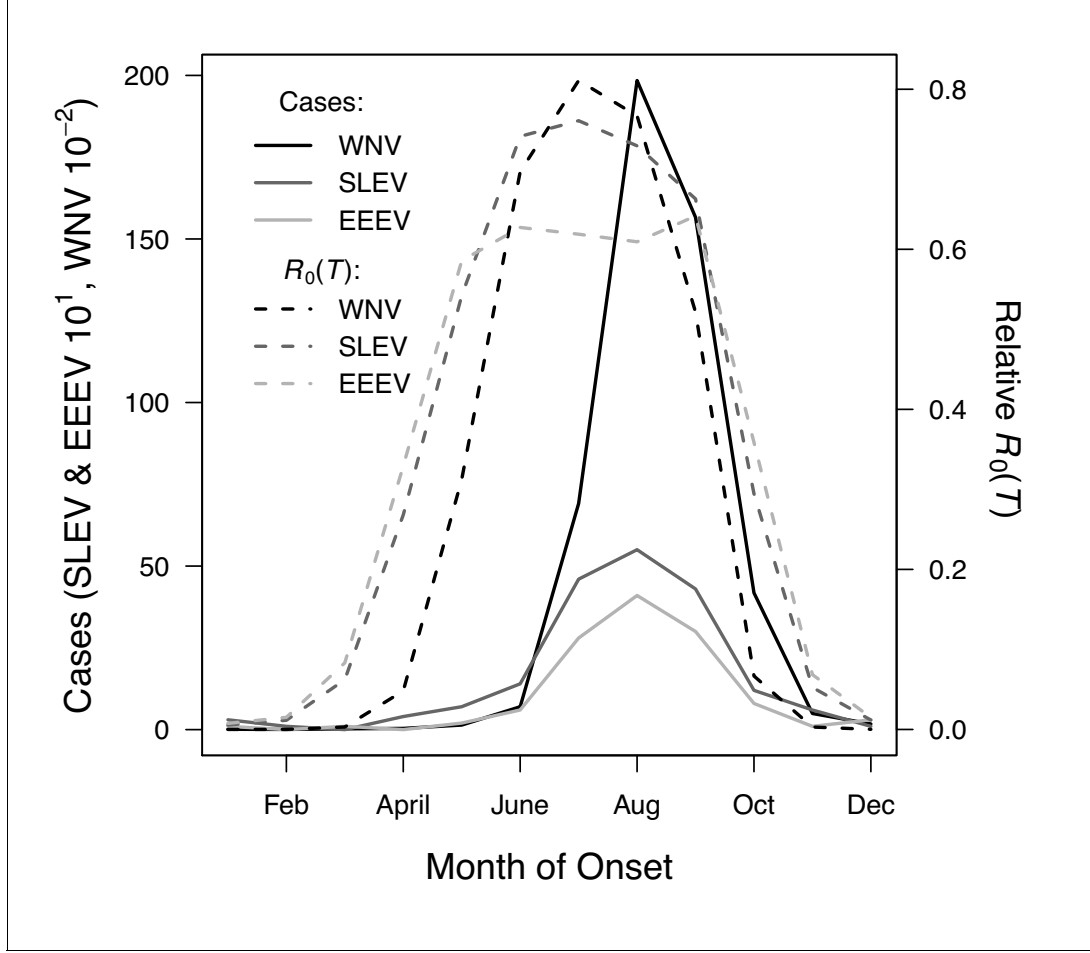

**Figure 9.** $R_0(T)$ predicts the seasonal pattern of human cases of mosquito-borne viral diseases. Incidence (solid lines) lags behind predicted temperature-dependent $R_0$ (dashed lines) for human cases of neuroinvasive disease caused by West Nile virus (WNV, black), St. Louis encephalitis virus (SLEV, dark gray), and Eastern Equine Encephalitis virus (EEEV, light gray) by 2 months. This lag matches patterns in other mosquito-borne diseases and is caused by the time required for mosquito populations to increase, become infectious, and bite humans, and for humans to present symptoms and seek medical care. However, the predicted lag is not present at the end of the transmission season in October, when $R_0(T)$ and incidence decline in tandem.

*Ae. aegypti* (*Mordecai et al., 2017*). Models for several vector–virus pairs also had cooler lower thermal limits (medians: 8.7–19.0°C) than those of diseases with more tropical distributions (medians: 16.0–17.8°C)(*Mordecai et al., 2019*). In combination with similar upper thermal limits (see below), these patterns led to wider thermal breadths (18.2–27.7°C; *Figure 7*) for most of the viruses here compared to the more tropical pathogens (11.7–16.7°C), excepting WNV in *Cx. quinquefasciatus* (12.7°C), the vector most restricted to lower latitude, sub-tropical geographic areas (*Figure 2*). These results match a previous finding that temperate insects had wider thermal breadths than tropical insects (*Deutsch et al., 2008*), and may reflect thermal adaptation to greater variation in temperature in temperate areas compared to tropical areas (*Sunday et al., 2011*). Additionally, SINV—a virus with substantial transmission at very high latitudes in Finland (*Adouchief et al., 2016*)—had the second coolest lower thermal limit (*Figure 7*, *Table 2*). Further, lower latitude *Cx. quinquefasciatus* outperformed higher latitude *Cx. pipiens* at warmer temperatures for proportion ovipositing (*pO*; *Figure 3A*), while the reverse occurred at cooler temperatures for egg viability (*EV*; *Figure 3C*). Collectively, these results imply that, to some extent, measurements of physiological traits can predict geographic patterns of vectors or disease transmission at broad scales. However, geographic range differences (*Figure 2*) did not consistently predict variation in thermal responses among the *Culex* species in this study (e.g. biting rate [*a*, *Figure 3C*] and adult lifespan [*lf*, *Figure 3D*]), indicating that

life history and transmission trait responses at constant temperatures do not always predict the geographic distributions of species. Instead, the ability to tolerate temperature extremes may limit species distributions more than their performance at average or constant temperatures (*Overgaard et al., 2014*). Moreover, although diseases like malaria and dengue are generally considered to be 'tropical', historically their distributions extended further into temperate regions (*Brathwaite Dick et al., 2012*; *Hay et al., 2004*). Thus, current distributions of disease may reflect a realized niche restricted by societal factors more than a fundamental niche based on ecological factors like temperature.

In contrast to the optima, lower thermal limits, and thermal breadths, the upper thermal limits for the vector–virus pairs in this study (31.9–34.9°C, excluding *Ae. taeniorhynchus* models; *Figure 7D*, *Table 2*) closely matched those of more tropical diseases (31.5–34.7°C) (*Johnson et al., 2015*; *Mordecai et al., 2017*; *Mordecai et al., 2013*; *Shocket et al., 2018*; *Tesla et al., 2018*). This similarity may arise because maximum summer temperatures in temperate areas can match or even exceed maximum temperatures in tropical areas (*Sunday et al., 2011*). Accordingly, there may be a fundamental upper thermal constraint on transmission that applies similarly to all mosquito-borne diseases, driven by short mosquito lifespans at high temperatures. The relatively high upper thermal limits in both *Ae. taeniorynchus* transmission models were driven by the thermal response of lifespan (*lf*), which was fit to few data points; more data are needed to determine whether it reflects the true thermal response in that species (*Appendix 1—figure 1*). These results indicate that as temperatures rise due to climate change, temperate diseases are unlikely to be displaced by warming alone, although they may also expand toward the poles, even as tropical diseases may expand farther into temperate zones.

Independent human case data support unimodal thermal responses for transmission and the importance of temperature in shaping geographic patterns of mosquito-borne disease. Human cases of WNV (*Ahmadnejad et al., 2016*; *Hahn et al., 2015*; *Marcantonio et al., 2015*; *Platonov et al., 2008*; *Reisen et al., 2006*; *Semenza et al., 2016*; *Shand et al., 2016*) and SINV (*Brummer-Korvenkontio et al., 2002*; *Jalava et al., 2013*) are often positively associated with temperature. Here, we found that incidence of neuroinvasive WNV disease peaked at intermediate mean summer temperatures (24°C) across counties in the US (*Figure 8*) that matched the optima predicted by our models. This result adds to prior evidence for reduced transmission of WNV (*Mallya et al., 2018*) and other mosquito-borne diseases (*Gatton et al., 2005*; *Mordecai et al., 2013*; *Peña-García et al., 2017*; *Perkins et al., 2015*; *Shah et al., 2019*) at high temperatures. Although we did not detect lower or upper thermal limits for West Nile neuroinvasive disease (*Figure 8*), this result is unsurprising based on fundamental differences between the types of temperature data used to parameterize and validate the models. The $R_0$ model prediction is derived from data collected in a controlled laboratory environment at constant temperatures, while average incidence in the field reflects temperatures that vary at a variety of temporal scales (daily, seasonal, and interannual). Thus, we hypothesize that temperature variation over time may sustain transmission in regions with otherwise unsuitable mean summer temperatures by providing time windows that are suitable for transmission.

The temperature-dependent models also predict the general seasonality of human cases of WNV, EEEV, and SLEV (*Figure 9*). The 2-month lag between climate suitability and the onset of human cases, which matches previous results from other mosquito-borne diseases (*Mordecai et al., 2017*; *Shocket et al., 2018*), arises from the time following the onset of suitable conditions required for mosquito populations to increase (*Stewart Ibarra et al., 2013*), become infectious, and bite humans, and for humans to present symptoms and seek medical care (*Hu et al., 2006*; *Jacups et al., 2008*). Transmission of the more temperate viruses here may incur additional lags because human cases result from enzootic transmission, and multiple rounds of amplification within reservoir hosts may be required before prevalence is sufficiently high to spill over into humans. Additionally, as wild birds begin to migrate in late summer, both *Cx. pipiens* and *Cx. tarsalis* shift their feeding preferences from birds to humans, which should increase transmission to people later in the year (*Kilpatrick et al., 2006*). However, we found that cases decreased more quickly in autumn than expected from temperature effects alone. Human behavior may partially compensate for the shift in feeding preference and explain why the decrease of cases in autumn did not show the expected 2-month lag from temperature-dependent relative $R_0$. For instance, if people wear clothing that exposes less skin and spend less time outdoors due to school schedules and changing daylight it may reduce contact with mosquitoes. Drought, precipitation, and reservoir and human immunity

also strongly drive transmission of WNV (*Ahmadnejad et al., 2016*; *Marcantonio et al., 2015*; *Paull et al., 2017*; *Shand et al., 2016*) and may interact with temperature. SLEV, EEEV, and WEEV are less common in nature, and thus less well-studied, but the lower thermal limits in our study support previous findings that transmission of WEEV is favored over SLEV in cooler conditions (*Hess et al., 1963*). Additionally, the seasonal patterns of incidence data (*Figure 9*) provide some support for the model prediction that SLEV transmission is possible at cooler temperatures than WNV by North American vectors (*Table 2*). By contrast, mean temperature is not associated with outbreaks of RVFV, although they are highly predictable based on precipitation driven by El Niño–Southern Oscillation cycles (*Anyamba et al., 2009*; *Linthicum et al., 1999*). Thus, disease dynamics depend on the interaction between temperature and other environmental factors, and the relative importance of temperature versus other drivers varies across systems.

Most prior studies with mechanistic models for temperature-dependent transmission of WNV do not capture the unimodal thermal response that our mechanistic models predict and that we observe in the human case data (*Table 3*). Two previous models predicted that transmission of WNV would increase up to the warmest temperatures they considered, 28°C (*Vogels et al., 2017*) and 35°C (*Kushmaro et al., 2015*). In both cases, the vector daily survival rates estimated from lab experiments were far less sensitive to temperature than our measure of adult lifespan (*lf*), and neither model was validated with field data. A third study with models for *Cx. pipiens*, *Cx. quinquefasciatus*, and *Cx. tarsalis*, like our study, predicted unimodal thermal responses for transmission, with very similar optima but with lower thermal limits that were ~5°C warmer, resulting in much narrower thermal breadths (*Appendix 1—figure 23*; *Paull et al., 2017*). This previous set of models (*Paull et al., 2017*) was validated with annual, state-level WNV human case data (in contrast to our county-level data averaged over multiple years), and detected a positive effect of temperature, with no decline at high temperatures (*Paull et al., 2017*). The best spatial and temporal scales for validating temperature-dependent transmission models and detecting the impacts of temperature remain an open question. For instance, different approaches may be necessary to detect thermal optima and thermal limits. Critically, differences in modeling and validation approaches can lead to strongly divergent conclusions and predictions for the impact of climate change.

Given the unimodal relationship between temperature and transmission of these temperate mosquito-borne pathogens, we expect climate warming to lead to predictable shifts in disease transmission (*Lafferty, 2009*; *Lafferty and Mordecai, 2016*; *Ryan et al., 2015*). Warming should extend the transmission season earlier into the spring and later into the fall and increase transmission potential in higher latitudes and altitudes, although this prediction may be impacted by changes in bird migrations. However, the thermal optima for these temperate vector–virus pairs are relatively cool, so in many locations, warming could result in summer temperatures that exceed the thermal optima for transmission more frequently, reducing overall transmission or creating a bimodal transmission season (*Molnár et al., 2013*). Based on the average summer temperature data (2001–2016) in our analysis (*Figure 8*), currently the majority of people (70%) and counties (68%) are below the optimal temperature for transmission (23.9°C, fit by the GAM). The numbers are similar when restricted to counties with observed West Nile virus cases: 69% and 70%, respectively. Thus, all else being equal, we might expect a net increase in transmission of West Nile virus in response to the warming climate, even as hot temperatures suppress transmission in some places. Still, warming is unlikely to eliminate any of these more temperate pathogens since the upper thermal limits for transmission are well above temperatures pathogens regularly experience in their current geographic ranges. More generally, our results raise concerns about the common practice of extrapolating monotonic relationships between temperature and disease incidence fit from observational data into warmer climate regimes to predict future cases (*Marcantonio et al., 2015*; *Semenza et al., 2016*).

While the data-driven models presented here represent the most comprehensive synthesis to date of trait thermal response data and their impact on transmission for these mosquito–pathogen systems with substantial transmission in temperate regions, additional temperature-dependent trait data would increase the accuracy and decrease the uncertainty in these models where data were sparse or missing. Our data synthesis and uncertainty analysis suggest prioritizing pathogen development rate (*PDR*) and vector competence (*bc*) data and biting rate (*a*) data because those thermal responses varied widely among vector–virus pairs and determined the lower thermal limits and optima for transmission in many models. Additionally, vector competence (*bc*) and/or pathogen development rate (*PDR*) data were missing in many cases (WNV in *Cx. quinquefasciatus* and *Cx.*

*modestus* [an important vector in Europe], EEEV in *Cs. melanura*, RVFV in vectors from endemic areas, transmission efficiency [*b*] for SINV) or sparse (EEEV and WNV in *Cx. univittatus*), as were biting rate data (*Cx. univittatus*, RVFV vectors). Lifespan (*lf*) data—key for determining transmission optima and upper thermal limits—were the missing for *Ae. triseriatus*, *Cs. melanura*, *Cx. univittatus*, and RVFV vectors, and at temperatures below 14°C for all vector species, so it was unclear which functional form these thermal responses should take (monotonic, saturating, or unimodal). While the other mosquito demographic traits did not determine thermal limits for transmission in models here, fecundity (typically as eggs per female per day, *EFD*), larval-to-adult survival (*pLA*), and egg viability (*EV*) determined thermal limits for malaria (*Mordecai et al., 2013*) and Ross River virus (*Shocket et al., 2018*). Thus, more fecundity data (missing for *Cx. tarsalis*, *Cx. univittatus*, and *Ae. triseriatus*; sparse for *Cx. pipiens* and *Cx. quinquefasciatus*) would also increase our confidence in the models. New data are particularly important for RVFV: the virus has a primarily tropical distribution in Africa and the Middle East, but the model depends on traits measured in *Cx. pipiens* collected from temperate regions and infection traits measured in *Ae. taeniorhynchus*, a North American species. This substitution of a mosquito species that is not a naturally occurring vector could reduce the relevance and utility of this model. RVFV is transmitted by a diverse community of vectors across the African continent, but experiments should prioritize hypothesized primary vectors (e.g. *Ae. circumluteolus* or *Ae. mcintoshi*) or secondary vectors that already have partial trait data (e.g. *Ae. vexans* or *Cx. theileri*) (*Braack et al., 2018*; *Linthicum et al., 2016*). Although temperature itself does not predict the occurrence of RVFV outbreaks, it may affect the size of epidemics once they are triggered by precipitation. More generally, thermal responses may vary across vector populations (*Kilpatrick et al., 2010*) and/or virus isolates even within the same species. Several studies have found differences in thermal performance across different populations of the same mosquito species (*Dodson et al., 2012*; *Mogi, 1992*; *Reisen, 1995*; *Ruybal et al., 2016*) or pathogen strains (*Kilpatrick et al., 2008*), but this variation was not systematically associated with their thermal environments of origin. Accordingly, the potential for thermal adaption in mosquitoes and their pathogens remains an open question. Regardless, more data may improve the accuracy of all the models, even those without missing data.

Our trait-based $R_0$ models effectively isolated the physiological effects of temperature on transmission. However, in nature many other environmental and biological factors also impact transmission of mosquito-borne disease. For example, potential factors include rainfall, habitat and land-use, reservoir host community composition, host immunity, viral and mosquito genotypes, mosquito microbiome, vector control efforts, vector behavior, and human behavior (*Kilpatrick et al., 2010*; *Kilpatrick et al., 2006*; *Paull et al., 2017*; *Shocket et al., 2020*; *Vaidyanathan and Scott, 2007*; *Vazquez-Prokopec et al., 2010*). Our analyses here suggest that temperature is important for shaping broad-scale spatial and seasonal patterns of disease when cases are averaged over time and space. Other factors may be more important at finer spatial and temporal scales, and explain additional variation in human cases. For instance, a study of WNV and two other (non-mosquito-borne) pathogens found that biotic factors were significant drivers of disease distributions at local scales, while climate factors were only significant drivers at larger regional scales (*Cohen et al., 2016*). Given that our $R_0$ models for WNV predicted very similar thermal optima across three distantly-related vector species, it is likely that our results are generalizable to other temperate locations with the same vectors (e.g. parts of Europe with transmission by *Cx. pipiens*) at similarly broad spatial and temporal scales, even if the other factors influencing local-scale patterns are quite different than in the US.

As carbon emissions continue to increase and severe climate change becomes increasingly inevitable (*Intergovernmental Panel on Climate Change, 2014*), it is critical that we understand how temperature change will affect the transmission of mosquito-borne diseases in a warmer future world. While data gaps are still limiting, the mechanistic, trait-based approach presented here is powerful for predicting similarities and differences across vectors and viruses and for making predictions for the impact of climate change (*Mordecai et al., 2019*). Accounting for the effects of temperature variation (*Bernhardt et al., 2018*; *Lambrechts et al., 2011*; *Paaijmans et al., 2010*) is an important next step for using these types of models to accurately predict transmission. In nature, mosquitoes and pathogens experience daily temperature variation that can dramatically alter performance compared to constant temperatures of the same mean (*Lambrechts et al., 2011*; *Paaijmans et al., 2010*). Rate summation is the most common method for predicting performance in

variable temperatures based on experimental data at constant temperatures (*Bernhardt et al., 2018*; *Lambrechts et al., 2011*). This approach is ideal because mean temperature and daily temperature variation vary somewhat independently over space and time, and measuring vector and pathogen performance at sufficient combinations of both is logistically difficult. However, its accuracy for predicting mosquito and pathogen traits or mosquito-borne disease transmission has not been rigorously evaluated. Additionally, the potential for adaptive evolution to warmer climates is uncertain because of limited knowledge on the level of genetic variation in thermal responses for vectors or their pathogens within or between populations. Further, vectors and pathogens may experience different selective pressures, as mosquito populations may depend on either increased fecundity or longevity at high temperatures, while pathogens require longer vector lifespans (*Mordecai et al., 2019*). Thus, future trajectories of these diseases will depend not just on suitability of mean temperatures but also on temperature variation, thermal adaptation of vectors and viruses, land use (which governs mosquito–wildlife–human interactions), vector control activities, human and wildlife immune dynamics, and potential future emergence and spread of new vectors and viruses.

## Materials and methods

All analyses were conducted using R 3.1.3 (*R Development Core Team, 2016*).

### Vector species range maps

The distributions of *Cx. pipiens* and *Cx. quinquefasciatus* are georectified maps adapted from *Farajollahi et al., 2011*; *Smith and Fonseca, 2004*. The northern boundary of *Cx. tarsalis* was taken from *Darsie and Ward, 2016*. For the southern boundary, we drew a convex polygon using five datasets (*Huerta Jiménez, 2018*; *López Cárdenas, 2018*; *Ortega Morales, 2018*; *Ponce García, 2018*; *Walter Reed Biosystematics Unit, 2018*) in the Global Biodiversity Information Facility (https://www.gbif.org/).

### Temperature-dependent trait data

We found 38 studies with appropriate temperature-dependent trait data from controlled laboratory experiments (*Paull et al., 2017*; *Reisen et al., 2006*; *Kilpatrick et al., 2008*; *Ruybal et al., 2016*; *Andreadis et al., 2014*; *Brust, 1967*; *Buth et al., 1990*; *Chamberlain and Sudia, 1955*; *Ciota et al., 2014*; *Cornel et al., 1993*; *Dodson et al., 2012*; *Dohm et al., 2002*; *Kramer et al., 1983*; *Li et al., 2017*; *Loetti et al., 2011*; *Lundström et al., 1990*; *Madder et al., 1983*; *Mahmood and Crans, 1997*; *Mahmood and Crans, 1998*; *McHaffey, 1972a*; *Mogi, 1992*; *Mpho et al., 2001*; *Mpho et al., 2002a*; *Mpho et al., 2002b*; *Nayar, 1972*; *Oda et al., 1980*; *Oda et al., 1999*; *Rayah and Groun, 1983*; *Reisen et al., 1992*; *Reisen et al., 1993*; *Reisen, 1995*; *Rueda et al., 1990*; *Shelton, 1973*; *Tekle, 1960*; *Teng and Apperson, 2000*; *Trpiš and Shemanchuk, 1970*; *Turell et al., 1985*; *Turell and Lundström, 1990*; *van der Linde TC de et al., 1990*). When necessary, we digitized the data using Web Plot Digitizer (*Rohatgi, 2018*), a free online tool. When lifespan data were reported by sex, only female data were used. Vector competence trait data (*b*, *c*, or *bc*) were only included if time at sampling surpassed the estimated extrinsic incubation period (the inverse of *PDR*) at that temperature, which resulted in the exclusion of some studies (*Fros et al., 2015*; *Vogels et al., 2016*).

### Fitting thermal responses

We fit trait thermal responses with a Bayesian approach using the 'r2jags' package (*Su and Yajima, 2009*), an R interface for the popular JAGS program (*Plummer, 2003*) for the analysis of Bayesian graphical models using Gibbs sampling. It is a (near) clone of BUGS (Bayesian inference Using Gibbs Sampling) (*Spiegelhalter et al., 2003*). In JAGS, samples from a target distribution are obtained via Markov Chain Monte Carlo (MCMC). More specifically, JAGS uses a Metropolis-within-Gibbs approach, with an Adaptive Rejection Metropolis sampler used at each Gibbs step (for more information on MCMC algorithms see *Gilks et al., 1998*).

For each thermal response being fit to trait data, we visually identified the most appropriate functional form (quadratic, Briére, or linear; *Equations 3–5*) for that specific trait–species combination (*Mordecai et al., 2019*). For traits with ambiguous functional responses, we fit the quadratic and Briere and used the deviance information criterion (DIC) (*Spiegelhalter et al., 2002*) to pick the best

fit. We assumed normal likelihood distributions with temperature-dependent mean values described by the appropriate function (*Equations 3–5*) and a constant standard deviation (σ) described by an additional fitted parameter ($\tau = 1/\sigma^2$). The 95% credible intervals in *Figures 3–6* estimate the uncertainty in the mean thermal response.

We set all thermal response functions to zero when $T < T_{min}$ and $T > T_{max}$ (for *Equation 3 and 4*) or when $T > -z/m$ (*Equation 5*) to prevent trait values from becoming negative. For traits that were proportions or probabilities, we also limited the thermal response functions at 1. For the linear thermal responses, we calculated the predicted thermal response in a piecewise manner in order to be conservative: for temperatures at or above the coldest observed data point, we used the trait values predicted by the fitted thermal response (i.e. the typical method); for temperatures below the coldest observed data point, we substituted the trait estimate at the coldest observed data point (i.e. forcing the thermal response to plateau, rather than continue increasing beyond the range of observed data).

For the fitting process, we ran three concurrent MCMC chains for 25,000 iterations each, discarding the first 5000 iterations for burn-in (convergence was checked visually). We thinned the resultant chains, saving every eighth step. These settings resulted in 7500 samples in the full posterior distribution that we kept for further analysis.

## Generating priors

We used data-informed priors to decrease the uncertainty in our estimated thermal responses and constrain the fitted thermal responses to be biologically plausible, particularly when data were sparse. These priors used our total dataset, which contained temperature-dependent trait data for all of the main species in the analysis (but with the focal species removed, see below), as well as from additional temperate *Aedes* and *Culex* species (*Buth et al., 1990*; *Ciota et al., 2014*; *Kiarie-Makara et al., 2015*; *Madder et al., 1983*; *McHaffey, 1972b*; *McHaffey and Harwood, 1970*; *Mogi, 1992*; *Muturi et al., 2011*; *Oda et al., 1999*; *Oda et al., 1980*; *Olejnícek and Gelbic, 2000*; *Parker, 1982*).

We fit each thermal response with a sequential two-step process, where both steps employed the same general fitting method (described above in *Fitting Thermal Responses*) but used different priors and data. In step 1, we generated high-information priors by fitting a thermal response to data from all species except the focal species of interest (i.e. a 'leave-one-out' approach). For example, for the prior for biting rate for *Cx. pipiens*, we used the biting rate data for all species except *Cx. pipiens*. For this step, we set general, low-information priors that represented minimal biological constrains on these functions (e.g. typically mosquitoes die if temperatures exceed 45°C, so all biological processes are expected to cease; $T_{min}$ must be less than $T_{max}$). The bounds of these uniformly distributed priors were: $0 < T_{min} < 24$, $26 < T_{max} < 45$ (quadratic) or $28 < T_{max} < 45$ (Briére), $0 < q < 1$, $-10 < m < 10$, and $0 < b < 250$. Then in step 2, we fit a thermal response to data from the focal species using the high-information priors from step 1.

Because we cannot directly pass posterior samples from JAGS as a prior, we modified the results from step 1 to use them in step 2. We used the 'MASS' package (*Venables and Ripley, 2002*) to fit a gamma probability distribution to the posterior distributions for each thermal response parameter ($T_{min}$, $T_{max}$, and $q$ [*Equation 3 and 4*]; or $m$ and $z$ [*Equation 5*]) obtained in step 1. The resulting gamma distribution parameters can be used directly to specify the priors in the JAGS model. Because the prior datasets were often very large, in many cases the priors were too strong and overdetermined the fit to the focal data. In a few other cases, we had philosophical reasons to strongly constrain the fit to the focal data even when they were sparse (e.g. to constrain $T_{max}$ to very high temperatures so that other traits with more information determine the upper thermal limit for $R_0$). Thus, we deflated or inflated the variance as needed (i.e., we fixed the gamma distribution mean but altered the variance by adjusting the parameters that describe the distribution accordingly). See Appendix 1 for more details and specific variance modifications for each thermal response.

## Constructing $R_0$ models

When data were missing for a vector–virus pair, we used two criteria to decide which thermal response to use as a substitute: 1) the ecological similarly (i.e. geographic range overlap) of species with available thermal responses, and 2) how restrictive the upper and lower bounds of the available

thermal responses were. All else being equal, we chose the more conservative (i.e. least restrictive) option so that $R_0$ would be less likely to be determined by trait thermal responses that did not originate from the focal species. See Appendix 1 for more information about specific models.

When there was more than one option for how to parameterize a model (e.g. vector competence data for WEEV in *Cx. tarsalis* were available in two forms: separately as *b* and *c,* and combined as *bc*), we calculated $R_0$ both ways. The results were very similar, except for the model for RVFV with lifespan data from *Cx. pipiens* lifespan in place of *Ae. taeniorhynchus* (*Appendix 1—figure 22*). See Appendix 1 for sensitivity and uncertainty methods and *Appendix 1—figures 11–20* for results.

## Model validation: spatial analysis

We obtained county-level neuroinvasive WNV disease data from 2001 to 2016 for the contiguous US (*n* = 3109) through the CDC's county-level disease monitoring program (*Centers for Disease Control and Prevention, 2018c*). Data were available as total human cases per year, which we adjusted to average cases per 1000 people (using 2010 US county-level census data) to account for population differences. We averaged cases across years beginning with the first year that had reported cases in a given county to account for the initial spread of WNV and the strong impact of immunity on interannual variation (*Paull et al., 2017*). Ninety-eight percent of human cases of WNV in the US occur between June and October (data described below), and cases of mosquito-borne disease often lag behind temperature by 1–2 months (*Shocket et al., 2018*; *Stewart Ibarra et al., 2013*). Thus, we extracted monthly mean temperature data between the months of May–September for all years between 2001 and 2016 and averaged the data to estimate typical summer conditions for each county. Specifically, we took the centroid geographic coordinate for every county in the contiguous US with the 'rgeos' package *Bivand and Rundel, 2012* and extracted corresponding historic climate data for monthly mean temperatures (Climate Research Unit 3.1 rasters) (*Harris et al., 2014*) from $0.5^{\circ 2}$ cells (approximately 2500–3000 km$^2$) using the 'raster' package (*Hijmans, 2020*). The monthly mean temperatures in this climate product are calculated by averaging daily mean temperatures at the station level (based on 4–8 observations per day at regular intervals) and interpolating these over a grid (*World Meteorological Organization, 2009*).

We fit a generalized additive model (GAM) for average incidence as a function of average summer temperature using the 'mgcv' package (*Wood, 2006*). We used a gamma distribution with a log-link function to restrict incidence to positive values and capture heteroskedasticity in the data (i.e. higher variance with higher predicted means), adding a small, near-zero constant (0.0001) to all incidence values to allow the log-transformation for counties with zero incidence. GAMs use additive functions of smooth predictor effects to fit responses that are extremely flexible in the shape of the response. We restricted the number of knots to minimize overfitting (*k* = 7; see *Appendix 1—figure 24* for results across varying values of *k*). For comparison, we also used the 'loess' function in base R 'stats' package (*R Development Core Team, 2016*) to fit locally estimated scatterplot smoothing (LOESS) regressions of the same data. LOESS regression is a simpler but similarly flexible method for estimating the central tendency of data. See *Appendix 1—figure 25* for LOESS model results. See *Appendix 1—figure 26* for non-binned county-level data.

## Model validation: seasonality analysis

We calculated monthly temperature-dependent relative $R_0$ to compare with month-of-onset data for neuroinvasive WNV, EEEV, and SLEV disease aggregated nationwide (the only spatial scale available) from 2001 to 2016 (*Centers for Disease Control and Prevention, 2018c*; *Curren et al., 2018*; *Lindsey et al., 2018*), using the same county-level monthly mean temperature data as above. For WNV, we used the subset of counties with reported cases (68% of counties). For SLEV and EEEV we used all counties from states with reported cases (16 and 20 states, respectively). We calculated a monthly $R_0(T)$ for each county, and then weighted each county $R_0(T)$ by its population size to calculate a national monthly estimate of $R_0(T)$. For WNV, the county-level estimates of $R_0(T)$ used models for three *Culex* species (*Cx. pipiens*, *Cx. quinquefasciatus*, and *Cx. tarsalis*) weighted according to the proportion of WNV-positive mosquitoes reported at the state level, reported in *Paull et al., 2017*. SLEV and EEEV both only had one $R_0$ model. The estimated monthly temperature-dependent relative $R_0$ values and month-of-onset data were compared visually.

## Availability of data and material

All data and code are available on Github in the following repository: https://github.com/mshocket/Six-Viruses-Temp (*Shocket, 2020*; copy archived at https://github.com/elifesciences-publications/Six-Viruses-Temp). All data and code are also available in the Dryad Data Repository.

## Acknowledgements

We gratefully acknowledge the students of the Spring 2017 Stanford University Introductory Seminar course BIO 2N: Global Change and the Ecology and Evolution of Infectious Diseases, who helped with preliminary literature searches, data collection, and model fitting: Uche Amakiri, Michelle Bach, Isabelle Carpenter, Phillip Cathers, Audriana Fitzmorris, Alex Fuentes, Margaux Giles, Gillian Gittler, Emma Leads Armstrong, Erika Malaspina, Elise Most, Stephen Moye, Jackson Rudolph, Simone Speizer, William Wang, and Ethan Wentworth. We thank the Stanford University Introductory Seminars program for support. We thank Michelle Evans for creating *Figure 2*. We thank Marc Fischer, Nicole Lindsey, and Lyle Peterson at the CDC for providing the month-of-onset case data, and Sara Paull for providing state-level data for proportion of WNV vectors. We thank Nicholas Skaff for guidance with EEEV vector ecology, and Eric Pedersen for guidance with the GAM.

## Additional information

### Funding

| Funder | Grant reference number | Author |
| --- | --- | --- |
| National Science Foundation | DEB-1518681 | Marta Shocket<br>Mailo G Numazu<br>Jeremy M Cohen<br>Leah Johnson<br>Erin A Mordecai |
| National Science Foundation | DMS-1750113 | Leah Johnson |
| National Institutes of Health | NIGMS R35 MIRA: 1R35GM133439-01 | Erin A Mordecai |

The funders had no role in study design, data collection and interpretation, or the decision to submit the work for publication.

### Author contributions

Marta S Shocket, Conceptualization, Data curation, Formal analysis, Supervision, Investigation, Visualization, Methodology, Writing - original draft, Writing - review and editing; Anna B Verwillow, Mailo G Numazu, Hani Slamani, Data curation, Investigation; Jeremy M Cohen, Data curation, Formal analysis, Investigation, Visualization, Methodology, Writing - original draft, Writing - review and editing; Fadoua El Moustaid, Data curation, Supervision, Investigation, Writing - review and editing; Jason Rohr, Supervision, Investigation, Writing - review and editing; Leah R Johnson, Conceptualization, Supervision, Investigation, Methodology, Writing - review and editing; Erin A Mordecai, Conceptualization, Resources, Supervision, Funding acquisition, Investigation, Writing - original draft, Project administration, Writing - review and editing

### Author ORCIDs

Marta S Shocket (iD) https://orcid.org/0000-0002-8995-4446

### Decision letter and Author response

Decision letter https://doi.org/10.7554/eLife.58511.sa1
Author response https://doi.org/10.7554/eLife.58511.sa2

## Additional files

### Supplementary files

• Transparent reporting form

### Data availability

Data and code are available on Github (https://github.com/mshocket/Six-Viruses-Temp; copy archived at https://github.com/elifesciences-publications/Six-Viruses-Temp) and in the Dryad Data Repository.

The following dataset was generated:

| Author(s) | Year | Dataset title | Dataset URL | Database and Identifier |
|---|---|---|---|---|
| Shocket MS, Verwillow AB, Numazu MG, Slamani H, Cohen JM, El Moustaid F, Rohr J, Johnson LR, Mordecai EA | 2020 | Transmission of West Nile and five other temperate mosquito-borne viruses peaks at temperatures between 23-26°C | https://doi.org/10.5068/D1VW96 | Dryad, 10.5068/D1VW96 |

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

## Appendix 1

### $R_0$ Model Specifications

The equation for $R_0$ (*Equation 2* in main text) as a function of temperature ($T$) that was used in previous analyses (*Johnson et al., 2015*; *Mordecai et al., 2017*; *Mordecai et al., 2013*; *Parham and Michael, 2010*; *Shocket et al., 2018*; *Tesla et al., 2018*) has fecundity measured as eggs per female per day (*EFD*):

$$\text{Full } R_0: R_0(T) = \left( \frac{a(T)^2 bc(T) e^{-\frac{\mu(T)}{PDR(T)}} EFD(T) EV(T) pLA(T) MDR(T)}{N\, r\, \mu(T)^3} \right)^{1/2} \tag{2}$$

Fecundity data were not available directly as eggs per female per day, so we had to transform the available data to obtain the quantities needed for these models. The data for *Cx. pipiens* were reported as eggs per female per gonotrophic cycle (*EFGC*). To obtain *EFD*, we needed to divide *EFGC* by the length of the gonotrophic cycle. In general, the gonotrophic cycle is assumed to be approximately the inverse of the biting rate. In fact, our 'biting rate' (*a*) data were observations of gonotrophic cycle duration. Accordingly, *EFD* = *EFGC* * *a*, resulting in the following equation for $R_0$:

$$R_0(T) = \left( \frac{a(T)^3 bc(T) e^{-\frac{\mu(T)}{PDR(T)}} EFGC(T) EV(T) pLA(T) MDR(T)}{N\, r\, \mu(T)^3} \right)^{1/2} \tag{A1}$$

All but two of the vector–virus parameterizations used this form (*Equation A1*) of the $R_0$ model (see *Appendix 1—table 1*, exceptions described below).

The fecundity data for *Cx. quinquefasciatus* were reported as eggs per raft (*ER*). Females lay rafts once per gonotrophic cycle. Thus, in order to obtain an approximation to *EFD* (eggs per female per day), we again divide by the number of days per gonotrophic cycle and, further, we multiply by the proportion of females ovipositing (*pO*), since not every female lays an egg raft. These changes result in the following equation for $R_0$:

$$R_0(T) = \left( \frac{a(T)^3 bc(T) e^{-\frac{\mu(T)}{PDR(T)}} ER(T) pO(T) EV(T) pLA(T) MDR(T)}{N\, r\, \mu(T)^3} \right)^{1/2} \tag{A2}$$

The *Cx. quinquefasciatus*–WNV model used *Equation A2*.

The *Ae. triseriatus*–EEEV model also used *Equation A2* (i.e., included *pO*) but substituted the *Cx. pipiens* thermal response for *EFGC* in place of the *Cx. quinquefasciatus* thermal response for *ER* for the following reasons. There were no fecundity trait data available for *Ae.* triseriatus. (*Ae. triseratus* was chosen as the focal species for the EEEV model because it is the only species with temperature-dependent vector competence data available, and it is a possible bridge vector for EEEV transmission to humans). *Cs. melanura* is the primary vector for maintaining enzootic cycles of EEEV in birds (*Mahmood and Crans, 1998*), more often cited in the literature in association with EEEV (e.g. [*Weaver and Barrett, 2004*]), and had data for *pO* (proportion ovipositing) available. Thus, we chose to include this thermal response in model because it contained information that could affect the upper and lower bounds of transmission (even though most models did not include *pO* [proportion ovipositing], because they use the *Cx. pipiens* EFGC [eggs per female per gonotrophic cycle] thermal response that includes *pO* implicitly). Then we needed to choose which egg production metric to include. We chose the *Cx. pipiens* EFGC thermal response over the *Cx. quinquefasciatus* ER thermal response because the former was the better choice according to both criteria: *Cx. pipiens* has a more similar species range to *Ae. triseriatus* and *Cs. melanura* and its thermal response was slightly more conservative (less restrictive = cooler lower thermal limit and warmer upper thermal limit). Although technically the units are not correct (see above), the thermal responses for *Cx. pipiens* EFGC and *Cx. quinquefasciatus* ER are so similar despite having different units (*Figure 4B*), we decided that the other two criteria were more important than being strict with regard to the units, as it is feasible to have an ER thermal response that is quite similar to the EFGC thermal response. Ultimately, because the thermal responses for EFGC and ER are so similar, this decision

only has a small impact on the $R_0$ results (see *Appendix 1—figure 22* comparing four alternative model specifications/parameterizations for the *Ae. triseriatus*-EEEV model).

In *Equations 2, A1, and A2*, the remaining parameters that depend on temperature (*T*) are: adult mosquito mortality (μ, the inverse of lifespan [*lf*]), pathogen development rate (*PDR*, the inverse of the extrinsic incubation period: the time required for exposed mosquitoes to become infectious), egg viability (proportion of eggs hatching into larvae, *EV*), proportion of larvae surviving to adulthood (*pLA*), and mosquito development rate (*MDR*, the inverse of the development period), and vector competence (*bc*, the proportion of exposed mosquitoes that become infectious). Vector competence is the product of infection efficiency (*c*, the proportion of exposed mosquitoes that develop a disseminated infection) and transmission efficiency (*b*, the proportion of infected mosquitoes that become infectious, with virus present in saliva). The form of vector competence varied between models based on the availability of data: *bc*(*T*) [reported a single parameter], *c*(*T*)*b*(*T*) [both parameters reported separately], *c*(*T*) only, or *b*(*T*) only (see *Appendix 1—table 1*). The two remaining parameters do not depend on temperature: human density (*N*) and the rate at which infected hosts recover and become immune (*r*).

**Appendix 1—table 1.** Trait thermal responses used in transmission ($R_0$) models.

Viruses: West Nile (WNV), Eastern and Western Equine Encephalitis (EEEV and WEEV), St. Louis Encephalitis (SLEV), Sindbis (SINV), and Rift Valley Fever (RVFV). *Ae. vex.* = *Ae. vexans*, *Cs. mel.* = *Culiseta melanura*; all other vectors (*Cx.* = *Culex*) listed under model names. Traits are: fecundity (as eggs/female/gonotrophic cycle [*EFGC*] or eggs per raft*proportion ovipositing [*ER*pO*]), egg viability (*EV*), larval-to-adult survival (*pLA*), mosquito development rate (*MDR*), lifespan (*lf*), biting rate (*a*), vector competence (*bc*, *b* [transmission efficiency], *c* [infection efficiency], or *b*c*, as available), and parasite development rate (*PDR*). The WNV–*Cx. quinquefasciatus* model uses *Equation A2* (*ER*pO*); the EEEV–*Ae. triseriatus* model uses *EFGC* from *Cx. pipiens* and *pO* from *Cs. melanura*; all other models use *Equation A1* (*EFGC*). When data were missing for a vector–virus pair, we substituted the most conservative (i.e. least restrictive of transmission) trait thermal response from a vector that occurs within the geographic range of disease transmission. Several models had multiple potentially valid choices for traits; we explain and show compare these alternative models with the main text versions in *Appendix 1—figure 22*. Checkmarks indicate a thermal response from the vector in the model name. The parasite development rate data for SINV was insensitive to temperature (*Figure 4*), so the trait thermal response was omitted from the SINV models ('NA').

| Model: virus–vector | EFGC or ER*pO | EV | pLA | MDR | lf | a | bc, c*b c, or b | PDR |
|---|---|---|---|---|---|---|---|---|
| WNV–*Cx. pipiens* | ✓ | ✓ | ✓ | ✓ | ✓ | ✓ | ✓ (bc) | ✓ |
| WNV–*Cx. quinquefasciatus* | ✓ | | ✓ | ✓ | ✓ | ✓ | ✓ | Cx. uni. (bc) | ✓ |
| WNV–*Cx. tarsalis* | Cx. pip. | Cx. pip | ✓ | ✓ | ✓ | ✓ | ✓ (b) | ✓ |
| WNV–*Cx. univittatus* | Cx. pip. | Cx. pip. | Cx. pip. | Cx. pip. | Cx. pip. | Cx. pip. | ✓ (bc) | ✓ |
| WEEV–*Cx. tarsalis* | Cx. pip. | Cx. pip. | ✓ | ✓ | ✓ | ✓ | ✓ (c*b) | ✓ |
| SLEV–*Cx. tarsalis* | Cx. pip. | Cx. pip. | ✓ | ✓ | ✓ | ✓ | ✓ (c*b) | ✓ |
| EEEV–*Ae. triseriatus* | Cx. pip., Cs. mel. | Cx. pip. | ✓ | | ✓ | Cx. pip. | Cs. mel. | ✓ (bc) | ✓ |
| SINV–*Cx. pipiens* | ✓ | ✓ | ✓ | ✓ | ✓ | ✓ | ✓ (c) | NA |
| SINV–*Ae. taeniorhynchus* | Cx. pip. | Ae. vex. | Ae. vex. | Ae. vex. | ✓ | Cx. pip. | ✓ (c) | NA |
| RVFV–*Ae. taeniorhynchus* | Cx. pip. | Cx. the. | Ae. vex. | Ae. vex. | ✓ | Cx. pip. | ✓ (bc) | ✓ |

**Appendix 1—table 2.** Trait thermal response functions, data sources, and posterior estimates: biting rate and fecundity traits.

Asymmetrical responses fit with Brière function (**B**): $B(T) = qT(T - T_{min})(T_{max} - T)^{1/2}$; symmetrical responses fit with quadratic function (**Q**): $Q(T) = -q(T - T_{min})(T - T_{max})$. Median function coefficients and optima (with 95% credible intervals).

| Trait/Species data source] | F (x) | q (CIs) | $T_{min}$ (CIs) | $T_{max}$ (CIs) | $T_{opt}$ (CIs) |
|---|---|---|---|---|---|
| **Biting rate (a)** | | | | | |
| Cx. pipiens (*Li et al., 2017*; *Madder et al., 1983*; *Ruybal et al., 2016*; *Tekle, 1960*) | B | $1.70 \cdot 10^{-4}$ (1.18–2.29·10⁻⁴) | 9.4 (2.8–13.4) | 39.6 (37.9–40.6) | 32.7 (31.3–33.6) |
| Cx. quinquefasciatus (*Reisen et al., 1992*; *Tekle, 1960*) | B | $7.28 \cdot 10^{-5}$ (5.31–11.8·10⁻⁵) | 3.1 (0.1–10.9) | 39.3 (38.0–40.8) | 31.9 (30.6–33.3) |
| Cx. tarsalis (*Reisen et al., 1992*) | B | $1.67 \cdot 10^{-4}$ (0.87–2.56·10⁻⁴) | 2.3 (0.1–9.4) | 32.0 (30.6–41.7) | 25.9 (24.8–33.9) |
| Cs. melanura (*Mahmood and Crans, 1997*) | B | $1.87 \cdot 10^{-4}$ (1.49–2.31·10⁻⁴) | 7.8 (5.5–11.4) | 31.8 (31.0–33.4) | 26.4 (25.7–27.9) |
| **Fecundity** | | | | | |
| Cx. pipiens (EFGC) (*Li et al., 2017*) | Q | $5.98 \cdot 10^{-1}$ (4.31–7.91·10⁻¹) | 5.3 (2.6–8.5) | 38.9 (36.2–41.8) | 22.1 (20.1–24.4) |
| Cx. quinquefasciatus (ER) (*Mogi, 1992*; *Oda et al., 1980*) | Q | $6.36 \cdot 10^{-1}$ (4.50–9.05·10⁻¹) | 5.0 (1.3–9.8) | 37.7 (34.8–40.7) | 21.4 (18.9–24.4) |
| **Proportion ovipositing (pO)** | | | | | |
| Cx. pipiens (*Ciota et al., 2014*; *Tekle, 1960*) | Q | $4.45 \cdot 10^{-3}$ (2.54–7.77·10⁻³) | 8.2 (4.6–12.1) | 33.2 (30.1–37.5) | 20.8 (18.6–23.4) |
| Cx. quinquefasciatus (*Ciota et al., 2014*; *Oda et al., 1980*; *Tekle, 1960*) | B | $6.67 \cdot 10^{-4}$ (5.80–7.91·10⁻⁴) | 1.7 (0.2–4.8) | 31.8 (31.1–32.2) | 24.9 (21.8–26.0) |
| Cs. melanura (*Mahmood and Crans, 1997*) | Q | $6.31 \cdot 10^{-3}$ (4.52–7.89·10⁻³) | 8.7 (6.9–10.4) | 33.6 (32.5–35.4) | 20.7 (16.9–22.3) |
| **Egg viability (EV)** | | | | | |
| Ae. vexans (*McHaffey, 1972a*) | | $1.24 \cdot 10^{-3}$ (0.73–1.95·10⁻³) | 0 (0–1.6) | 55.5 (45.9–74.1) | 27.6 (20.4–34.0) |
| Cx. pipiens (*Li et al., 2017*) | Q | $2.11 \cdot 10^{-3}$ (1.36–3.05·10⁻³) | 3.2 (0.5–7.1) | 42.6 (39.7–48.3) | 23.0 (20.7–26.3) |
| Cx. quinquefasciatus (*Oda et al., 1980*; *Rayah and Groun, 1983*) | B | $0.47 \cdot 10^{-3}$ (0.34–0.62·10⁻³) | 13.6 (9.3–16.8) | 38.0 (37.2–38.7) | 32.1 (31.3–32.7) |
| Cx. theileri (*van der Linde TC de et al., 1990*) | Q | $2.54 \cdot 10^{-3}$ (1.86–3.41·10⁻³) | 5.5 (2.6–8) | 45.4 (42.4–49.0) | 23.6 (18.2–27.0) |

Additional data sources for other species used for fitting priors only (priors were fit using all data except that of the focal species). Fecundity (ER): *Cx. pipiens molestus* (*Oda et al., 1980*), *Cx. pipiens pallens* (*Mogi, 1992*), and *Ae. dorsalis* (*Parker, 1982*). Proportion ovipositing (pO): *Cx. pipiens molestus* (*Oda et al., 1980*) and *Ae. dorsalis* (*Parker, 1982*). Egg viability (EV): *Cx. pipiens molestus* (*Oda et al., 1980*), *Aedes dorsalis* (*McHaffey and Harwood, 1970*), and *Ae. nigromaculis* (*McHaffey, 1972b*). See Appendix 1 section: *Priors for trait thermal responses*.

**Appendix 1—table 3.** Trait thermal response functions, data sources, and posterior estimates: larval traits.

Asymmetrical responses fit with Brière function (**B**): $B(T) = qT(T - T_{min})(T_{max} - T)^{1/2}$; symmetrical responses fit with quadratic function (**Q**): $Q(T) = -q(T - T_{min})(T - T_{max})$. Median function coefficients and optima (with 95% credible intervals).

| Trait/Species (data source) | F (x) | q (CIs) | $T_{min}$ (CIs) | $T_{max}$ (CIs) | $T_{opt}$ (CIs) |
|---|---|---|---|---|---|
| Mosquito Dev. Rate (MDR) | | | | | |
| Ae. triseriatus (Shelton, 1973) | B | $4.30 \cdot 10^{-5}$ $(3.01–$ $5.83 \cdot 10^{-5})$ | 0.8 $(0–$ 7.5) | 36.5 $(34.6–$ 39.5) | 29.3 $(27.8–$ 31.9) |
| Ae. vexans (Brust, 1967; Trpiš and Shemanchuk, 1970) | B | $4.33 \cdot 10^{-5}$ $(3.34–$ $5.50 \cdot 10^{-5})$ | 1.9 $(0.1–$ 10.5) | 38.2 $(37.0–$ 39.5) | 30.9 $(29.8–$ 32.2) |
| Cx. pipiens (Ciota et al., 2014; Loetti et al., 2011; Madder et al., 1983; Mpho et al., 2002a; Mpho et al., 2002b Ruybal et al., 2016; Tekle, 1960) | B | $3.76 \cdot 10^{-5}$ $(3.36–$ $4.47 \cdot 10^{-5})$ | 0.1 $(0–$ 4.0) | 38.5 $(37.6–$ 39.8) | 30.9 $(30.2–$ 31.9) |
| Cx. quinquefasciatus (Ciota et al., 2014; Mpho et al., 2001; Rueda et al., 1990; Shelton, 1973; Tekle, 1960) | B | $4.14 \cdot 10^{-5}$ $(3.46–$ $5.26 \cdot 10^{-5})$ | 0.1 $(0–$ 5.5) | 38.6 $(37.4–$ 40.6) | 31.0 $(30.0–$ 32.6) |
| Cx. tarsalis (Buth et al., 1990; Dodson et al., 2012; Reisen, 1995) | B | $4.12 \cdot 10^{-5}$ $(3.15–$ $5.47 \cdot 10^{-5})$ | 4.3 $(0–$ 8.4) | 39.9 $(37.9–$ 42.2) | 32.3 $(31.0–$ 34.0) |
| Cs. melanura (Mahmood and Crans, 1998) | B | $2.74 \cdot 10^{-5}$ $(1.64–$ $4.72 \cdot 10^{-5})$ | 8.6 $(0–$ 16.8) | 37.6 $(35.1–$ 40.4) | 31.1 $(28.7–$ 33.7) |
| Larval survival ($p_{LA}$) | | | | | |
| Ae. triseriatus (Shelton, 1973; Teng and Apperson, 2000) | Q | $3.26 \cdot 10^{-3}$ $(1.95–$ $5.18 \cdot 10^{-3})$ | 8.3 $(4.9–$ 11.4) | 35.7 $(32.9–$ 39.7) | 22.0 $(19.9–$ 24.6) |
| Ae. vexans (Brust, 1967; Trpiš and Shemanchuk, 1970) | Q | $3.29 \cdot 10^{-3}$ $(2.65–$ $4.24 \cdot 10^{-3})$ | 9.1 $(8.1–$ 10.6) | 40.8 $(38.4–$ 43.6) | 25.0 $(23.9–$ 26.2) |
| Cx. pipiens (Ciota et al., 2014; Loetti et al., 2011; Madder et al., 1983; Mpho et al., 2002a; Mpho et al., 2002b; Ruybal et al., 2016; Tekle, 1960) | Q | $3.60 \cdot 10^{-3}$ $(2.96–$ $4.42 \cdot 10^{-3})$ | 7.8 $(6.1–$ 9.3) | 38.4 $(37.1–$ 39.9) | 23.1 $(22.2–$ 24.0) |
| Cx. quinquefasciatus (Ciota et al., 2014; Mogi, 1992; Mpho et al., 2001; Oda et al., 1999; Rueda et al., 1990; Shelton, 1973; Tekle, 1960) | Q | $4.26 \cdot 10^{-3}$ $(3.51–$ $5.17 \cdot 10^{-3})$ | 8.9 $(7.6–$ 9.9) | 37.7 $(36.2–$ 39.2) | 23.3 $(22.5–$ 24.0) |
| Cx. tarsalis (Buth et al., 1990; Dodson et al., 2012; Reisen, 1995) | Q | $2.12 \cdot 10^{-3}$ $(1.52–$ $3.08 \cdot 10^{-3})$ | 5.9 $(3.0–$ 8.8) | 43.1 $(39.8–$ 47.5) | 24.6 $(22.9–$ 26.4) |
| Cs. melanura (Mahmood and Crans, 1998) | Q | $3.03 \cdot 10^{-3}$ $(1.55–$ $5.68 \cdot 10^{-3})$ | 10.1 $(5.7–$ 15.1) | 36.2 $(32.8–$ 40.7) | 23.2 $(20.4–$ 26.5) |

Additional data sources for other species used for fitting priors only (priors were fit using all data except that of the focal species). Mosquito Development Rate (MDR): Cx. pipiens molestus (Kiarie-Makara et al., 2015; Olejnícek and Gelbic, 2000), Cx. pipiens pallens (Kiarie-Makara et al., 2015), Cx. restuans (Buth et al., 1990; Madder et al., 1983; Muturi et al., 2011; Shelton, 1973), Cx. salinarius (Shelton, 1973), Ae. solicitans (Shelton, 1973), and Ae. nigromaculis (Brust, 1967). Larval survival ($p_{LA}$): Cx. pipiens molestus (Oda et al., 1999; Olejnícek and Gelbic, 2000), Cx. pipiens pallens (Mogi, 1992), Cx. restuans (Buth et al., 1990; Ciota et al., 2014; Madder et al., 1983; Muturi et al., 2011; Shelton, 1973), Cx. salinarius (Shelton, 1973), Ae. sollicitans (Shelton, 1973), Ae. nigromaculis (Brust, 1967). See Appendix 1 section: Priors for trait thermal responses.

**Appendix 1—table 4.** Trait thermal response functions, data sources, and posterior estimates: vector competence traits.

Asymmetrical responses fit with Brière function (B): $B(T) = qT(T - T_{min})(T_{max} - T)^{1/2}$; symmetrical responses fit with quadratic function (Q): $Q(T) = -q(T - T_{min})(T - T_{max})$. Median function coefficients and optima (with 95% credible intervals).

| Trait/Species (data source) | F(x) | q (CIs) | $T_{min}$ (CIs) | $T_{max}$ (CIs) | $T_{opt}$ (CIs) |
|---|---|---|---|---|---|
| **Transmission efficiency (b)** | | | | | |
| SLEV \| Cx. tarsalis (**Reisen et al., 1993**) | Q | $2.98 \cdot 10^{-3}$ ($1.63–5.31 \cdot 10^{-3}$) | 10.8 (6.2–14.2) | 41.6 (36.8–49.1) | 26.2 (23.5–29.7) |
| WEEV \| Cx. tarsalis (**Reisen et al., 1993**) | Q | $3.17 \cdot 10^{-3}$ ($1.65–5.06 \cdot 10^{-3}$) | 8.2 (5.1–10.7) | 33.5 (31.0–38.9) | 20.9 (19.2–23.2) |
| WNV \| Cx. tarsalis (**Reisen et al., 2006**) | Q | $2.94 \cdot 10^{-3}$ ($1.91–4.48 \cdot 10^{-3}$) | 11.3 (7.6–14.0) | 41.9 (37.7–47.0) | 26.6 (23.9–29.3) |
| **Infection efficiency (c)** | | | | | |
| SINV \| Ae. taeniorhynchus (**Turell and Lundström, 1990**) | Q | $1.24 \cdot 10^{-3}$ ($0.75–2.17 \cdot 10^{-3}$) | 1.4 (0–9.1) | 48.4 (40.8–57.1) | 25.4 (21.0–31.1) |
| SINV \| Cx. pipiens (**Lundström et al., 1990**) | Q | $1.33 \cdot 10^{-3}$ ($0.47–2.30 \cdot 10^{-3}$) | 0 (0–0) | 35.0 (28.1–61.1) | 17.5 (14.1–30.5) |
| WNV \| Cx. pipiens (**Dohm et al., 2002**; **Kilpatrick et al., 2008**) | Q | $2.56 \cdot 10^{-3}$ ($2.05–3.19 \cdot 10^{-3}$) | 15.6 (14.3–16.6) | 52.2 (48.4–56.6) | 33.9 (31.9–36.1) |
| SLEV \| Cx. tarsalis (**Reisen et al., 1993**) | Q | $2.03 \cdot 10^{-3}$ ($1.28–3.07 \cdot 10^{-3}$) | 8.8 (6.6–10.6) | 43.7 (38.9–51.4) | 26.2 (24.2–29.7) |
| WEEV \| Cx. tarsalis (**Kramer et al., 1983**; **Reisen et al., 1993**) | Q | $3.04 \cdot 10^{-3}$ ($2.52–3.68 \cdot 10^{-3}$) | 1.3 (0.4–2.9) | 38.8 (36.7–41.5) | 15.5 (13.4–19.7) |
| **Vector competence (bc)** | | | | | |
| RVFV \| Ae. taeniorhynchus (**Turell et al., 1985**) | Q | $1.51 \cdot 10^{-3}$ ($1.03–2.05 \cdot 10^{-3}$) | 7.1 (2.8–9.8) | 42.3 (39.3–46.5) | 24.7 (22.0–27.0) |
| EEEV \| Ae. triseriatus (**Chamberlain and Sudia, 1955**) | Q | $1.51 \cdot 10^{-3}$ ($0.96–2.24 \cdot 10^{-3}$) | 7.0 (2.9–11.9) | 50.3 (42.3–63.1) | 28.8 (23.6–35.8) |
| WNV \| Cx. pipiens (**Kilpatrick et al., 2008**) | Q | $3.05 \cdot 10^{-3}$ ($1.68–4.87 \cdot 10^{-3}$) | 16.8 (15–17.9) | 38.9 (36.1–44.1) | 27.8 (26.6–30.1) |
| WEEV \| Cx. tarsalis (**Kramer et al., 1983**) | Q | $1.17 \cdot 10^{-3}$ ($0.55–2.36 \cdot 10^{-3}$) | 5.1 (0.6–13.3) | 37.0 (33.5–46.0) | 21.4 (18.1–27.3) |
| WNV \| Cx. univittatus (**Cornel et al., 1993**) | Q | $2.32 \cdot 10^{-3}$ ($1.58–3.68 \cdot 10^{-3}$) | 4.2 (1.5–7.1) | 45.2 (39.6–53.0) | 23.7 (19.4–27.3) |

**Appendix 1—table 5.** Trait thermal response functions, data sources, and posterior estimates: parasite development rate.
Asymmetrical responses fit with Brière function (**B**): $B(T) = qT(T - T_{min})(T_{max} - T)^{1/2}$; symmetrical responses fit with quadratic function (**Q**): $Q(T) = -q(T - T_{min})(T - T_{max})$. Median function coefficients and optima (with 95% credible intervals).

| Trait/Species (data source) | F(x) | q (CIs) | $T_{min}$ (CIs) | $T_{max}$ (CIs) | $T_{opt}$ (CIs) |
|---|---|---|---|---|---|
| **Parasite Dev. Rate (PDR)** | | | | | |
| RVFV \| Ae. taeniorhynchus (**Turell et al., 1985**) | B | $8.84 \cdot 10^{-5}$ ($2.51–15.5 \cdot 10^{-5}$) | 9.0 (5.4–13.8) | 45.9 (41.9–50.3) | 37.8 (34.5–41.3) |
| EEEV \| Ae. triseriatus (**Chamberlain and Sudia, 1955**) | B | $7.05 \cdot 10^{-5}$ ($5.21–9.68 \cdot 10^{-5}$) | 11.6 (7.0–16.4) | 44.8 (40.6–49.4) | 37.2 (33.8–41.1) |
| WNV \| Cx. pipiens (**Dohm et al., 2002**; **Kilpatrick et al., 2008**) | B | $7.38 \cdot 10^{-5}$ ($5.38–9.94 \cdot 10^{-5}$) | 11.4 (7.3–15.0) | 45.2 (40.7–50.3) | 37.5 (33.8–41.6) |
| WNV \| Cx. quinquefasciatus (**Paull et al., 2017**) | B | $7.12 \cdot 10^{-5}$ ($4.58–10.2 \cdot 10^{-5}$) | 19.0 (12.9–21.0) | 44.1 (38.8–50.4) | 37.7 (33.6–42.7) |
| SLEV \| Cx. tarsalis (**Reisen et al., 1993**) | B | $7.11 \cdot 10^{-5}$ ($5.60–8.95 \cdot 10^{-5}$) | 12.8 (10.3–14.3) | 45.2 (40.2–51.5) | 37.7 (33.8–42.6) |
| WEEV \| Cx. tarsalis (**Kramer et al., 1983**; **Reisen et al., 1993**) | B | $6.43 \cdot 10^{-5}$ ($4.44–10.4 \cdot 10^{-5}$) | 4.0 (0–12.6) | 44. 0 (38.3–50.9) | 35.7 (31.0–41.4) |

*Continued on next page*

*Appendix 1—table 5 continued*

| Trait/Species (data source) | F(x) | q (CIs) | $T_{min}$ (CIs) | $T_{max}$ (CIs) | $T_{opt}$ (CIs) |
|---|---|---|---|---|---|
| WNV \| *Cx. tarsalis* (**Reisen et al., 2006**) | B | $6.57 \cdot 10^{-5}$ $(5.11–8.85 \cdot 10^{-5})$ | 11.2 (7.9–14.9) | 44.7 (40.4–49.4) | 37.0 (33.6–40.9) |
| WNV \| *Cx. univittatus* (**Cornel et al., 1993**) | B | $7.54 \cdot 10^{-5}$ $(4.13–11.1 \cdot 10^{-5})$ | 10.2 (7.1–15.3) | 34.4 (31.2–51.1) | 28.8 (26.1–42.5) |
| SINV \| *Ae. taeniorhynchus* (**Turell and Lundström, 1990**) | NA | Not fitted because lack of temperature sensitivity | | | |

**Appendix 1—table 6.** Trait thermal response functions, data sources, and posterior estimates: lifespan.

Responses fit with a linear function (**L**): L(*T*) = -*mT* + *z*. Median function coefficients and $T_{max}$ (with 95% credible intervals).

| Trait/Species (data source) | F(x) | m | z | Tmax = z/m |
|---|---|---|---|---|
| Lifespan (*lf*) | | | | |
| *Ae. taeniorhynchus* (**Nayar, 1972**) | L | 2.02 (1.59–3.19) | 85.9 (73.8–117.6) | 42.7 (34.5–48.5) |
| *Cx. pipiens* (**Andreadis et al., 2014**; **Ciota et al., 2014**; **Ruybal et al., 2016**) | L | 4.86 (3.83–5.84) | 169.8 (142.1–195.6) | 34.9 (32.9–37.9) |
| *Cx. quinquefasciatus* (**Ciota et al., 2014**; **Oda et al., 1999**) | L | 3.80 (1.85–5.29) | 136.3 (86.8–174.0) | 35.9 (32.1–48.5) |
| *Cx. tarsalis* (**Reisen, 1995**) | L | 1.69 (1.12–2.24) | 69.6 (55.8–83.5) | 41.3 (36.6–50.8) |

Additional data sources for other species used for fitting priors only (priors were fit using all data except that of the focal species). Lifespan (*lf*): *Cx. pipiens molestus* (**Kiarie-Makara et al., 2015**; **Oda et al., 1999**), *Cx. pipiens pallens* (**Kiarie-Makara et al., 2015**), and *Cx. restuans* (**Ciota et al., 2014**). See Appendix 1 section: *Priors for trait thermal responses*.

**Appendix 1—table 7.** Priors for trait thermal response functions: mosquito traits with unimodal responses.

Gamma distribution parameters (α [shape] and β [rate]) for priors for fitting thermal response parameters ($T_{min}$, $T_{max}$, and q). Scaled variances are noted in parentheses, either by the system name (applied to all parameters) or by individual parameters. See Appendix 1 section: *Priors for trait thermal responses*.

| Trait/System | q: $\alpha$ | q: $\beta$ | $T_{min}$: $\alpha$ | $T_{min}$: $\beta$ | $T_{max}$: $\alpha$ | $T_{max}$: $\beta$ |
|---|---|---|---|---|---|---|
| Biting rate (*a*) | | | | | | |
| *Cx. pipiens* (0.5) | 8.84 | 64200 | 1.91 | 0.367 | 103 | 3.00 |
| *Cx. quinquefasciatus* | 39.1 (0.1) | 234133 (0.1) | 8.82 (0.1) | 0.997 (0.1) | 2992 | 75.8 |
| *Cx. tarsalis* | 40.1 (0.05) | 227752 (0.05) | 18.7 (0.05) | 1.745 (0.05) | unif. | unif. |
| *Cs. melanura* | 35.4 (0.75) | 229694 (0.75) | 7.77 (0.75) | 0.895 (0.75) | 2714 (0.1) | 68.5 (0.1) |
| Fecundity | | | | | | |
| *Cx. pipiens* (*EFGC*) (3) | 9.23 | 15.6 | 2.38 | 0.419 | 139 | 3.52 |
| *Cx. quinquefasciatus* (*ER*) | 19.1 | 30.44 | 2.87 | 0.600 | 486 | 13.2 |
| Prop. ovipositing (*pO*) | | | | | | |

*Continued on next page*

*Appendix 1—table 7 continued*

| Trait/System | q: $\alpha$ | q: $\beta$ | $T_{min}$: $\alpha$ | $T_{min}$: $\beta$ | $T_{max}$: $\alpha$ | $T_{max}$: $\beta$ |
|---|---|---|---|---|---|---|
| *Cx. pipiens* (0.5) | 9.50 | 1823 | 14.8 | 1.495 | 263 | 7.14 |
| *Cx. quinquefasciatus* | 32.9 | 55242 | 1.41 | 0.397 | 3346 | 106 |
| *Cs. melanura* | 14.4 | 2635 | 22.0 | 2.254 | 588 | 16.8 |
| Egg viability (*EV*) | | | | | | |
| *Ae. vexans* (0.01) | 26.6 | 12259 | 11.6 | 1.916 | 486 | 10.8 |
| *Cx. pipiens* (0.2) | 29.4 | 14525 | 8.83 | 1.579 | 514 | 11.1 |
| *Cx. quinquefasciatus* (0.1) | 101 | 262268 | 1.08 | 1.032 | 1361 | 34.9 |
| *Cx. theileri* | 5.86 | 2266 | 4.46 | 0.591 | 266 | 6.06 |
| Mos. dev. rate (*MDR*) | | | | | | |
| *Ae. triseriatus* (0.2) | 118 | 2697528 | 1.93 | 0.703 | 5542 | 145 |
| *Ae. vexans* (0.5) | 119 | 2739401 | 1.89 | 0.689 | 6661 | 174 |
| *Cx. pipiens* (0.1) | 71.9 | 1545915 | 2.03 | 0.596 | 2912 | 76.5 |
| *Cx. quinquefasciatus* (0.1) | 113 | 2569782 | 1.81 | 0.651 | 5900 | 155 |
| *Cx. tarsalis* (0.1) | 129 | 2940582 | 1.49 | 0.660 | 6431 | 169 |
| *Cs. melanura* (0.1) | 129 | 2941063 | 1.78 | 0.685 | 6915 | 181 |
| Larval survival ($p_{LA}$) | | | | | | |
| *Ae. triseriatus* (0.05) | 163 | 46723 | 231 | 27.3 | 4667 | 122 |
| *Ae. vexans* (0.05) | 135 | 37701 | 210 | 24.6 | 4040 | 107 |
| *Cx. pipiens* (0.1) | 102 | 27382 | 217 | 24.2 | 2872 | 76.7 |
| *Cx. quinquefasciatus* (0.1) | 88.8 | 26461 | 123 | 14.8 | 2608 | 68.4 |
| *Cx. tarsalis* (0.025) | 94.6 | 23240 | 237 | 26.2 | 2564 | 69.5 |
| *Cs. melanura* (0.05) | 148.9 | 41533 | 239 | 27.8 | 4391 | 116 |

**Appendix 1—table 8.** Priors for trait thermal response functions: infection traits.
Gamma distribution parameters ($\alpha$ [shape] and $\beta$ [rate]) for priors for fitting thermal response parameters ($T_{min}$, $T_{max}$, and $q$). Scaled variances are noted in parentheses, either by the system name (applied to all parameters) or by individual parameters. See Appendix 1 section: *Priors for trait thermal responses.*

| Trait/System | q: $\alpha$ | q: $\beta$ | $T_{min}$: $\alpha$ | $T_{min}$: $\beta$ | $T_{max}$: $\alpha$ | $T_{max}$: $\beta$ |
|---|---|---|---|---|---|---|
| Transmission efficiency (*b*) | 7.72 | 3202 | 9.97 | 1.268 | 114 | 2.9 |
| SLEV | *Cx. tarsalis* (0.5) | 9.49 | 2373 | 79.6 | 6.181 | 153 | 3.74 |
| WEEV | *Cx. tarsalis* (0.1) | 8.46 | 3056 | 12.1 | 1.455 | 134 | 3.5 |
| WNV | *Cx. tarsalis* | 7.72 | 3202 | 9.97 | 1.268 | 114 | 2.9 |
| Infection efficiency (*c*) | | | | | | |
| SINV | *Ae. taeniorhynchus* (0.1) | 61.7 | 45102 | 2.49 | 0.815 | 1214 | 25.1 |
| SINV | *Cx. pipiens* (0.01) | 57.3 | 40236 | 2.64 | 0.799 | 1124 | 23.28 |
| WNV | *Cx. pipiens* | 28.5 | 15944 | 1.44 | 0.852 | 237 | 5.393 |
| SLEV | *Cx. tarsalis* | 65.2 (0.01) | 46656 (0.01) | 1.67 (0.01) | 0.692 (0.01) | 1071 (0.1) | 22.2 (0.1) |
| WEEV | *Cx. tarsalis* (0.01) | 82.2 | 35791 | 392 | 30.502 | 1264 | 26.1 |
| Vector competence (*bc*) | | | | | | |
| RVFV | *Ae. taeniorhynchus* (2) | 8.4 | 4775 | 2.316 | 0.421 | 147 | 3.39 |
| EEEV | *Ae. triseriatus* | 6.68 (3) | 3612 (3) | 2.027 (3) | 0.383 (3) | 119 (0.01) | 2.86 (0.01) |

*Continued on next page*

*Appendix 1—table 8 continued*

| Trait/System | q: $\alpha$ | q: $\beta$ | $T_{min}$: $\alpha$ | $T_{min}$: $\beta$ | $T_{max}$: $\alpha$ | $T_{max}$: $\beta$ |
|---|---|---|---|---|---|---|
| WNV \| *Cx. pipiens* (0.5) | 17.6 | 7857 | 1.403 | 0.534 | 219 | 5.42 |
| WEEV \| *Cx. tarsalis* (0.5) | 9.56 | 5344 | 3.021 | 0.498 | 180 | 4.05 |
| WNV \| *Cx. univittatus* | 13.7 (0.01) | 2327 (0.01) | 380 (0.01) | 22.434 (0.01) | 527 (0.1) | 14.4 (0.1) |
| Parasite dev. rate (*PDR*) | | | | | | |
| RVFV \| *Ae. taeniorhynchus* | 20.2 (0.2) | 331065 (0.2) | 8.69 (2) | 0.893 (2) | 227 (2) | 4.96 (2) |
| EEEV \| *Ae. triseriatus* (2) | 13.2 | 167635 | 6.76 | 0.609 | 183 | 4.05 |
| WNV \| *Cx. pipiens* | 8.71 (2) | 113904 (2) | 3.51 (5) | 0.356 (5) | 140 (2) | 3.17 (2) |
| WNV \| *Cx. quinquefasciatus* | 15.8 | 201154 | 8.09 | 0.772 | 202 | 4.44 |
| SLEV \| *Cx. tarsalis* | 11.8 | 151149 | 6.31 | 0.584 | 179 | 3.97 |
| WEEV \| *Cx. tarsalis* | 10.3 | 117795 | 9.97 (0.05) | 0.768 (0.05) | 162 | 3.62 |
| WNV \| *Cx. tarsalis* (2) | 11.7 | 148079 | 5.92 | 0.541 | 169 | 3.77 |
| WNV \| *Cx. univittatus* | 12.3 | 146439 | 9.02 (3) | 0.773 (3) | 174 (0.2) | 3.87 (0.2) |

**Appendix 1—table 9.** Priors for trait thermal response functions: lifespan.
Gamma distribution parameters ($\alpha$ [shape] and $\beta$ [rate]) for priors for fitting thermal response parameters (*m* and *z*). Scaled variances are noted in parentheses, either by the system name (applied to all parameters) or by individual parameters. See Appendix 1 section: *Priors for trait thermal responses*.

| Trait/System (var.) | m: $\alpha$ | m: $\beta$ | z: $\alpha$ | z: $\beta$ |
|---|---|---|---|---|
| Lifespan (*lf*) | | | | |
| *Ae. taeniorhynchus* (0.01) | 119 | 52.9 | 268 | 3.19 |
| *Cx. pipiens* (0.01) | 117 | 42.4 | 238 | 2.39 |
| *Cx. quinquefasciatus* (0.01) | 110 | 32.9 | 207 | 1.78 |
| *Cx. tarsalis* (0.1) | 124 | 43.0 | 249 | 2.42 |

**Appendix 1—table 10.** Model results for GAMs of mean incidence (per 1000 people) of West Nile neuroinvasive disease as a function of average summer temperature.
Statistics for models fit with differing numbers of knots: edf (estimated degrees of freedom), Ref-df, *F*, and p-value refer to the smoothed temperature term (see *Fig A24* for plots). Dev. exp. = percent deviance explained. $T_{opt}$ = temperature of peak incidence.

| Panel in *Fig A24* | # knots | edf | Ref-df | F | p-value | Adj. $R^2$ | Dev. exp. (%) | $T_{opt}$ |
|---|---|---|---|---|---|---|---|---|
| A | k = 4 | 2.96 | 2.99 | 15.87 | $4.03 \cdot 10^{-10}$ | 0.018 | 2.33 | 23.8°C |
| B | k = 5 | 3.77 | 3.97 | 11.11 | $4.64 \cdot 10^{-9}$ | 0.019 | 2.44 | 24.2°C |
| C | k = 6 | 4.71 | 4.96 | 11.97 | $4.77 \cdot 10^{-11}$ | 0.022 | 2.85 | 23.5°C |
| D | k = 7 | 5.53 | 5.92 | 11.01 | $1.31 \cdot 10^{-11}$ | 0.024 | 3.11 | 23.6°C |
| E | k = 8 | 6.55 | 6.93 | 11.12 | $2.73 \cdot 10^{-13}$ | 0.026 | 3.62 | 24.1°C |
| F | k = 9 | 7.19 | 7.80 | 10.06 | $3.17 \cdot 10^{-13}$ | 0.026 | 3.67 | 24.2°C |

## Priors for trait thermal responses

We used gamma distribution parameters ($\alpha$ [shape] and $\beta$ [rate]) for informative priors for each thermal response parameter (Brière and quadratic functions: $T_{min}$, $T_{max}$, and q; linear functions: *m* and *z*).

First, we fit a thermal response function (with uniform priors) to all the *Aedes* and *Culex* data for a given trait except that of the focal vector species or vector–virus pair (i.e. the parameters for the priors for *a* for *Culex pipiens* were fit to the *a* data for all species except *Cx. pipiens*). Then we used the 'MASS' package in R to fit a gamma distribution hyperparameters to the distribution from each thermal response parameters.

The mean of the gamma distribution is equal to α/β, while the variance is determined by the magnitude of the parameters (smaller values = higher variance). When fitting thermal responses, the appropriate strength for the priors depends on the amount of data used to fit the priors and the amount of the data for the focal trait. Prior strengths can be modified by scaling the variance (i.e. multiplying the gamma parameters by <1 to increase the variance or >1 to decrease the variance) without impacting the mean. In many cases we had to increase the variance because of the large number of data points used to fit priors. In a few cases, we had to decrease the variance (e.g. to constrain $T_{max}$ for Briere functions for *PDR* where we had no observations at high temperatures, in order to make it so *PDR* would not constrain $R_0$ where there was no data). For biting rate (*a*) for *Culex tarsalis*, we used a likelihood function where $T_{min}$ and *q* had data informed priors and $T_{max}$ had uniform priors (as used to fit the priors) in order to best capture the thermal response of the data.

## Sensitivity and uncertainty analyses

We performed two sensitivity analyses and one uncertainty analysis to understand what traits were most important for determining and contributing to uncertainty in the thermal limits and optima. For the first sensitivity analysis, we calculated the partial derivatives of $R_0$ with respect to each trait across temperature (*T*) and multiplied it by the derivative of the trait with temperature (i.e. the slope of the thermal response). *Equations A3-A6* (below) apply to both versions of the $R_0$ model (*Equations A1 and A2*). *Equation A3* is for to all traits (x) that appear once in the numerator. *Equation A4*, for biting rate (*a*), differs from previous analyses (*Johnson et al., 2015*; *Mordecai et al., 2017*; *Mordecai et al., 2013*; *Shocket et al., 2018*; *Tesla et al., 2018*) because biting rate was cubed to account for fecundity measured per gonotrophic cycle rather than per day. *Equation A5* is for parasite development rate (*PDR*), and *equation A6* is for lifespan (*lf*).

$$\frac{\partial R_0}{\partial x} \cdot \frac{\partial x}{\partial T} = \frac{R_0}{2x} \cdot \frac{\partial x}{\partial T} \tag{A3}$$

$$\frac{\partial R_0}{\partial a} \cdot \frac{\partial a}{\partial T} = \frac{3R_0}{2a} \cdot \frac{\partial a}{\partial T} \tag{A4}$$

$$\frac{\partial R_0}{\partial PDR} \cdot \frac{\partial PDR}{\partial T} = \frac{R_0}{2\,lf\,PDR^2} \cdot \frac{\partial PDR}{\partial T} \tag{A5}$$

$$\frac{\partial R_0}{\partial lf} \cdot \frac{\partial lf}{\partial T} = \frac{R_0(1+3PDR)}{2\,PDR\,lf^2} \cdot \frac{\partial lf}{\partial T} \tag{A6}$$

For the second sensitivity analysis, we held single traits constant while allowing all other traits to vary with temperature. For the uncertainty analysis, we calculated the 'total uncertainty' across temperature as the width of the 95% highest posterior density (HPD) interval across temperature for the full model. Then, we calculated the HPD for 'uncertainty for each trait' by fixing all traits except the focal trait at their posterior median value across temperature, while keeping the full posterior sample of the focal trait. Then, we divided the uncertainty for each trait by the total uncertainty, calculated across temperature, to estimate the proportion of uncertainty in $R_0$ that was due to the uncertainty in the focal trait.

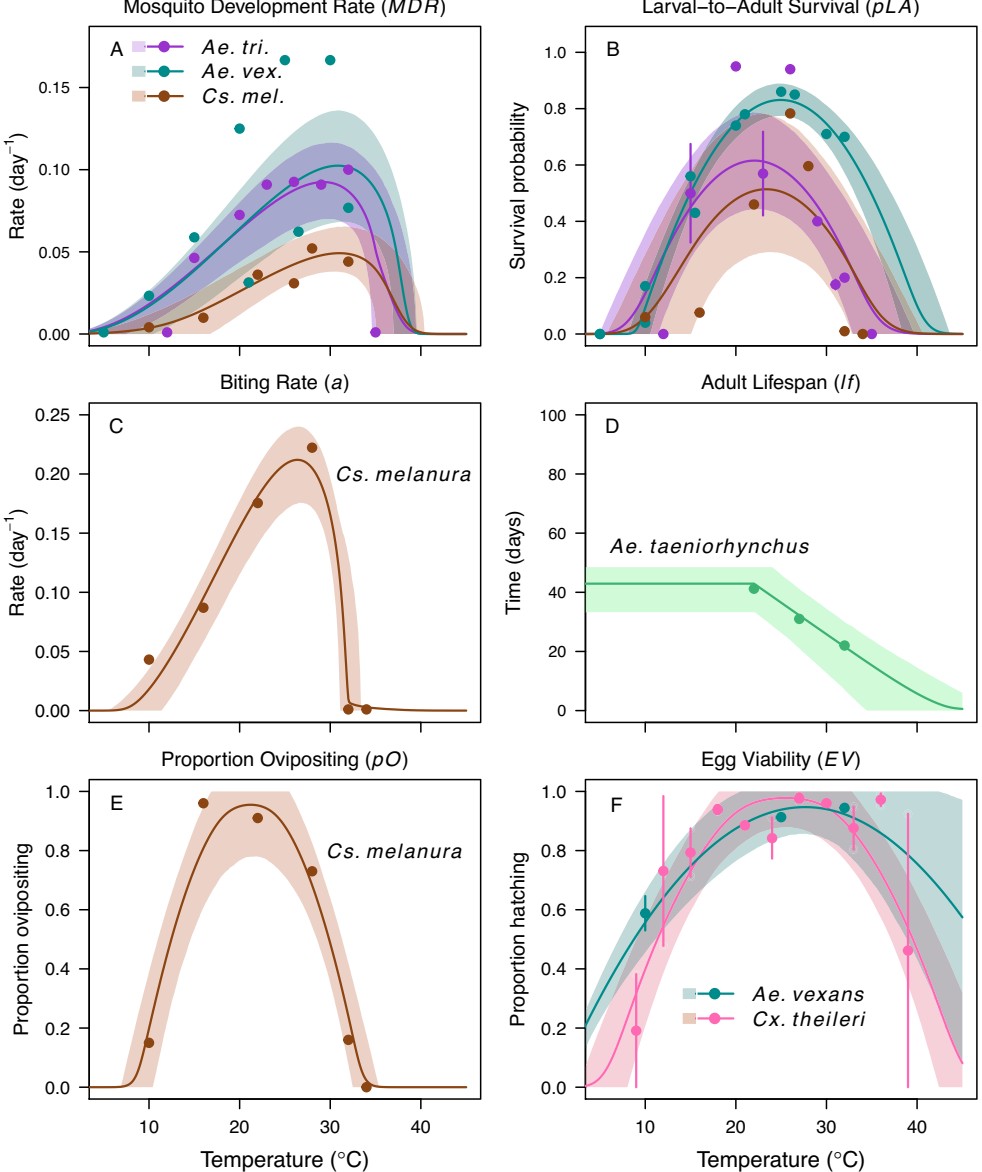

**Appendix 1—figure 1.** Thermal responses for mosquito traits in additional vector species. *Ae. taeniorhynchus* (green), *Ae. triseriatus* (violet), *Aedes vexans* (teal), *Cx. theileri* (pink), and *Culiseta melanura* (brown). (**A**) Mosquito development rate (*MDR*), (**B**) larval-to-adult survival (*pLA*), and (**C**) biting rate (a), (**D**) lifespan (*lf*), (**E**) proportion ovipositing (*pO*) and (**F**) egg viability (*EV*). Points without error bars are reported means from single studies; points with error bars are averages of means from multiple studies (+ / - standard error, for visual clarity only; thermal responses were fit to reported means). Solid lines are posterior distribution means; shaded areas are 95% credible intervals. The median thermal responses for these traits were printed in *Mordecai et al., 2019* (as part of *Figure 3*) without the trait data and 95% CIs and along with thermal responses for six other vectors.

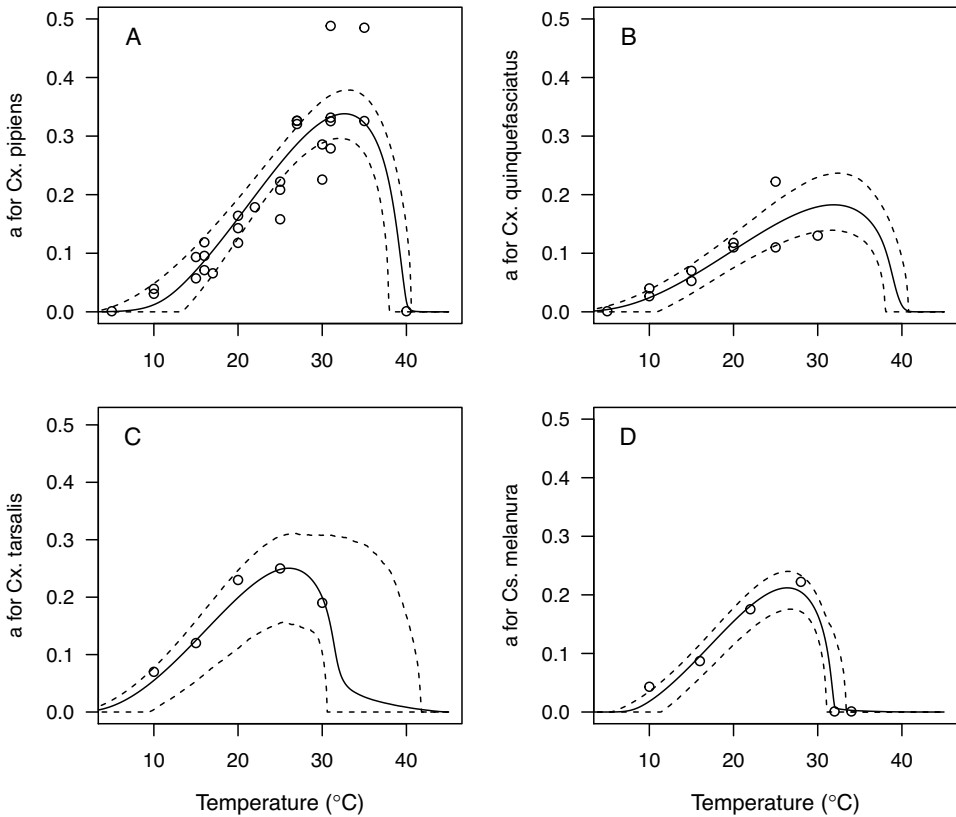

**Appendix 1—figure 2.** Thermal responses for biting rate (a) showing individual data points. (**A**) *Culex pipiens*, (**B**), *Cx. quinquefasciatus*, (**C**) *Cx. tarsalis*, and (**D**) *Culiseta melanura*. Solid lines are posterior distribution means for the mean thermal response; black dashed lines are 95% credible intervals for the mean thermal response; red dashed lines are 95% prediction intervals for observed data (incorporating the fitted variance).

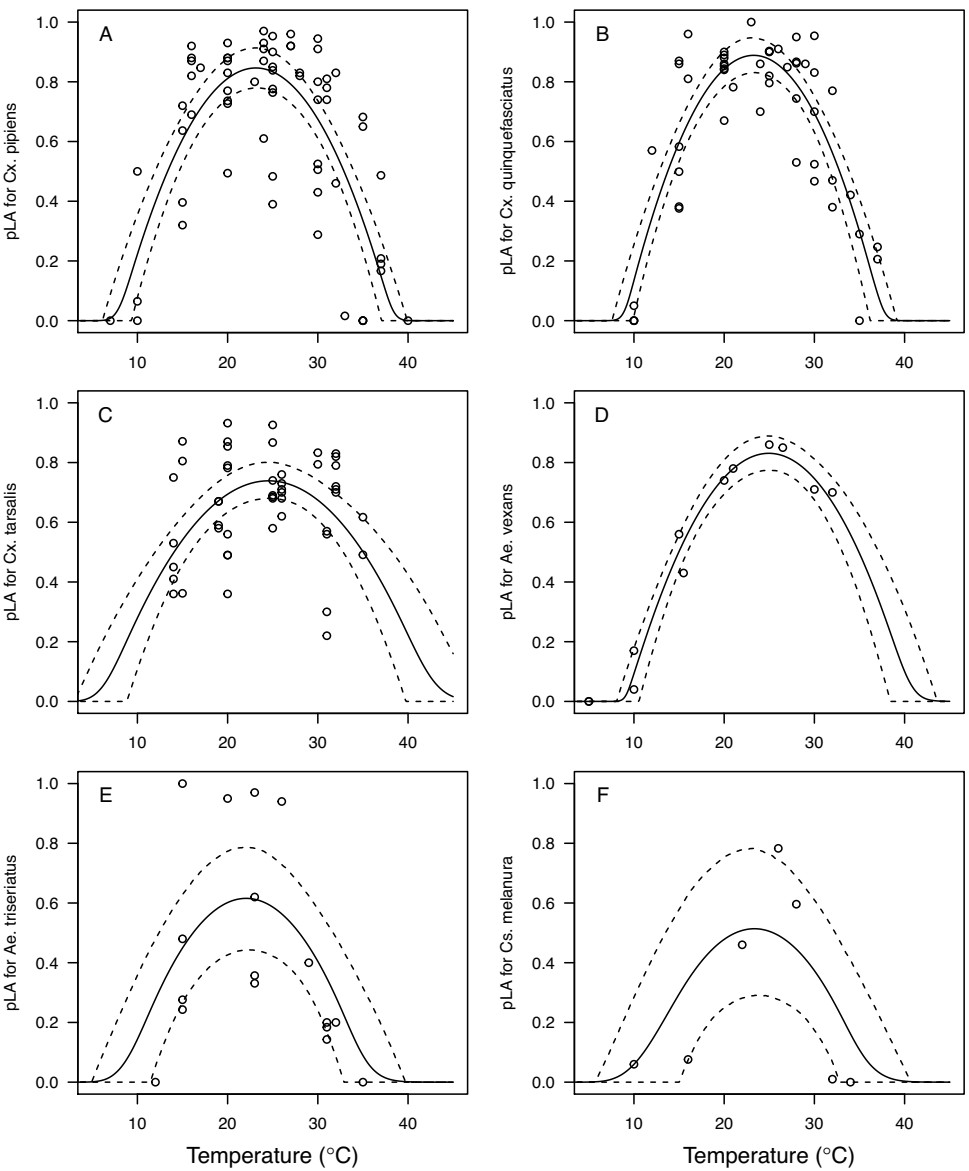

**Appendix 1—figure 3.** Thermal responses for larval-to-adult survival (*pLA*) showing individual data points. (**A**) *Culex pipiens*, (**B**), *Cx. quinquefasciatus*, (**C**) *Cx. tarsalis*, (**D**) *Aedes vexans*, (**E**) *Ae. triseriatus*, and (**F**) *Culiseta melanura*. Solid lines are posterior distribution means for the mean thermal response; black dashed lines are 95% credible intervals for the mean thermal response; red dashed lines are 95% prediction intervals for observed data (incorporating the fitted variance).

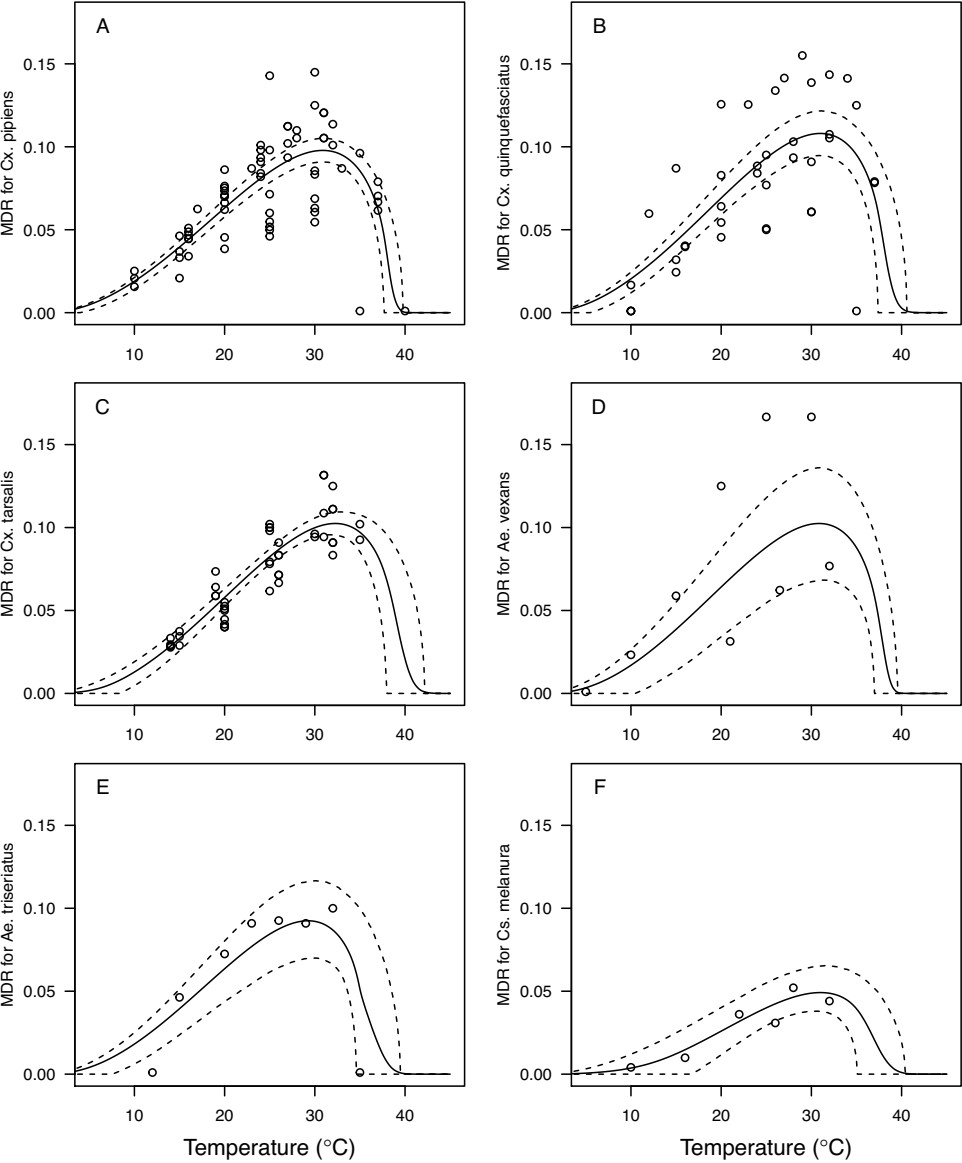

**Appendix 1—figure 4.** Thermal responses for mosquito development rate (*MDR*) showing individual data points. (**A**) *Culex pipiens*, (**B**), *Cx. quinquefasciatus*, (**C**) *Cx. tarsalis*, (**D**) *Aedes vexans*, (**E**) *Ae. triseriatus*, and (**F**) *Culiseta melanura*. Solid lines are posterior distribution means for the mean thermal response; black dashed lines are 95% credible intervals for the mean thermal response; red dashed lines are 95% prediction intervals for observed data (incorporating the fitted variance).

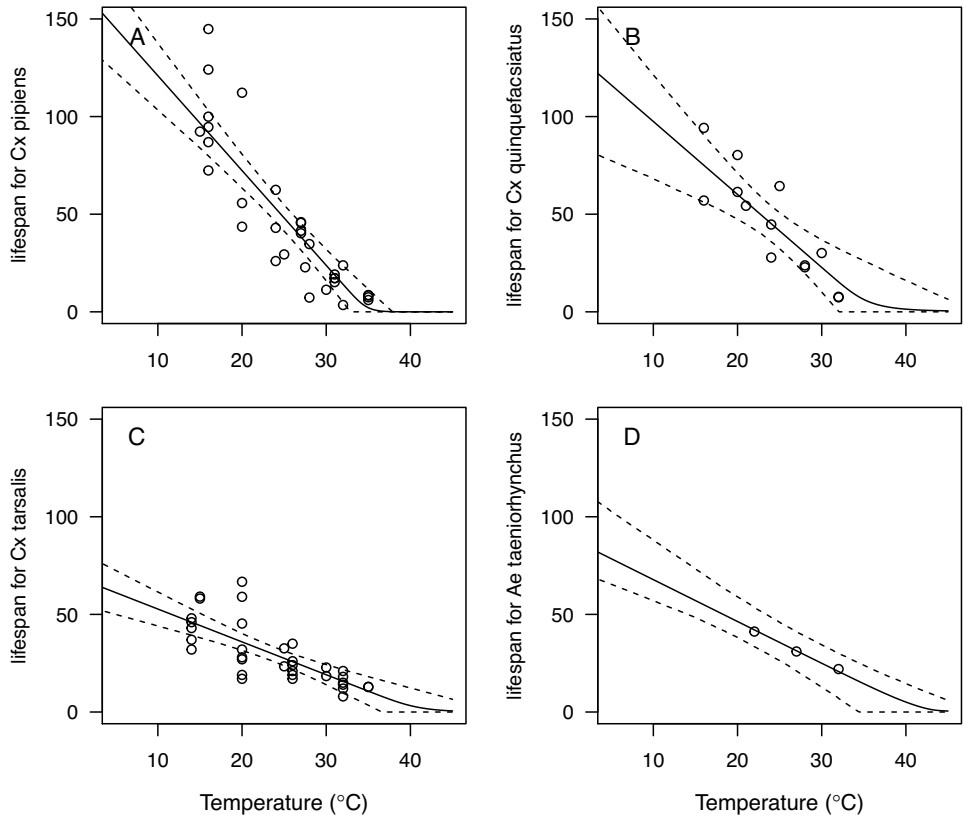

**Appendix 1—figure 5.** Thermal responses for adult mosquito lifespan (*lf*) showing individual data points. (**A**) *Culex pipiens*, (**B**), *Cx. quinquefasciatus*, (**C**) *Cx. tarsalis*, and (**D**) *Aedes taeniorhynchus*. When data were reported by sex, only female data were used. Solid lines are posterior distribution means for the mean thermal response; black dashed lines are 95% credible intervals for the mean thermal response; red dashed lines are 95% prediction intervals for observed data (incorporating the fitted variance).

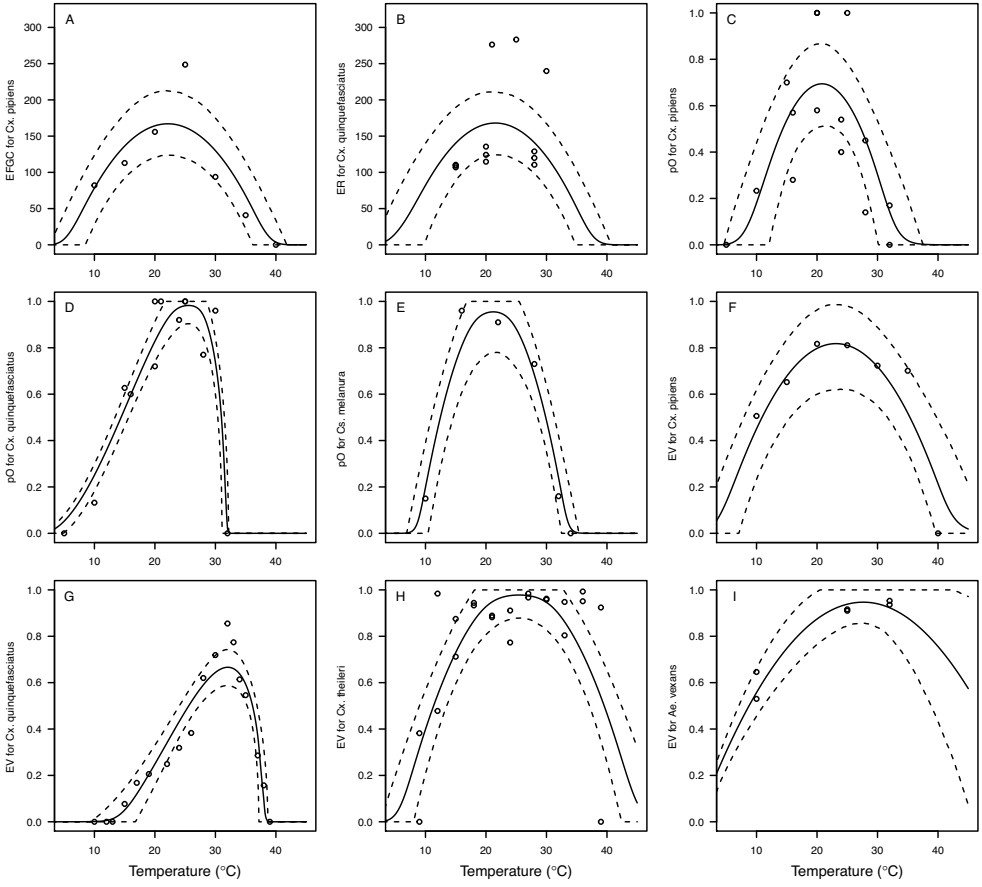

**Appendix 1—figure 6.** Thermal responses for fecundity traits showing individual data points. Traits: (**A**) Reproduction measured as eggs per female per gonotrophic cycle (*EFGC*), (**B**) reproduction measured as eggs per raft (*ER*) (C–E) proportion ovipositing (*pO*), and (F–I) egg viability (*EV*). Vector species: (**A,C,F**) *Culex pipiens*, (**B,D,G**) *Cx. quinquefasciatus*, (**E**) *Culiseta melanura*, (**H**) *Cx. theileri*, and (**I**) *Aedes vexans*. Solid lines are posterior distribution means for the mean thermal response; black dashed lines are 95% credible intervals for the mean thermal response; red dashed lines are 95% prediction intervals for observed data (incorporating the fitted variance).

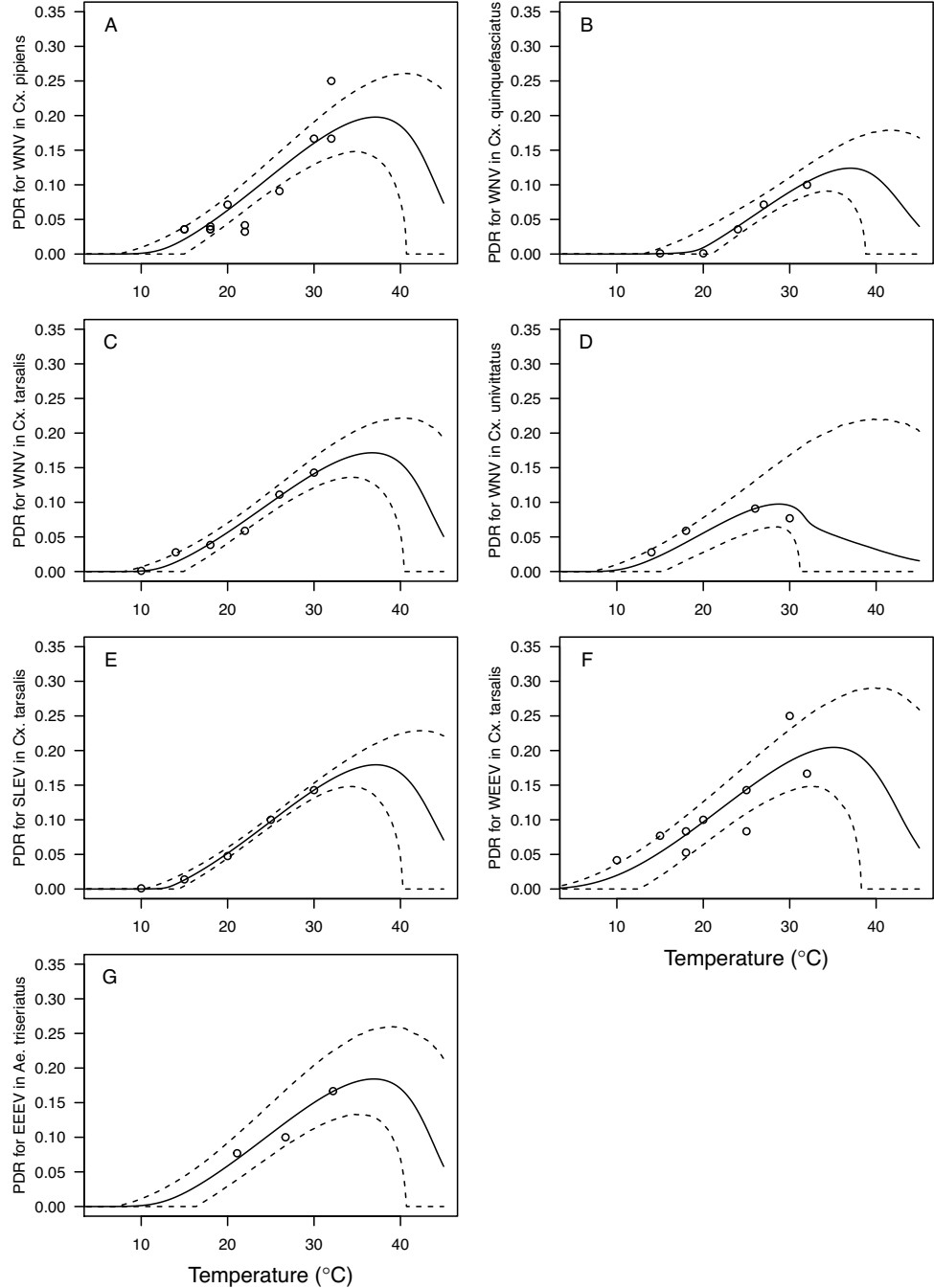

**Appendix 1—figure 7.** Thermal responses for pathogen development rate (*PDR*) showing individual data points. (**A**) West Nile virus (WNV) in *Culex pipiens*, (**B**), WNV in *Cx. quinquefasciatus*, (**C**) WNV in *Cx. tarsalis*, (**D**) WNV in *Cx. univittatus*, (**E**) St. Louis Encephalitis virus (SLEV) in *Cx. tarsalis*, (**F**) Western Equine Encephalitis virus (WEEV) in *Cx. tarsalis*, and (**G**) Eastern Equine Encephalitis virus (EEEV) in *Aedes triseriatus*. Solid lines are posterior distribution means for the mean thermal response; black dashed lines are 95% credible intervals for the mean thermal response; red dashed lines are 95% prediction intervals for observed data (incorporating the fitted variance).

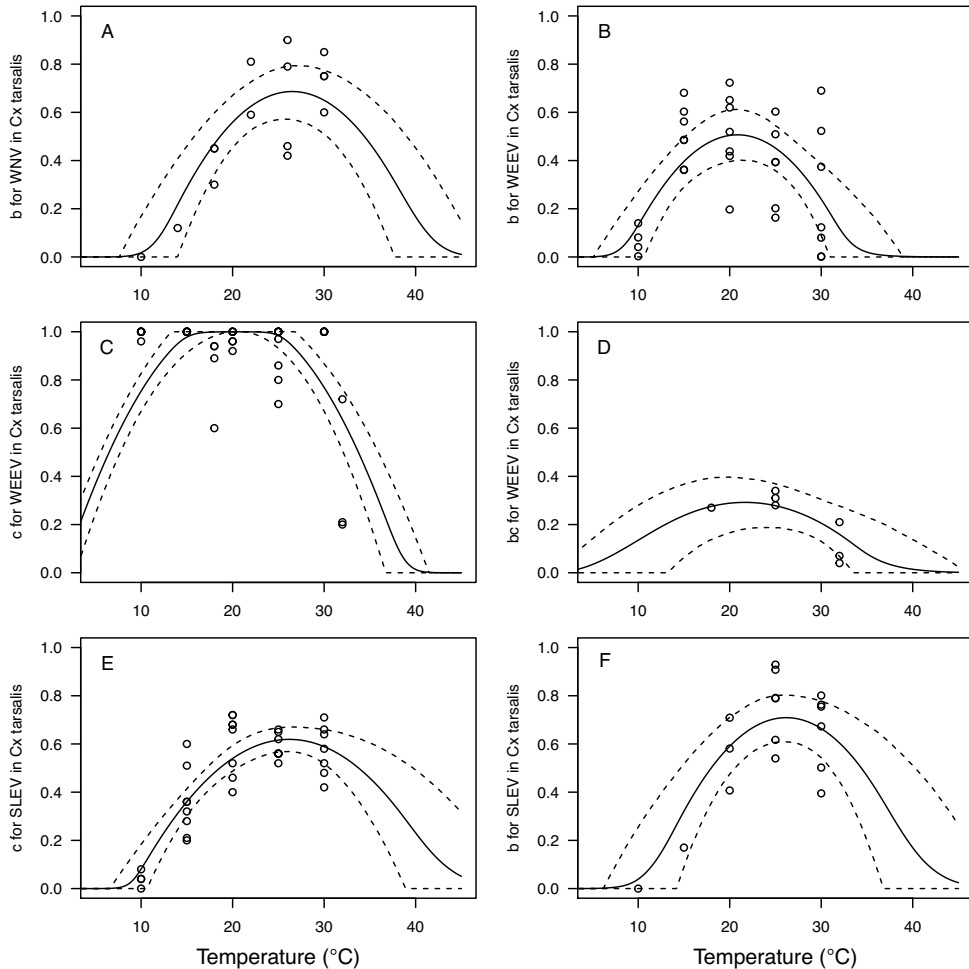

**Appendix 1—figure 8.** Thermal responses for vector competence traits in *Culex tarsalis*, showing individual data points. Traits: (**A,B,F**) transmission efficiency (*b*, # transmitting / # infected), (**C,E**) infection efficiency (*c*, # infected / # exposed), and (**D**) vector competence (*bc*, # infected / # exposed). Viruses: (**A**) West Nile virus (WNV), (**B–D**) Western Equine Encephalitis virus (WEEV), (**E,F**) St. Louis Encephalitis virus (SLEV). Solid lines are posterior distribution means for the mean thermal response; black dashed lines are 95% credible intervals for the mean thermal response; red dashed lines are 95% prediction intervals for observed data (incorporating the fitted variance).

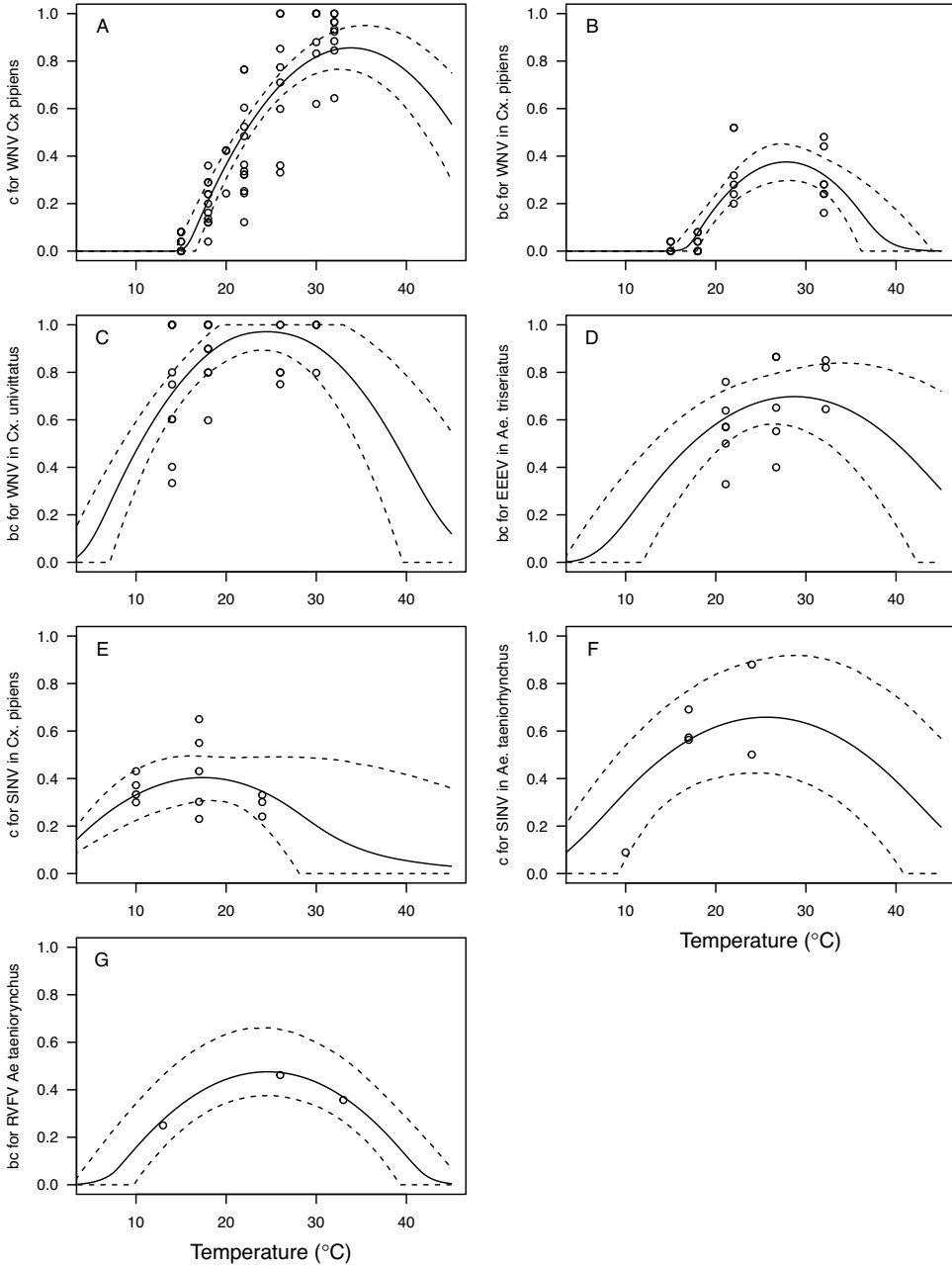

**Appendix 1—figure 9.** Thermal responses for vector competence traits showing individual data points. Traits: (**A,E,F**) infection efficiency (*c*, # infected / # exposed) and (**B,C,D,G**) vector competence (*bc*, # infected / # exposed). Viruses and vectors: (**A,B**) West Nile virus (WNV) in *Culex pipiens*, (**C**) WNV in *Cx. univittatus*, (**D**) Eastern Equine Encephalitis virus (EEEV) in *Ae. triseriatus*, (**E**) Sindbis virus (SINV) in *Culex pipiens*, (**F**) SINV in *Aedes taeniorhynchus*, and (**G**) Rift Valley Fever virus (RVFV) in *Ae. taeniorhynchus*. Solid lines are posterior distribution means for the mean thermal response; black dashed lines are 95% credible intervals for the mean thermal response; red dashed lines are 95% prediction intervals for observed data (incorporating the fitted variance).

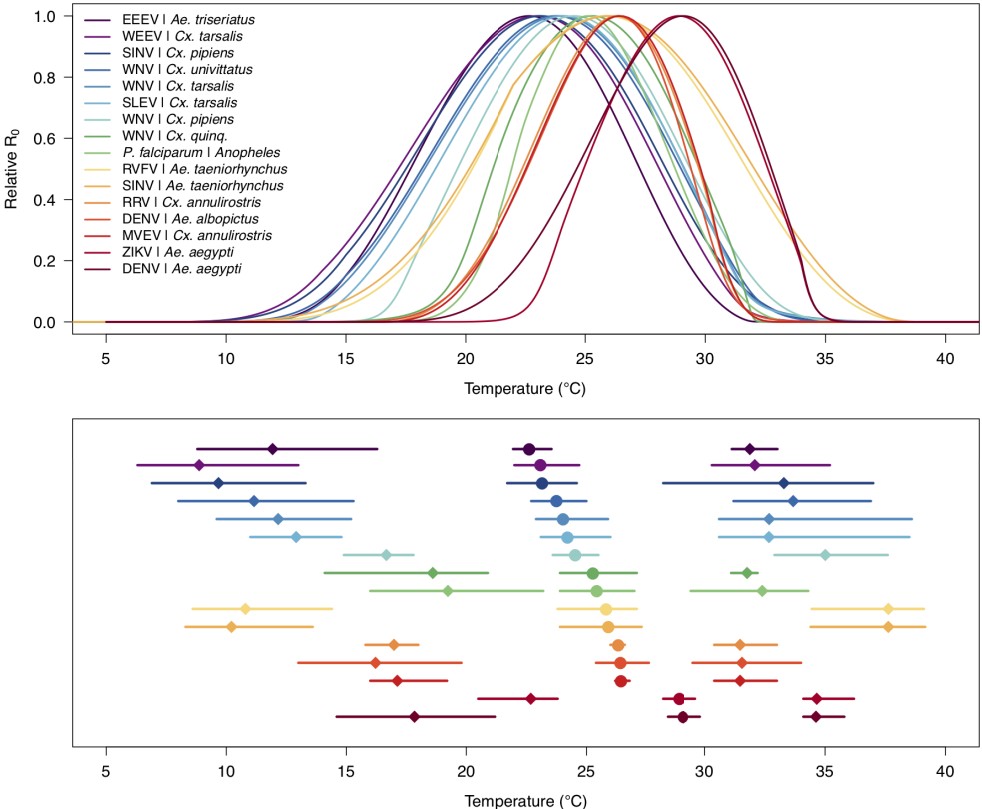

**Appendix 1—figure 10.** Medians and 95% credible intervals for thermal limits and optima of $R_0$ models across temperate and tropical mosquito-borne disease systems. Models in order from top to bottom: Eastern Equine Encephalitis virus (EEEV) in *Aedes triseriatus* (dark purple; this paper), Western Equine Encephalitis virus (WEEV) in *Culex. tarsalis* (light purple; this paper), Sindbis virus (SINV) in *Cx. pipiens* (dark blue; this paper), West Nile virus (WNV) in *Cx. univittatus* (medium blue; this paper), WNV in *Cx. tarsalis* (light blue, this paper), St. Louis Encephalitis virus (SLEV) in *Cx. tarsalis* (dark teal; this paper), WNV in *Cx. pipiens* (light teal; this paper), WNV in *Cx. quinquefasciatus* (dark green; this paper), *Plasmodium falciparum* malaria in *Anopheles* spp. (light green; *Johnson et al., 2015*), Rift Valley Fever virus (RVFV) in *Ae. taeniorhynchus* (yellow; this paper), SINV in *Ae. taeniorhynchus* (light orange; this paper), Ross River virus (RRV) in *Cx. annulirostris* (medium orange; *Shocket et al., 2018*), dengue virus (DENV) in *Ae. albopictus* (dark orange; *Mordecai et al., 2017*), Murray Valley Encephalitis virus (MVEV) in *Cx. annulirostris* (light red; *Shocket et al., 2018*), Zika virus (ZIKV) in *Ae. aegypti* (medium red; *Tesla et al., 2018*), DENV in *Ae. aegypti* (dark red; *Mordecai et al., 2017*). Figure is identical to Figure 2 in *Mordecai et al., 2019*.

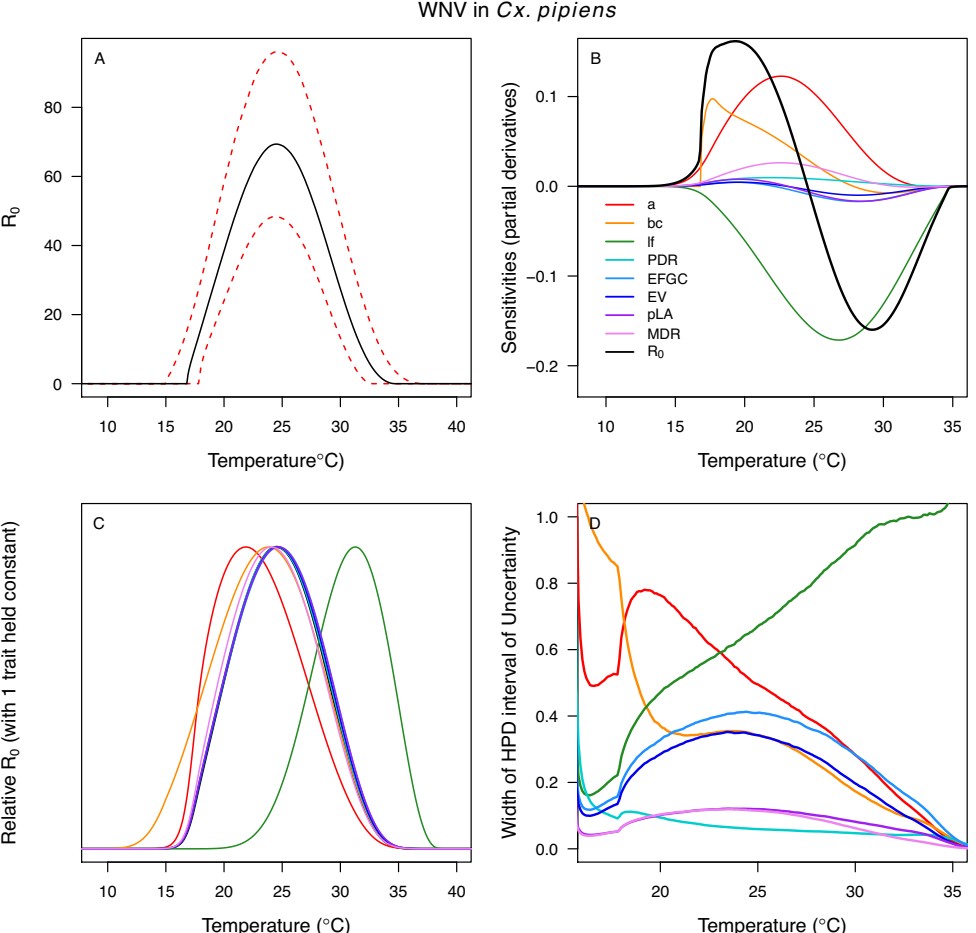

**Appendix 1—figure 11.** Temperature-dependent $R_0$, sensitivity analyses, and uncertainty analysis for model of West Nile Virus (WNV) in *Culex pipiens*. (**A**) Median temperature-dependent $R_0$ (black line) with 95% credible intervals (dashed red lines). (**B**) Sensitivity analysis #1: derivative with respect to temperature for $R_0$ (black) and partial derivatives with respect to temperature for each trait. (**C**) Sensitivity analysis #2: relative $R_0$ calculated with single traits held constant. (**D**) Uncertainty analysis using highest posterior density (HPD) interval widths: the proportion of total uncertainty due to each trait. (**B–D**) Trait colors: biting rate (*a*, red), vector competence (*bc*, orange), adult lifespan (*lf*, green), parasite development rate (*PDR*, cyan), fecundity (*EFGC*, light blue), egg viability (*EV*, dark blue), larval survival (*pLA*, purple), and mosquito development rate (*MDR*, pink). All traits from *Cx. pipiens*.

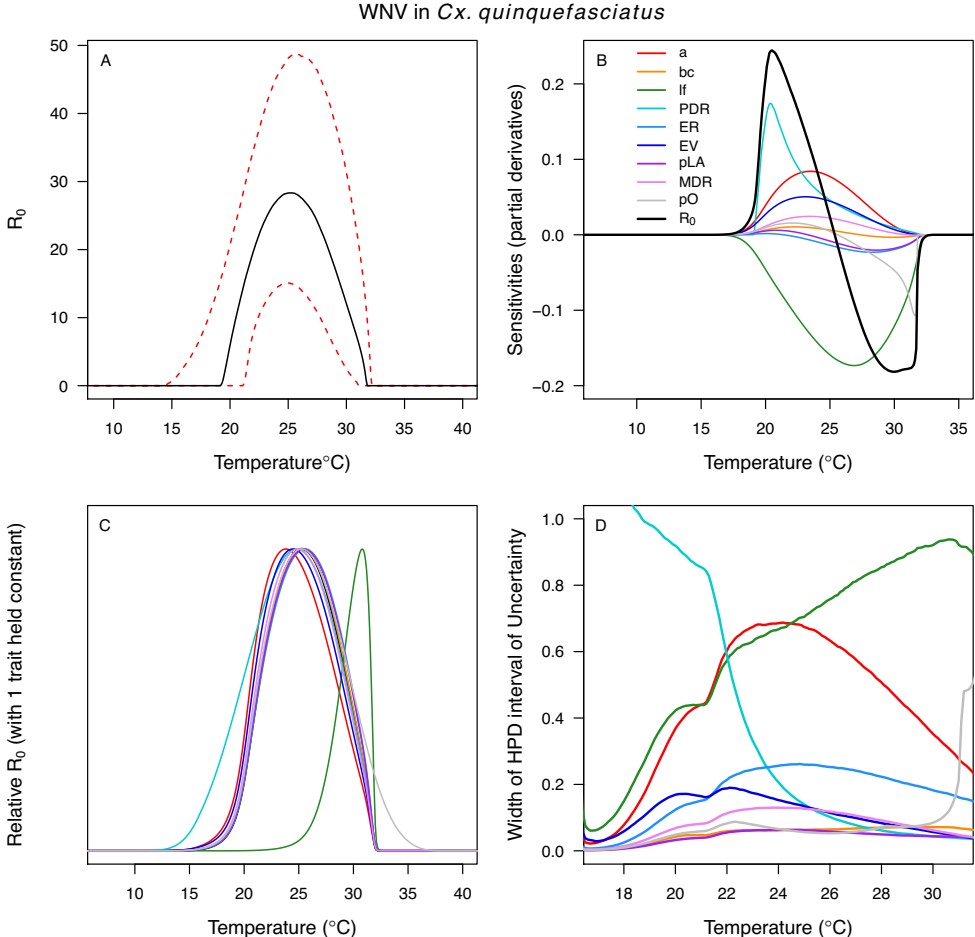

**Appendix 1—figure 12.** Temperature-dependent $R_0$, sensitivity analyses, and uncertainty analysis for model of West Nile Virus (WNV) in *Culex quinquefasciatus*. (**A**) Median temperature-dependent $R_0$ (black line) with 95% credible intervals (dashed red lines). (**B**) Sensitivity analysis #1: derivative with respect to temperature for $R_0$ (black) and partial derivatives with respect to temperature for each trait. (**C**) Sensitivity analysis #2: relative $R_0$ calculated with single traits held constant. (**D**) Uncertainty analysis using highest posterior density (HPD) interval widths: the proportion of total uncertainty due to each trait. (**B–D**) Trait colors: biting rate (*a*, red), vector competence (*bc*, orange), adult lifespan (*lf*, green), parasite development rate (*PDR*, cyan), fecundity (*EFGC*, light blue), egg viability (*EV*, dark blue), larval survival (*pLA*, purple), mosquito development rate (*MDR*, pink), and proportion ovipositing (*pO*, grey). Vector competence (*bc*) from *Cx. univitattus*; all other traits from *Cx. quinquefasciatus*.

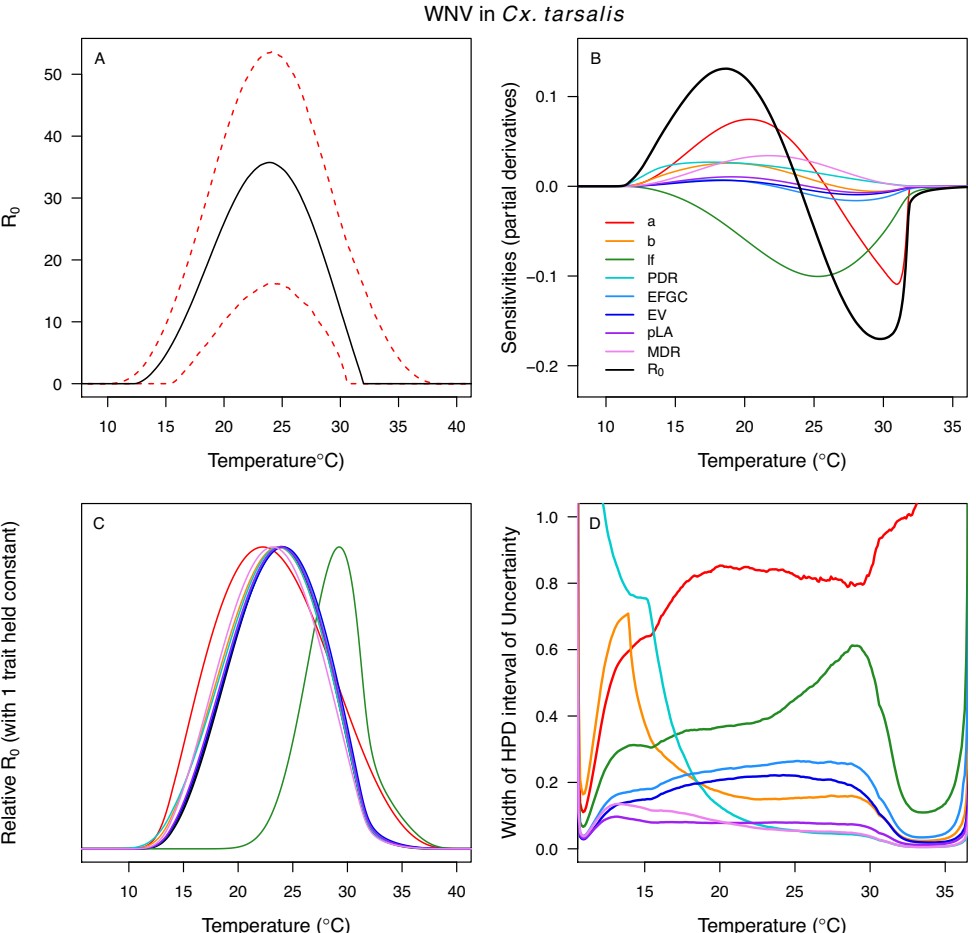

**Appendix 1—figure 13.** Temperature-dependent $R_0$, sensitivity analyses, and uncertainty analysis for model of West Nile Virus (WNV) in *Culex tarsalis*. (**A**) Median temperature-dependent $R_0$ (black line) with 95% credible intervals (dashed red lines). (**B**) Sensitivity analysis #1: derivative with respect to temperature for $R_0$ (black) and partial derivatives with respect to temperature for each trait. (**C**) Sensitivity analysis #2: relative $R_0$ calculated with single traits held constant. (**D**) Uncertainty analysis using highest posterior density (HPD) interval widths: the proportion of total uncertainty due to each trait. (**B–D**) Trait colors: biting rate (*a*, red), transmission efficiency (*b*, orange), adult lifespan (*lf*, green), parasite development rate (*PDR*, cyan), fecundity (*EFGC*, light blue), egg viability (*EV*, dark blue), larval survival (*pLA*, purple), and mosquito development rate (*MDR*, pink). Fecundity (*EFGC*) and egg viability (*EV*) from *Cx. pipiens*; all other traits from *Cx. tarsalis*.

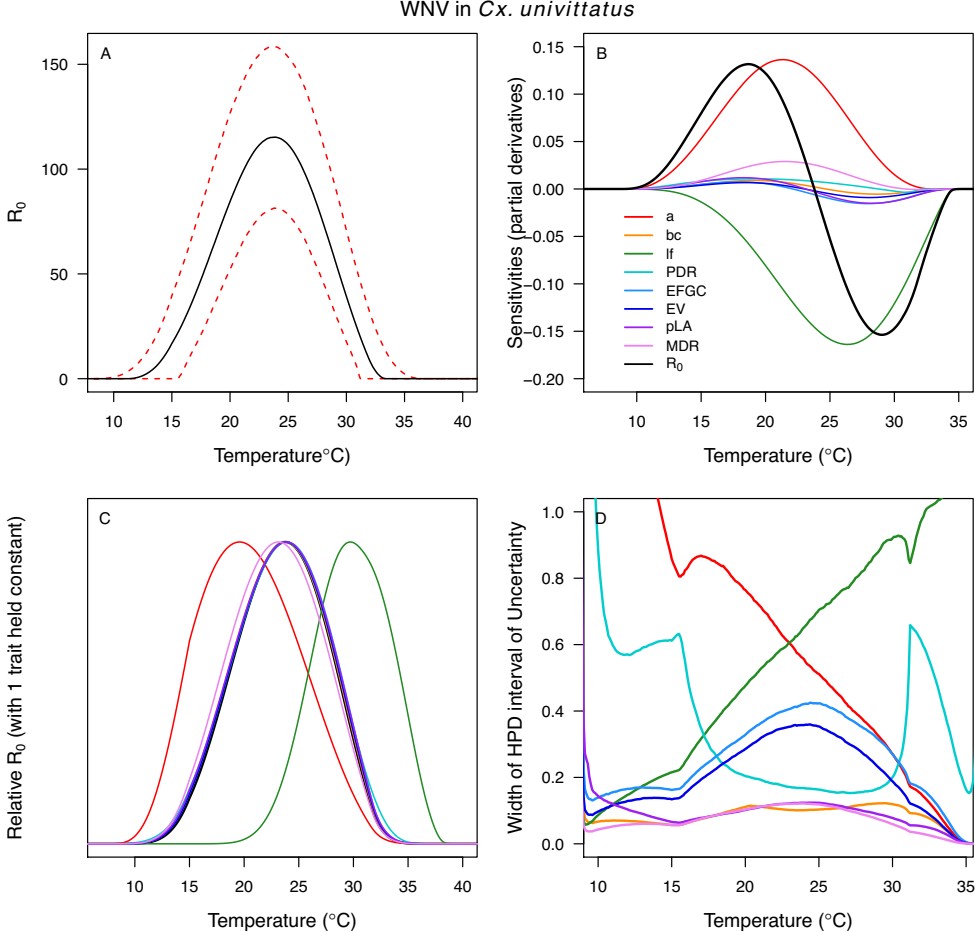

**Appendix 1—figure 14.** Temperature-dependent $R_0$, sensitivity analyses, and uncertainty analysis for model of West Nile Virus (WNV) in *Culex univittatus*. (**A**) Median temperature-dependent $R_0$ (black line) with 95% credible intervals (dashed red lines). (**B**) Sensitivity analysis #1: derivative with respect to temperature for $R_0$ (black) and partial derivatives with respect to temperature for each trait. (**C**) Sensitivity analysis #2: relative $R_0$ calculated with single traits held constant. (**D**) Uncertainty analysis using highest posterior density (HPD) interval widths: the proportion of total uncertainty due to each trait. (**B–D**) Trait colors: biting rate (*a*, red), vector competence (*bc*, orange), adult lifespan (*lf*, green), parasite development rate (*PDR*, cyan), fecundity (*EFGC*, light blue), egg viability (*EV*, dark blue), larval survival (*pLA*, purple), and mosquito development rate (*MDR*, pink). Infection traits (*bc* and *PDR*) from *Cx. univittatus*; all other traits from *Cx. pipiens*.

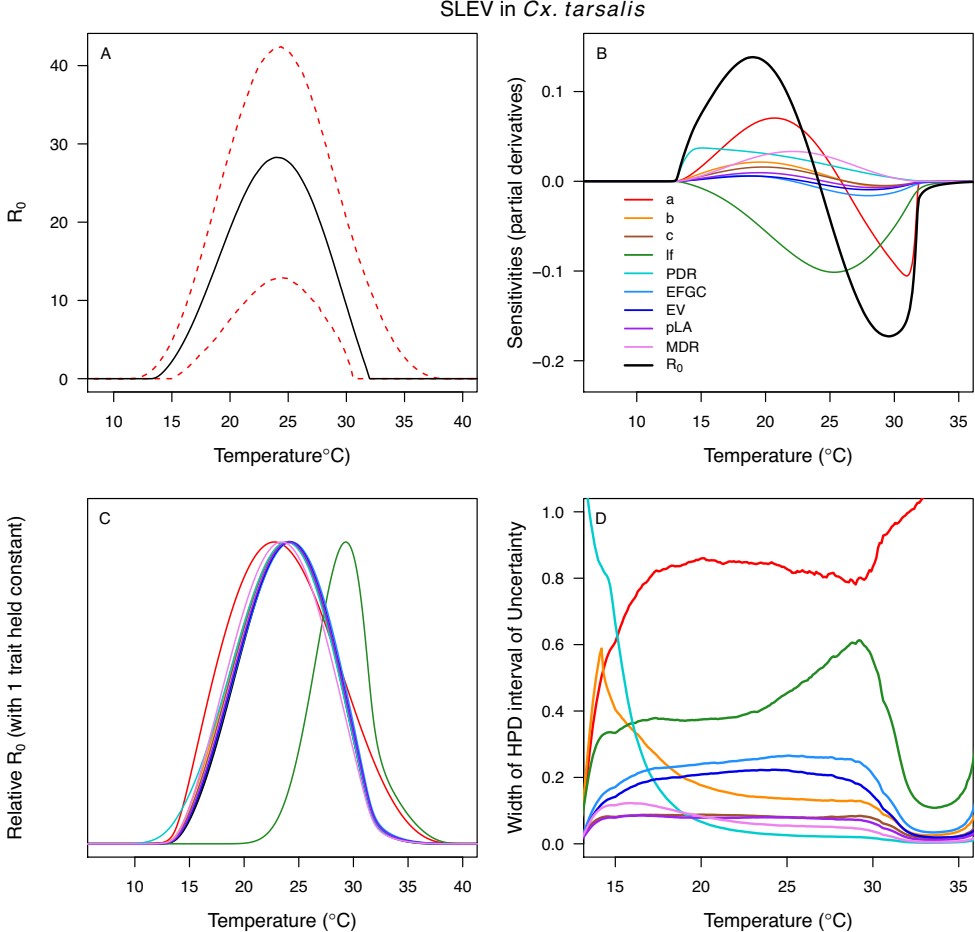

**Appendix 1—figure 15.** Temperature-dependent $R_0$, sensitivity analyses, and uncertainty analysis for St. model of St. Louis Encephalitis Virus (SLEV) in *Culex tarsalis*. (**A**) Median temperature-dependent $R_0$ (black line) with 95% credible intervals (dashed red lines). (**B**) Sensitivity analysis #1: derivative with respect to temperature for $R_0$ (black) and partial derivatives with respect to temperature for each trait. (**C**) Sensitivity analysis #2: relative $R_0$ calculated with single traits held constant. (**D**) Uncertainty analysis using highest posterior density (HPD) interval widths: the proportion of total uncertainty due to each trait. (**B–D**) Trait colors: biting rate (*a*, red), transmission efficiency (*b*, orange), infection efficiency (*c*, brown), adult lifespan (*lf*, green), parasite development rate (*PDR*, cyan), fecundity (*EFGC*, light blue), egg viability (*EV*, dark blue), larval survival (*pLA*, purple), and mosquito development rate (*MDR*, pink). Fecundity (*EFGC*) and egg viability (*EV*) from *Cx. pipiens*; all other traits from *Cx. tarsalis*.

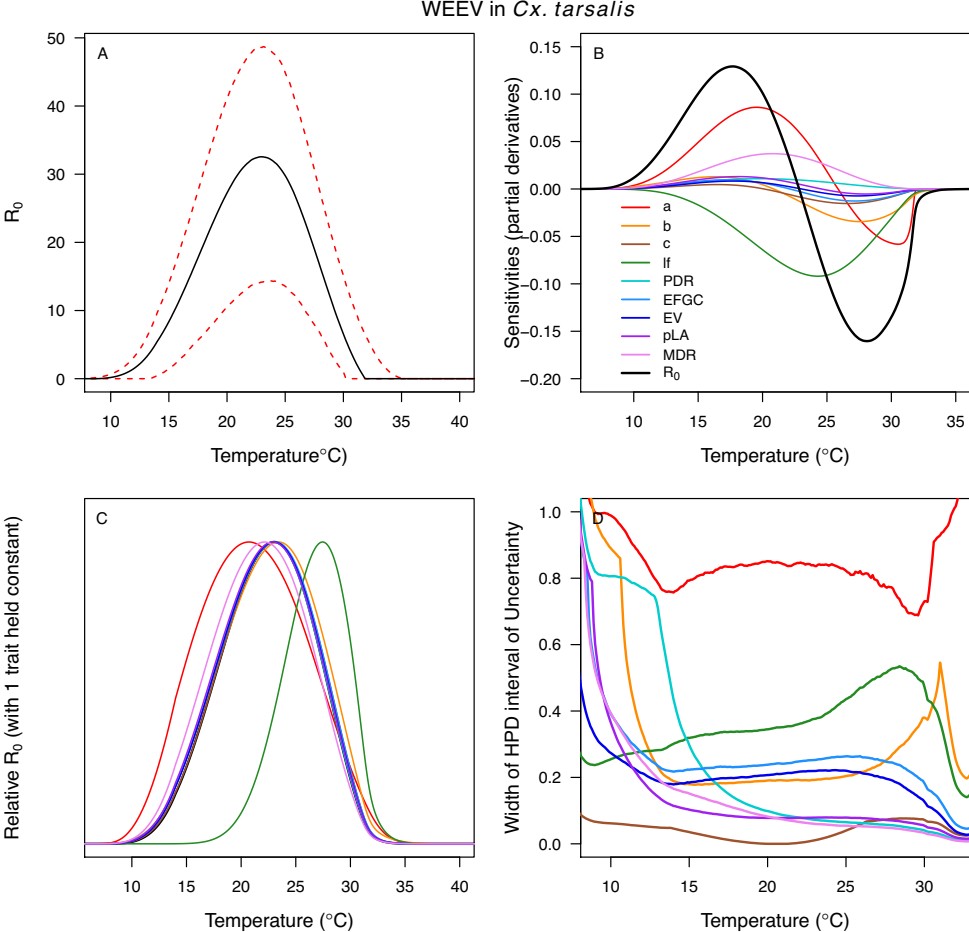

**Appendix 1—figure 16.** Temperature-dependent $R_0$, sensitivity analyses, and uncertainty analysis for model of Western Equine Encephalitis Virus (WEEV) in *Culex tarsalis*. (**A**) Median temperature-dependent $R_0$ (black line) with 95% credible intervals (dashed red lines). (**B**) Sensitivity analysis #1: derivative with respect to temperature for $R_0$ (black) and partial derivatives with respect to temperature for each trait. (**C**) Sensitivity analysis #2: relative $R_0$ calculated with single traits held constant. (**D**) Uncertainty analysis using highest posterior density (HPD) interval widths: the proportion of total uncertainty due to each trait. (**B–D**) Trait colors: biting rate (*a*, red), transmission efficiency (*b*, orange), infection efficiency (*c*, brown), adult lifespan (*lf*, green), parasite development rate (*PDR*, cyan), fecundity (*EFGC*, light blue), egg viability (*EV*, dark blue), larval survival (*pLA*, purple), and mosquito development rate (*MDR*, pink). Fecundity (*EFGC*) and egg viability (*EV*) from *Cx. pipiens*; all other traits from *Cx. tarsalis*.

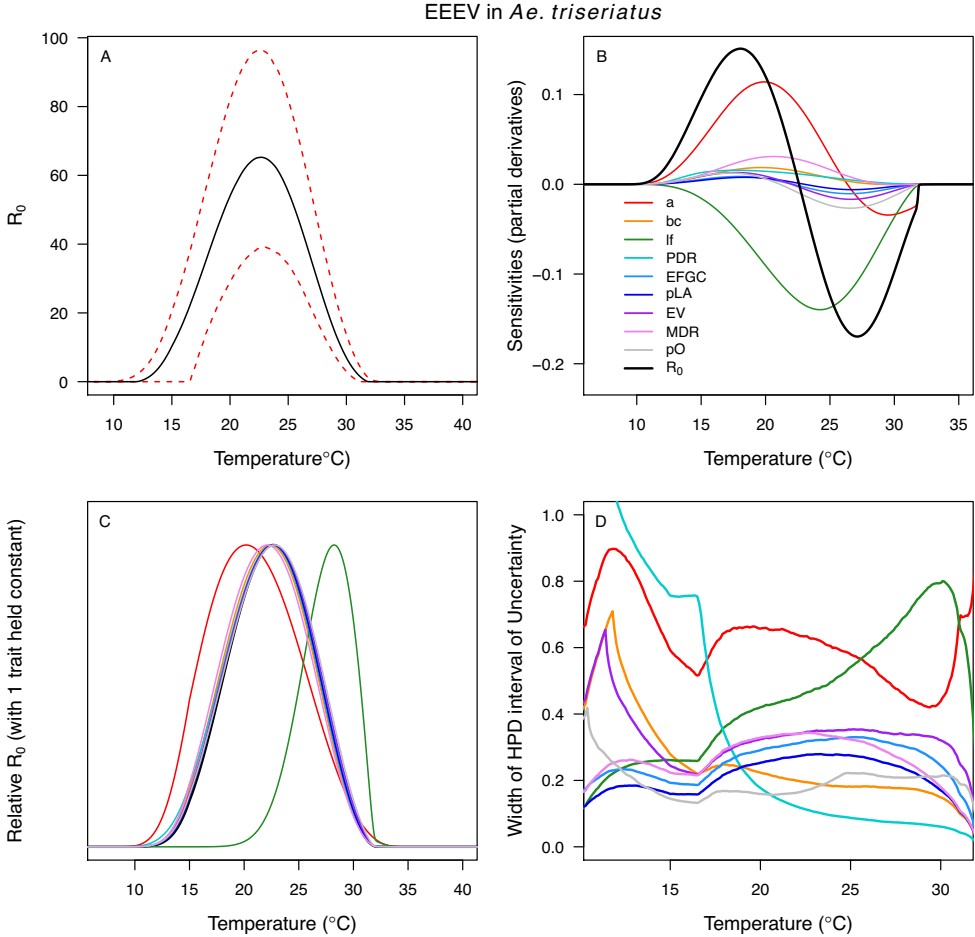

**Appendix 1—figure 17.** Temperature-dependent $R_0$, sensitivity analyses, and uncertainty analysis for model of Eastern Equine Encephalitis Virus in *Aedes triseriatus*. (**A**) Median temperature-dependent $R_0$ (black line) with 95% credible intervals (dashed red lines). (**B**) Sensitivity analysis #1: derivative with respect to temperature for $R_0$ (black) and partial derivatives with respect to temperature for each trait. (**C**) Sensitivity analysis #2: relative $R_0$ calculated with single traits held constant. (**D**) Uncertainty analysis using highest posterior density (HPD) interval widths: the proportion of total uncertainty due to each trait. (**B–D**) Trait colors: biting rate (*a*, red), vector competence (*bc*, orange), adult lifespan (*lf*, green), parasite development rate (*PDR*, cyan), fecundity (*EFGC*, light blue), egg viability (*EV*, dark blue), larval survival (*pLA*, purple), mosquito development rate (*MDR*, pink), and proportion ovipositing (*pO*, grey). Fecundity (*EFGC*), egg viability (*EV*), and lifespan (*lf*) from *Cx. pipiens*; biting rate (*a*) and proportion ovipositing (*pO*) from *Culiseta melanura*; all other traits from *Ae. triseriatus*. Note: technically fecundity as eggs per female per gonotrophic cycle (*EFGC*) has already accounted for the proportion ovipositing (*pO*). However, we selected this trait fit because it was very similar to the *ER* thermal response from *Cx. quinquefasciatus*, but slightly wider (more conservative).

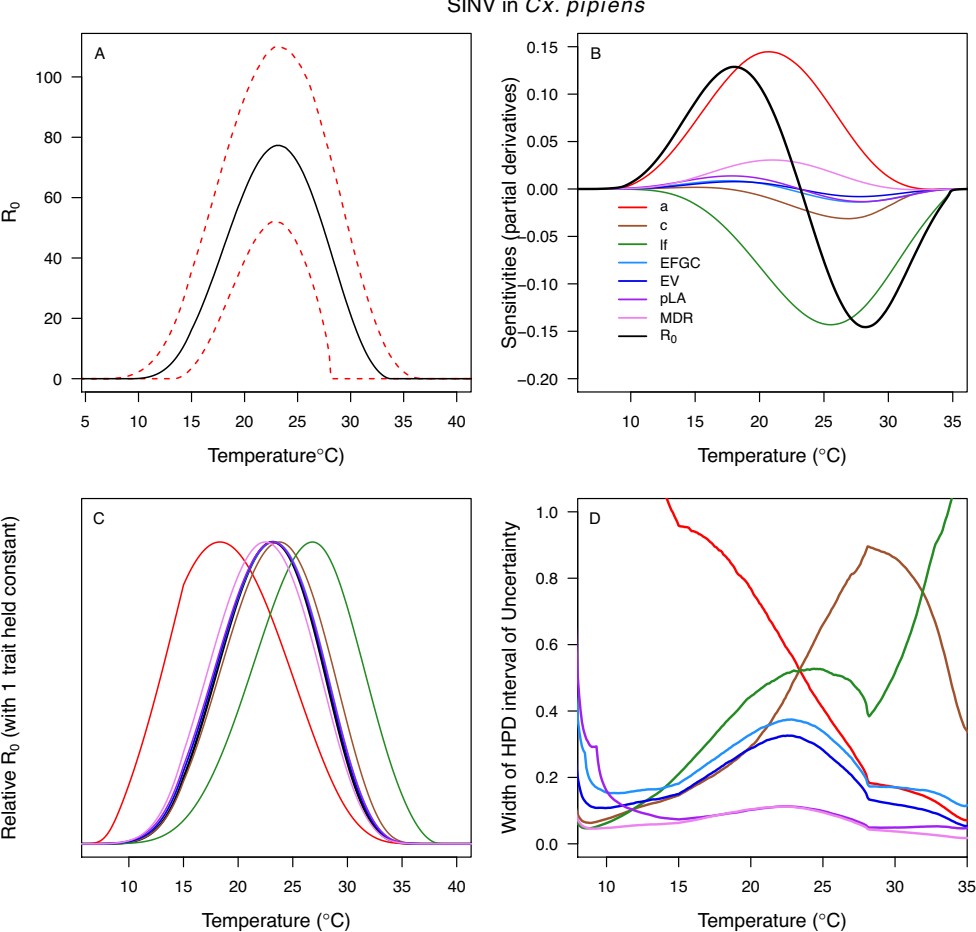

**Appendix 1—figure 18.** Temperature-dependent $R_0$, sensitivity analyses, and uncertainty analysis for model of Sindbis Virus in *Culex pipiens*. (**A**) Median temperature-dependent $R_0$ (black line) with 95% credible intervals (dashed red lines). (**B**) Sensitivity analysis #1: derivative with respect to temperature for $R_0$ (black) and partial derivatives with respect to temperature for each trait. (**C**) Sensitivity analysis #2: relative $R_0$ calculated with single traits held constant. (**D**) Uncertainty analysis using highest posterior density (HPD) interval widths: the proportion of total uncertainty due to each trait. (**B–D**) Trait colors: biting rate (*a*, red), infection efficiency (*c*, brown), adult lifespan (*lf*, green), fecundity (*EFGC*, light blue), egg viability (*EV*, dark blue), larval survival (*pLA*, purple), and mosquito development rate (*MDR*, pink). All traits from *Cx. pipiens*. NOTE: The raw $R_0$ calculation used *PDR* = 1, which is not biologically reasonable trait value.

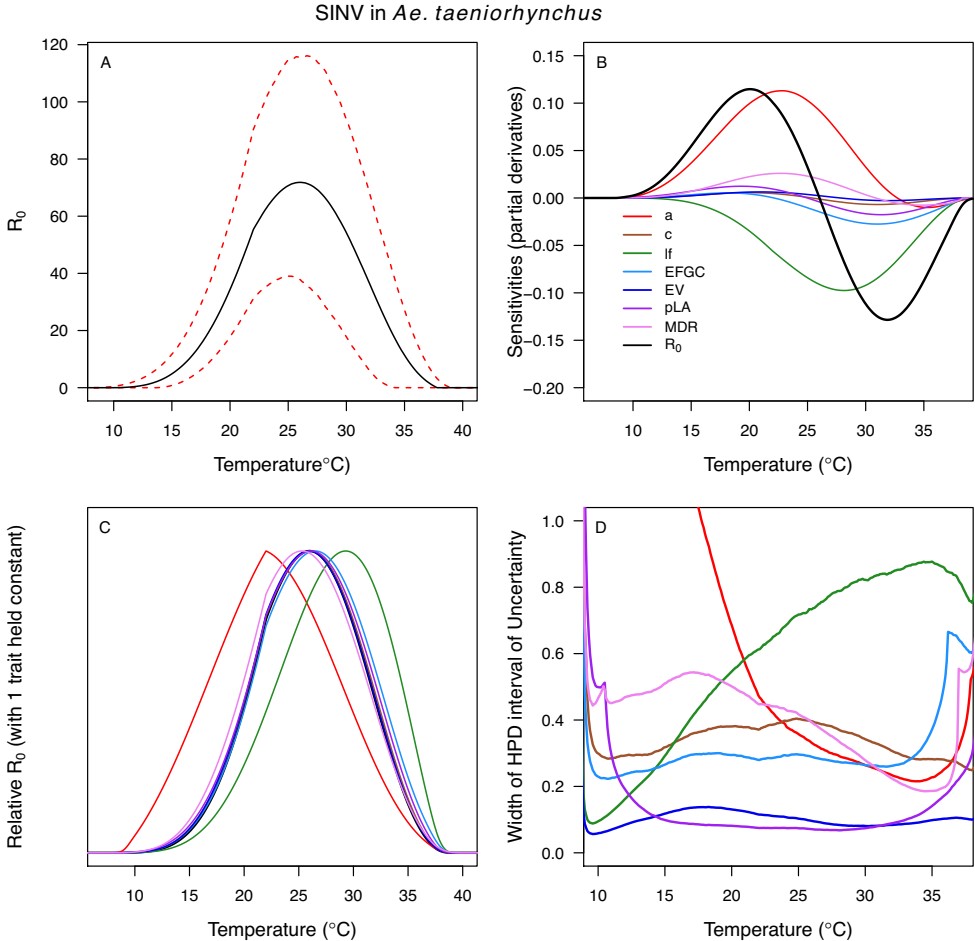

**Appendix 1—figure 19.** Temperature-dependent $R_0$, sensitivity analyses, and uncertainty analysis for model of Sindbis Virus in *Aedes taeniorhynchus*. (**A**) Median temperature-dependent $R_0$ (black line) with 95% credible intervals (dashed red lines). (**B**) Sensitivity analysis #1: derivative with respect to temperature for $R_0$ (black) and partial derivatives with respect to temperature for each trait. (**C**) Sensitivity analysis #2: relative $R_0$ calculated with single traits held constant. (**D**) Uncertainty analysis using highest posterior density (HPD) interval widths: the proportion of total uncertainty due to each trait. (**B–D**) Trait colors: biting rate (*a*, red), infection efficiency (*c*, brown), adult lifespan (*lf*, green), fecundity (*EFGC*, light blue), egg viability (*EV*, dark blue), larval survival (*pLA*, purple), and mosquito development rate (*MDR*, pink). Fecundity (*EFGC*) and biting rate (*a*) from *Culex pipiens*; egg viability (EV) and larval traits (*pLA* and *MDR*) from *Ae. vexans*; all other traits from *Ae. taeniorhynchus*. NOTE: The raw $R_0$ calculation used *PDR* = 1, which is not biologically reasonable trait value.

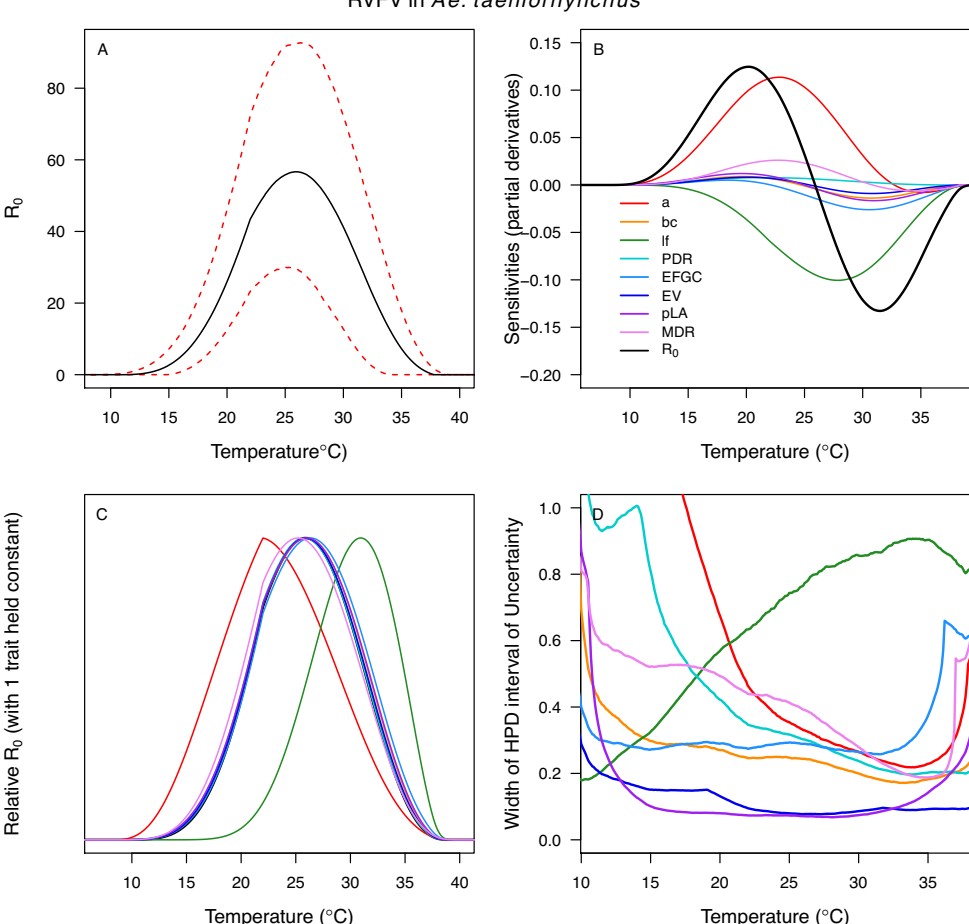

**Appendix 1—figure 20.** Temperature-dependent $R_0$, sensitivity analyses, and uncertainty analysis for model of Rift Valley Fever Virus in *Aedes taeniorhynchus*. (**A**) Median temperature-dependent $R_0$ (black line) with 95% credible intervals (dashed red lines). (**B**) Sensitivity analysis #1: derivative with respect to temperature for $R_0$ (black) and partial derivatives with respect to temperature for each trait. (**C**) Sensitivity analysis #2: relative $R_0$ calculated with single traits held constant. (**D**) Uncertainty analysis using highest posterior density (HPD) interval widths: the proportion of total uncertainty due to each trait. (**B–D**) Trait colors: biting rate (*a*, red), vector competence (*bc*, orange), adult lifespan (*lf*, green), parasite development rate (*PDR*, cyan), fecundity (*EFGC*, light blue), egg viability (*EV*, dark blue), larval survival (*pLA*, purple), and mosquito development rate (*MDR*, pink). Fecundity (*EFGC*) and biting rate (*a*) from *Culex pipiens*; egg viability (EV) from *Cx. theileri*; larval traits (*pLA* and *MDR*) from *Ae. vexans*; all other traits from *Ae. taeniorhynchus*.

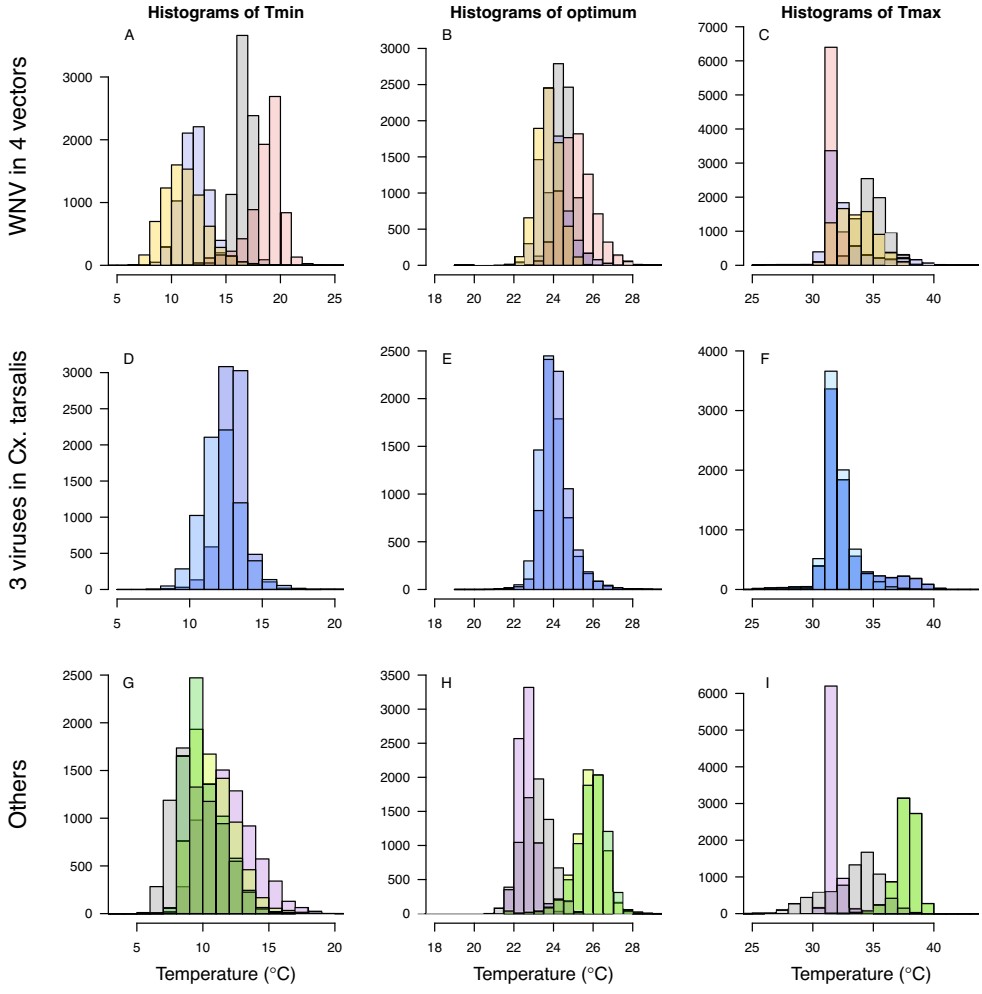

**Appendix 1—figure 21.** Histograms of $T_{min}$, optimum, and $T_{max}$ for transmission ($R_0$) models. $T_{min}$ (left column), optimum (center column), and $T_{max}$ (right column). Top row (A–C): West Nile virus (WNV) in four vectors: *Culex pipiens* (grey), *Cx. quinquefasciatus* (red), *Cx. tarsalis* (blue), and *Cx. univitattus* (orange). Middle row (D–F): three viruses in *Cx. tarsalis*: WNV (same as in top row, bright blue), Western Equine Encephalitis virus (WEEV, light blue), and St. Louis Encephalitis virus (SLEV, dark blue). Bottom row (H–J): Sindbis virus (SINV) in *Aedes taeniorhynchus* (grey), SINV in *Cx. pipiens* (dark green), Rift Valley Fever virus (RVFV) in *Ae. taeniorhynchus* (light green), and Eastern Equine Encephalitis virus (EEEV) in *Ae. triseriatus* (purple).

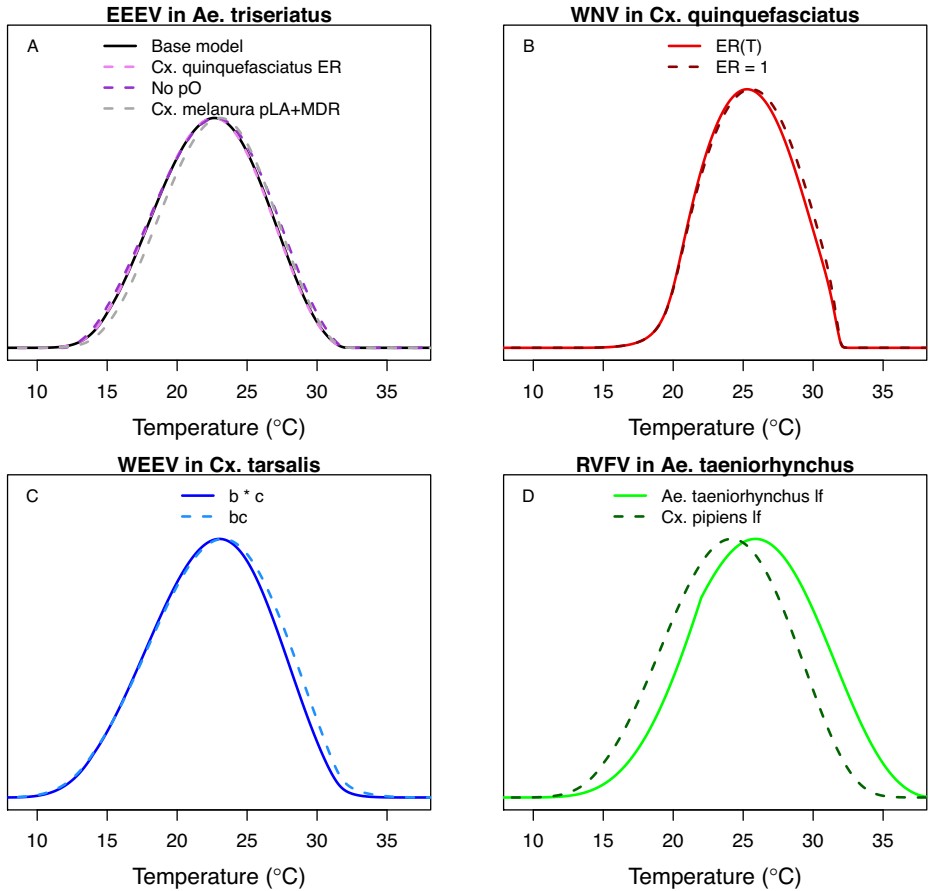

**Appendix 1—figure 22.** Comparing alternative model parameterizations. Several models had multiple potentially valid choices for traits; we show these alternative models here (dashed lines; base models from main text in solid lines) to show that they make very little difference, except in D. (**A**) Models for EEEV in *Ae. triseriatus* with larval traits (larval-to-adult survival [*pLA*] and mosquito development rate [*MDR*]) from *Ae. triseriatus* (violet, from the main text) and larval traits from *Cs. melanura* (black). We also show larval traits from *Cs. melanura* without proportion ovipositing (*pO*) in the model (grey), since the thermal responses for *EFGC* (eggs per female per gonotrophic cycle, in *Cx. pipiens*) and *ER* (eggs per raft, in *Cx. quinquefasciatus*) were nearly identical even though the units were different, probably because the ER data were not very informative and the priors strongly shaped the thermal response. (**B**) Models for WNV in *Cx. quinquefasciatus*, with (light red, from the main text) and without (dark red) the thermal response for fecundity (as eggs per raft, *ER*), for the same reason as in A. (**C**) Models for WEEV in *Cx. tarsalis* with vector competence estimated by infection efficiency (*c*, *Figure 6D*) and transmission efficiency (*b*, *Figure 6E*) measured separately (blue, from the main text) or by vector competence measured as a single trait (*bc*, *Figure 6F*; light blue). (**D**) Models for RVFV in *Ae. taeniorhynchus* with lifespan from *Ae. taeniorhynchus* (light green, from the main text) or from *Cx. pipiens* (dark green). We chose the *Ae. taeniorhynchus* version for the main text because it is the same species the infection traits (*PDR*, *bc*) were measured in, and that choice strongly impacted the results.

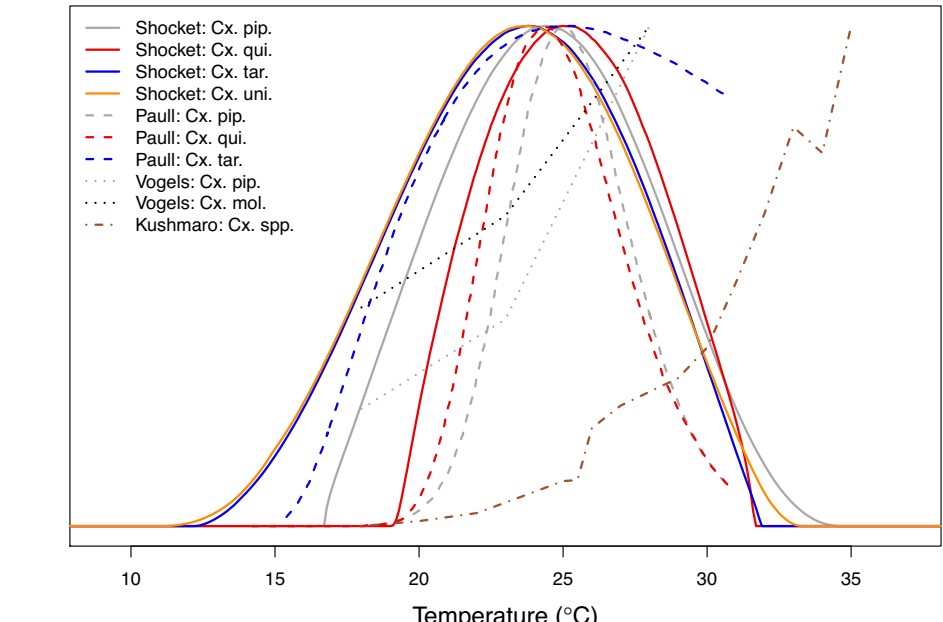

**Appendix 1—figure 23.** Comparison with previous $R_0$ models for transmission of West Nile virus. Models taken from this paper (solid lines: *Cx. pipiens* [grey], *Cx. quinquefasciatus* [red], *Cx. tarsalis* [blue], and *Cx. univittatus* [orange]), from ***Paull et al., 2017*** (dashed lines: *Cx. pipiens* [grey], *Cx. quinquefasciatus* [red], and *Cx. tarsalis* [blue]), from ***Vogels et al., 2017*** (*Cx. pipiens* [grey] and *Cx. pipiens molestus* [black]), and from ***Kushmaro et al., 2015*** (not species specific, dot-dashed line [brown]).

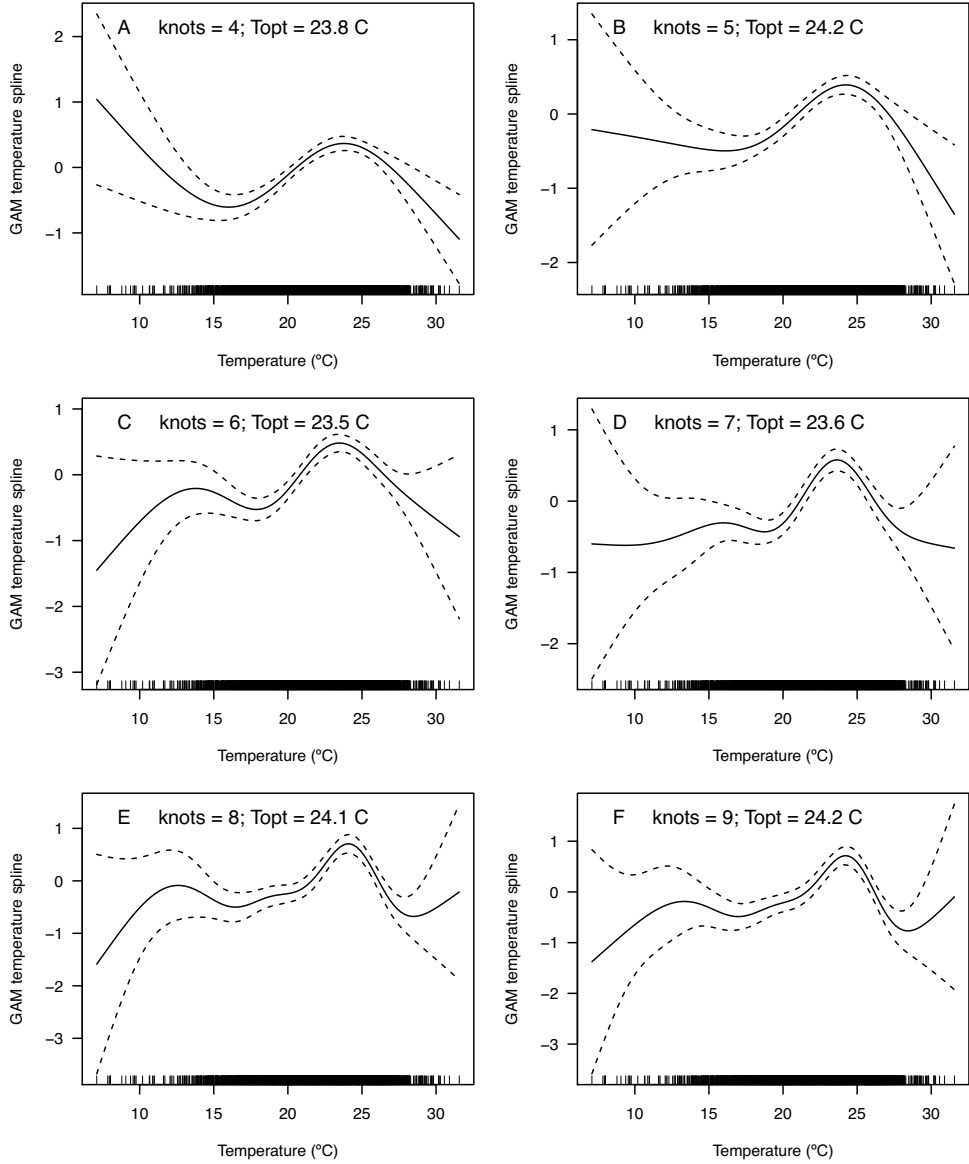

**Appendix 1—figure 24.** Temperature splines from GAMs of mean incidence (per 1000 people) of West Nile neuroinvasive disease as a function of average summer temperature. (**A–F**) Models are fit with differing numbers of knots (4–9). In all models, incidence peaks around 24°C ($T_{opt}$ = 23.5–24.2° C).

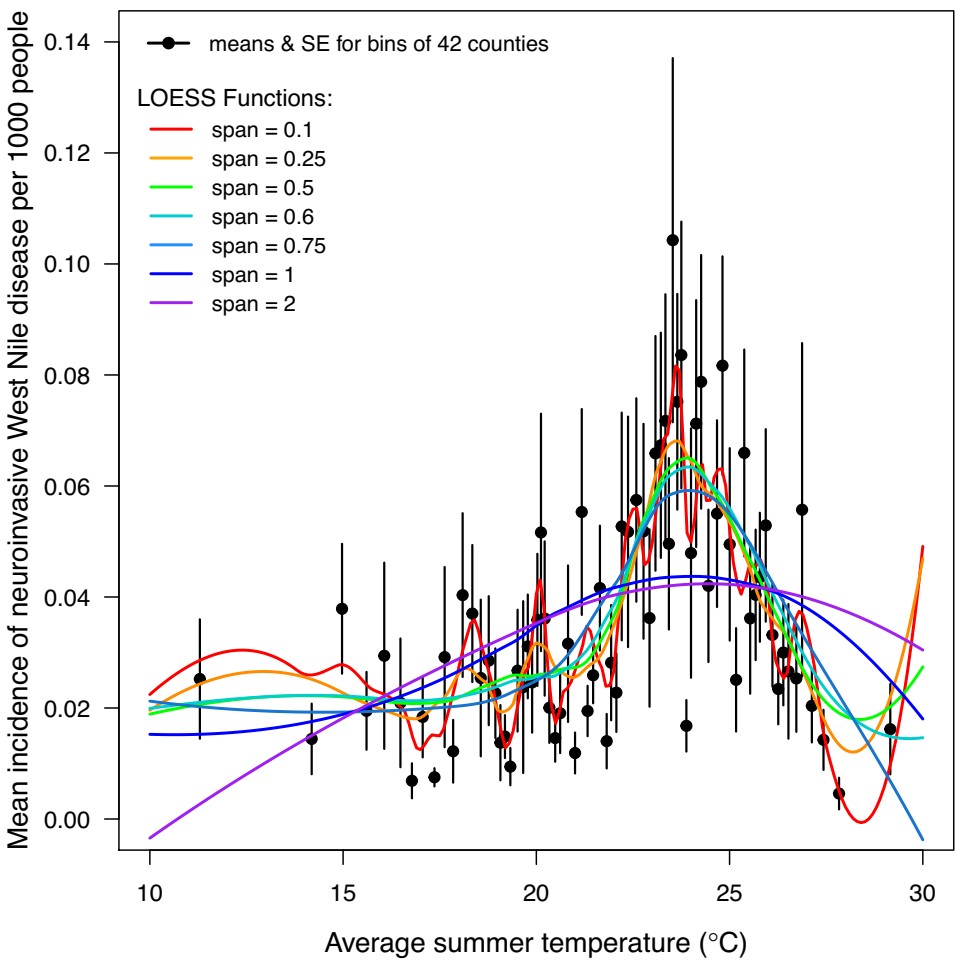

**Appendix 1—figure 25.** LOESS models of mean incidence (per 1000 people) of West Nile neuroinvasive disease (2000–2016) as a function of average summer temperature. Points are means for bins of 42 counties (+ / - SE). Lines are locally estimated scatterplot smoothing (LOESS) regression models with different smoothing (span) parameters: 0.1 (red), 0.25 (orange), 0.5 (green), 0.6 (cyan), 0.75 (light blue), 1 (dark blue), and 2 (violet). Models were fit to raw county-level data (n = 3109, binned for visual clarity). The best model (span = 0.6, which appropriately balances overfitting and underfitting the data) estimates that incidence peaks at 23.9˚C.

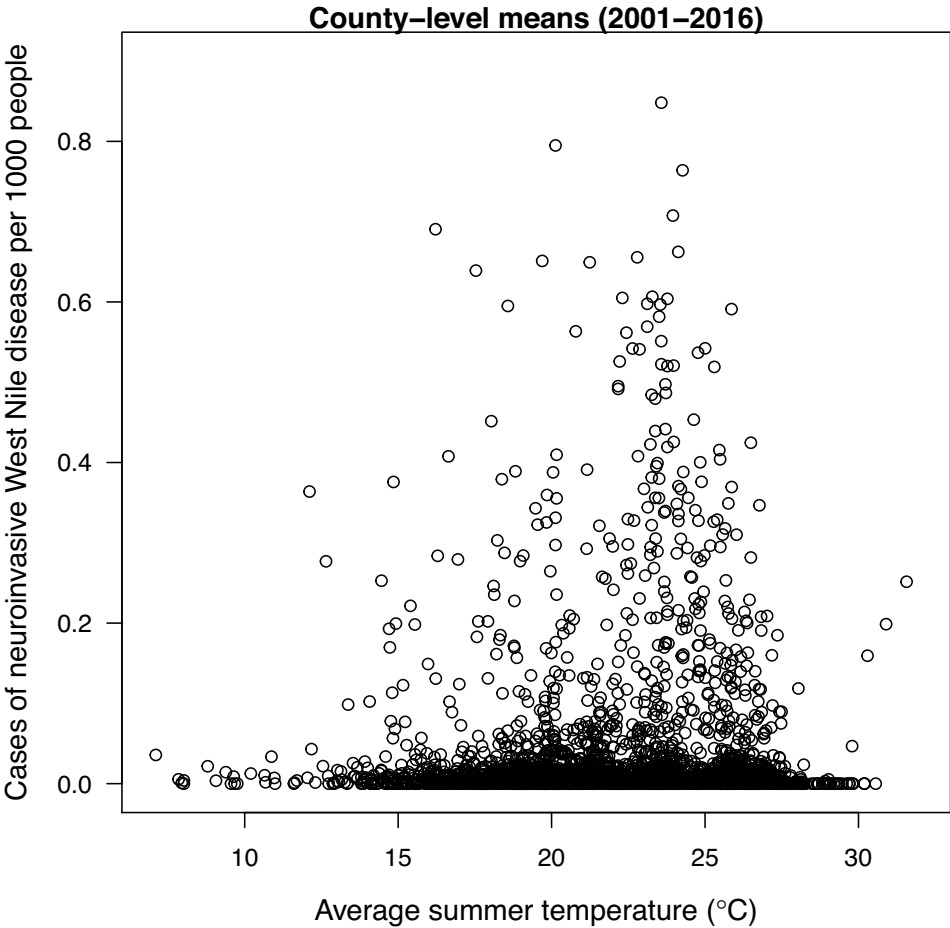

**Appendix 1—figure 26.** Raw county-level data (n = 3109) for mean incidence (per 1000 people) of neuroinvasive West Nile disease (2000–2016) as a function of average summer temperature.

