## [Decision Letter]

**Acceptance summary:**

This synthesis of a wide array of thermal trait data for mosquitoes and mosquito-borne viruses in temperate regions represents a significant amount of work, and allows comparing the temperature ranges over which transmission could potentially occur for various mosquito-virus pairs. The results are valuable given the potential implications for changes in mosquito-borne virus transmission in the face of climate change.

**Decision letter after peer review:**

Thank you for submitting your article "Transmission of West Nile and other temperate mosquito-borne viruses peaks at intermediate environmental temperatures" for consideration by *eLife*. Your article has been reviewed by a Senior Editor, a Reviewing Editor, and three reviewers. The following individual involved in review of your submission has agreed to reveal their identity: Alyssa Gehman (Reviewer #3).

As is customary in *eLife*, the reviewers have discussed their critiques with one another. What follows below is a lightly edited compilation of the essential and ancillary points provided by reviewers in their critiques and in their interaction post-review. Please aim to submit before too long a revised version that addresses these concerns directly. Although we expect that you will address these comments in your response letter we also need to see the corresponding revision in the text of the manuscript. Some of the reviewers' comments may seem to be simple queries or challenges that do not prompt revisions to the text. Please keep in mind, however, that readers may have the same perspective as the reviewers. Therefore, it is essential that you attempt to amend or expand the text to clarify the narrative accordingly.

Our expectation is that the authors will eventually carry out the additional work and report on how they affect the relevant conclusions either in a preprint on bioRxiv or medRxiv, or if appropriate, as a Research Advance in *eLife*, either of which would be linked to the original paper.

Summary:

In this study, Shocket et al., analyze an array of thermal trait data for mosquitos and mosquito-borne viruses in temperate regions to propose a comprehensive for studying variation in transmission across pathogens. This manuscript demonstrates that accounting for the non-linear responses to temperature of both host and parasite is imperative to understanding how temperature and climate change will influence disease distribution and abundance.

Essential revisions:

Overall the reviewers commended the breadth and depth of the data and work, and the insights it holds for the potential implications for transmission in the face of climate change. The main expected revisions are the following:

1) The main concern expressed by all reviewers was the potential overlap between the previous review paper by Mordecai et al., and the current submission. After reviewing both, we find that the submission is of important additional scientific interest given that it further details the methods and results that were summarized in the review. However, the authors must also address the risk of plagiarism and copyright infringement this entails. More specifically:

a) Please cite the Mordecai at al., 2019 review in the Introduction, indicating that the previous review included results based on this study and indicating the added value of this paper.

b) Please ensure that there is no plagiarism or text that is reused between the review and the current submission; we will be running text analyses on the final version to prevent plagiarism.

c) We are concerned about copyright infringement given that some of the figures appear to have been reused. The Mordecai et al., 2019 review has been published under a CC BY license, which allows redistributing and adapting the original material as long as proper attribution is given. This means that it is possible to reuse tables and modified figures as long as appropriate credit is given and any changes are highlighted. Therefore, we would ask the authors to include a citation in table and figure titles indicating where relevant that the table/figure was initially published in Mordecai et al., 2019 and in what ways it has been modified for this article (e.g. addition of prediction intervals, data points, color changes, lines of data removed from table, etc.). See: https://creativecommons.org/faq/#how-do-i-properly-attribute-material-offered-under-a-creative-commons-license. More particularly:

i) Appendix 1—figure 10 is identical to the Figure 2 in the review. Please provide attribution.

ii) Table 2 appears nearly identical to Table 2 in the review. Please provide attribution and detail modifications.

iii) Most of the figures in the main text and supplementary data appear to be modified versions of the figures in the Mordecai review with additional data/separated into different panels. Please provide attribution and indicate any changes where the same data is presented in a modified form.

iv) Please also attend to any other reused figures we may have missed.

2) The variable (r) in the R0 model is defined as the rate at which infected hosts recover and become immune. However, there is no information on how this variable was parameterized in the model. It is unclear whether this refers to recovery in humans or in wild bird and livestock hosts. This could potentially influence the downstream analysis and should be further detailed.

3) The variable (N) in the R0 model is defined as human density. However, humans do not contribute to transmission as they are dead end hosts. Please indicate how N was parameterized in the model.

4) Animal host preference is an important missing component in the model which is critical to determine transmission intensity. Preference to feed on birds may result in high levels in enzootic cycles, but may not necessarily lead to infections in humans. Please clarify if animal host preference and density were incorporated in the model, and if not to evaluate how this may impact results.

5) It is unclear which studies are contributing to each thermal trait in the figures. This could be fixed in the figure legends by either providing a call to a table with the references such as Appendix 1—table 3, or directly putting the references to included studies in the figure legends.

6) The acronym PDR is used inconsistently in the manuscript, sometimes as parasite development rate and sometimes as pathogen development rate. Please use a uniform terminology. The reviewers suggest that "pathogen development rate" or "extrinsic incubation time" may be a more appropriate term for a mosquito host.

7) There are various issues with the validation analyses that should be addressed:

a) Please provide more details on how average summer temperature was calculated as there are many potential ways to average temperature (ex. Min-max averages, average of day and night, day averages only, etc.)

b) Average summer temperature does not reflect the full range of temperature variability. Please consider validating model predictions against additional temperature variables such as the days within the R0 optimum, or ninetieth quantiles of temperature, or lower temperature limits of respective vectors (reviewer suggestions).

c) Please provide some discussion as to why there may not be a 2-month lag at the end of summer. While the reviewers did not provide suggestions, I would suggest that important changes in human behavior during the Fall in the US (back to school and work) might offer a plausible explanation.

d) It is not clear in the methods how a single estimate was obtained for Relative R0 and for the number of cases for each month in Figure 9. For R0, counties were weighted by their population size, but how was temperature averaged for each month? How were cases averaged across counties? Were they weighted by their population size? Or is the number presented the total number of cases nationwide?

8) Please provide some discussion on the generalizability of the model to different contexts, given intricate interactions between mosquito genotypes, virus genotype, and environment. Are the results mostly applicable to the US or can it be applied more broadly? In what regions were the mosquitos collected in studies used to inform thermal performance traits, and is it possible there could be regional variations in host/parasite traits?

9) You substituted a trait thermal response from other vectors when no data were available for a particular virus-vector pair. Please discuss the limitations of this assumption, as even between different populations of a same species there may be large variation in these traits, so there may also be large differences between different virus-vector pairs.

10) Please discuss the limitations of using data collected at constant temperatures to infer transmission in a context of fluctuating temperatures in the field.

11) Figure 1 does not include all mosquito vectors that can potentially transmit these viruses. Please either include all vectors for the listed viruses, or indicate why only these specific vectors were selected.

12) The reviewers question the modeling of adult lifespan as a linear decreasing function, given that there is almost certainly a minimal temperature where lifespan will be zero. They suggest that a modified flipped reverse Briere function (Briere, Gehman, Hall, Byers) based on freeze tolerance of mosquitos might be more realistic than the current function. Another suggestion was to use data on mud crab lifespan over temperature as another source of data given the similarities, as there is some precedence for lifespan optima being lower in marine crabs (Gehman, Hall and Byers, 2018). Please consider either fitting a modified Briere instead, or discuss the limitation of the linear assumption and how this may have affected results.

13) Please provide model code to be assessed by the reviewers, as we cannot publish it without having peer-reviewed it.

14) Table 1 WEEV: is there any evidence of infection in the US? Is the statement that the CDC doesn't report the disease indicating that there are no known cases in the US? Are there known cases elsewhere?

15) Appendix 1—table 1: Please redefine b, c, bc, b*c in the table legend.

16) Because there are many different variables analyzed in the context of this paper for the R0 formula, it would help the reader if variables were always referred to by both their full names and abbreviations every time they are mentioned in the text.

17) Please provide proper X and Y labels for Appendix 1—figure 24

18) Please divide the first sentence of the Introduction into two sentences to improve clarity.

19) The definition of "intermediate environmental temperatures" in the title is unclear. Please rephrase the title with more specific terms.

---

## [Author Response]

Essential revisions:Overall the reviewers commended the breadth and depth of the data and work, and the insights it holds for the potential implications for transmission in the face of climate change. The main expected revisions are the following:1) The main concern expressed by all reviewers was the potential overlap between the previous review paper by Mordecai et al., and the current submission. After reviewing both, we find that the submission is of important additional scientific interest given that it further details the methods and results that were summarized in the review. However, the authors must also address the risk of plagiarism and copyright infringement this entails.

We understand this concern. The two papers were originally submitted at the same time, but due to different trajectories of the review processes, the review/synthesis paper was published sooner. We want to emphasize that this paper is distinct for several reasons. First, as noted by the reviewers, it includes more details regarding the methods and results for the trait-level analyses. In particular, it directly compares the full 95% credible intervals on all trait thermal response curves between different vector and pathogen species, which the Mordecai et al., 2019 does not do (only plotting mean thermal responses). This paper also includes extensive sensitivity and uncertainty analyses for the *R_0_* calculations. Second, this paper describes the ecology of the vectors and vector-pathogen systems in more detail in both the Introduction and Discussion section. Finally, and most importantly, this manuscript also includes original analyses with human case data that are not published elsewhere (Figure 8 and Figure 9; although, a preliminary version of Figure 8 using a LOESS model rather than a GAM was published as Figures S3 of Mordecai et al., 2019). The conclusions of this paper, and the scope of our Discussion section, depend on this combination of both mechanistic models and analyses of human case data.

We thank you for providing the detailed list below for how to address the potential dual-publication issue.

More specifically:a) Please cite the Mordecai at al., 2019 review in the Introduction, indicating that the previous review included results based on this study and indicating the added value of this paper.

Preliminary results of this study—the thermal responses for traits and relative *R_0_* models—were included in a review and synthesis article that was published last year (Mordecai et al., 2019). The present publication presents the complete methods and results, describes the vector and pathogen ecology in more detail, and provides original analyses of human case data.

b) Please ensure that there is no plagiarism or text that is reused between the review and the current submission; we will be running text analyses on the final version to prevent plagiarism.

We appreciate your diligence as a publisher. For this revision we used an online plagiarism tool (www.copyleaks.com) to compare both manuscripts. The only hits aside from affiliations and references were the following four phrases that we did not consider to constitute plagiarism: “to understand the effect of temperature on”, “traits at three or more constant temperatures”, “relative importance of temperature versus other drivers”, and a partial list of pathogens.

c) We are concerned about copyright infringement given that some of the figures appear to have been reused. The Mordecai et al., 2019 review has been published under a CC BY license, which allows redistributing and adapting the original material as long as proper attribution is given. This means that it is possible to reuse tables and modified figures as long as appropriate credit is given and any changes are highlighted. Therefore, we would ask the authors to include a citation in table and figure titles indicating where relevant that the table/figure was initially published in Mordecai et al., 2019 and in what ways it has been modified for this article (e.g. addition of prediction intervals, data points, color changes, lines of data removed from table, etc.). See: https://creativecommons.org/faq/#how-do-i-properly-attribute-material-offered-under-a-creative-commons-license. More particularly:i) Appendix 1—figure 10 is identical to the Figure 2 in the review. Please provide attribution.ii) Table 2 appears nearly identical to Table 2 in the review. Please provide attribution and detail modifications.iii) Most of the figures in the main text and supplementary data appear to be modified versions of the figures in the Mordecai review with additional data/separated into different panels. Please provide attribution and indicate any changes where the same data is presented in a modified form.iv) Please also attend to any other reused figures we may have missed.

We have completed all of the attribution tasks listed above. Below are examples of the text that we used in the table and figure captions.

A version of this table (without thermal breadth, different order of *R_0_* models) was published in Mordecai et al., 2019 (as Table 2 in that paper).

The mean thermal responses for these traits were printed in Mordecai et al. 2019 (as part of Figure 4) without the trait data and 95% CIs, combined onto fewer panels, and along with thermal responses for six other vectors. See Appendix—table 2 and Appendix 1—table 3 for data sources.

2) The variable (r) in the R0 model is defined as the rate at which infected hosts recover and become immune. However, there is no information on how this variable was parameterized in the model. It is unclear whether this refers to recovery in humans or in wild bird and livestock hosts. This could potentially influence the downstream analysis and should be further detailed.3) The variable (N) in the R0 model is defined as human density. However, humans do not contribute to transmission as they are dead end hosts. Please indicate how N was parameterized in the model.

These and the subsequent point raise the important issue of parameters that are not directly temperature-dependent. Since our analyses focus on the effects of temperature on *R_0_*, and the variables *r* and *N* do not depend on temperature, they do not affect the model results. Therefore, they were not included in the relative *R_0_* models parameterized here, although we mention them briefly in the text to be mathematically thorough. We now clarify this point in the main text (subsection “Model overview”). Additionally, we thank you for pointing out that humans are dead-end hosts for these pathogens, so we have redefined *N* and *r* as referring to generic ‘hosts’ in the text.

4) Animal host preference is an important missing component in the model which is critical to determine transmission intensity. Preference to feed on birds may result in high levels in enzootic cycles, but may not necessarily lead to infections in humans. Please clarify if animal host preference and density were incorporated in the model, and if not to evaluate how this may impact results.

We agree that mosquito host preference and host density are important drivers of mosquito-borne disease in general, and West Nile virus transmission dynamics specifically. Our model isolates the direct (physiological) effects of temperature on vectors and viruses alone and does not incorporate these host factors. We have now expanded our discussion of this issue (Discussion section), quoted below:

“Additionally, as wild birds begin to migrate in late summer, both *Cx. pipiens* and *Cx. tarsalis* shift their feeding preferences from birds to humans, which should increase transmission to people later in the year (Kilpatrick et al., 2006). However, we found that cases decreased more quickly in autumn than expected from temperature effects alone. Human behavior may partially compensate for the shift in feeding preference and explain why the decrease of cases in autumn did not show the expected two-month lag from temperature-dependent relative *R_0_*. For instance, if people wear clothing that exposes less skin and spend less time outdoors due to school schedules and changing daylight it may reduce contact with mosquitoes. Drought, precipitation, and reservoir and human immunity also strongly drive transmission of WNV (Ahmadnejad et al., 2016; Marcantonio et al., 2015; Paull et al., 2017; Shand et al., 2016) and may interact with temperature.”

5) It is unclear which studies are contributing to each thermal trait in the figures. This could be fixed in the figure legends by either providing a call to a table with the references such as Appendix 1 —table 3, or directly putting the references to included studies in the figure legends.

We have added references to Appendix 1—table 2, Appendix 1—table 3, Appendix 1—table 4, Appendix 1—table 5, Appendix 1—table 6 (as appropriate) to all of the figure captions for figures with the trait thermal responses (see example in the response to item #1iv above).

6) The acronym PDR is used inconsistently in the manuscript, sometimes as parasite development rate and sometimes as pathogen development rate. Please use a uniform terminology. The reviewers suggest that "pathogen development rate" or "extrinsic incubation time" may be a more appropriate term for a mosquito host.

Thank you for pointing this out. We have changed all instances to “pathogen development rate.”

7) There are various issues with the validation analyses that should be addressed:a) Please provide more details on how average summer temperature was calculated as there are many potential ways to average temperature (ex. Min-max averages, average of day and night, day averages only, etc.)

The gridded, interpolated climate product that we used (from the University of East Anglia’s Climate Research Unit; Harris et al., 2014) contained historic monthly mean temperature data, so we did not calculate the monthly means ourselves. According to the World Meteorological Organization, these standard CLIMAT data are calculated by averaging daily mean temperatures at the station level (based on 4-8 observations per day at regular intervals) and interpolating these over a grid (Handbook on CLIMAT and CLIMAT TEMP Reporting, 2009 edition).

We added this information and the additional citation to the Materials and methods section.

b) Average summer temperature does not reflect the full range of temperature variability. Please consider validating model predictions against additional temperature variables such as the days within the R0 optimum, or ninetieth quantiles of temperature, or lower temperature limits of respective vectors (reviewer suggestions).

We agree that temperature variation is important and expanded our discussion of the effects of varying temperature in the Discussion section (excerpted in response to item #10). We also agree that these are excellent suggestions for building statistical models to answer key questions such as: What temperature metric is the best predictor of WNV transmission? What scales of temperature variation matter most for WNV transmission? How much variation in WNV transmission is explained by temperature? We believe these questions are beyond the scope of this paper, given its already extensive length and focus on building the *R_0_* models. Our goal was to look at broad-scale patterns and perform a validation that closely matched the format of our trait data input and *R_0_* model output (i.e., mean temperature as the independent variable) and could be compared to our *R_0_* model thermal response in terms of shape and key temperature values (optimum and thermal limits).

We are currently working a follow-up manuscript that performs a more in-depth analysis of the WNV case data, including looking at different measures of temperature and additional factors beyond temperature, and we look forward to incorporating these suggestions there.

c) Please provide some discussion as to why there may not be a 2-month lag at the end of summer. While the reviewers did not provide suggestions, I would suggest that important changes in human behavior during the Fall in the US (back to school and work) might offer a plausible explanation.

We appreciate this suggestion and incorporated it into the revised text (see excerpt above in response to item #4 re: feeding preferences).

d) It is not clear in the methods how a single estimate was obtained for Relative R0 and for the number of cases for each month in Figure 9. For R0, counties were weighted by their population size, but how was temperature averaged for each month? How were cases averaged across counties? Were they weighted by their population size? Or is the number presented the total number of cases nationwide?

This analysis uses the same monthly mean temperature data that were provided as a climate product and not calculated by us (see response above to item #7a). Month-of-onset case data are only available aggregated at the national scale (we inquired with the CDC about getting state or county level data and they declined to provide it), which dictated our approach. We acquired state-level data on the proportion of WNV positive mosquitoes for our three North American vector species (*Cx. pipiens*, *Cx. quinquefasciatus*, and *Cx. tarsalis*). We used these proportions to weight the three species-specific relative *R_0_* models to calculate a monthly *R_0_*(*T*) based on the county monthly mean temperature. We then weighted all of those county-level *R_0_*(*T*) values by population size to estimate a national value for *R_0_*(*T*).

We revised the Materials and methods section to make this approach more clear.

8) Please provide some discussion on the generalizability of the model to different contexts, given intricate interactions between mosquito genotypes, virus genotype, and environment. Are the results mostly applicable to the US or can it be applied more broadly? In what regions were the mosquitos collected in studies used to inform thermal performance traits, and is it possible there could be regional variations in host/parasite traits?

We have expanded our discussion of this topic, as follows(Discussion section).

“Our trait-based *R_0_* models effectively isolated the effect of temperature. However, in nature many other environmental and biological factors also impact transmission of mosquito-borne disease. For example, potential factors include rainfall, habitat and land-use, reservoir host community composition, host immunity, viral and mosquito genotypes, mosquito microbiome, vector control efforts, and human behavior (Shocket et al., 2020). Our analyses here suggest that temperature is important for shaping broad-scale spatial and seasonal patterns of disease when cases are averaged over time and space. Other factors may be more important at finer spatial or temporal scales, and may explain additional variation in human cases. For instance, a study of WNV and two other (non-mosquito-borne) pathogens found that biotic factors were significant drivers of disease distributions at local scales, while climate factors were only significant drivers at larger regional scales (Cohen et al., 2016). Given that our *R_0_* models for WNV predicted very similar thermal optima across three distantly-related vector species, it is likely that our results are generalizable to other temperate locations with the same vectors (e.g., *Cx. pipiens* in Europe) at similarly broad spatial and temporal scales, even if the other factors influencing local-scale patterns are quite different than in the US.”

9) You substituted a trait thermal response from other vectors when no data were available for a particular virus-vector pair. Please discuss the limitations of this assumption, as even between different populations of a same species there may be large variation in these traits, so there may also be large differences between different virus-vector pairs.

We added text to the Discussion section paragraph on limitations due to missing and low quality data in order to (1) emphasize this shortcoming in the methods and (2) expand our discussion on variation in thermal performance between different populations of the same species:

New data are particularly important for RVFV: the virus has a primarily tropical distribution in Africa and the Middle East, but the model depends on traits measured in *Cx. pipiens* collected from temperate regions and infection traits measured in *Ae. taeniorhynchus*, a North American species. This substitution of a mosquito species that is not a naturally occurring vector could reduce the relevance and utility of this model. RVFV is transmitted by a diverse community of vectors across the African continent, but experiments should prioritize hypothesized primary vectors (e.g., *Ae. circumluteolus* or *Ae. mcintoshi*) or secondary vectors that already have partial trait data (e.g., *Ae. vexans* or *Cx. theileri*) (Braack et al., 2018; Linthicum et al., 2016). […] More generally, thermal responses may vary across vector populations (Kilpatrick et al., 2010) and/or virus isolates even within the same species. Several studies have found differences in thermal performance across different populations of the same mosquito species (Dodson et al., 2012; Mogi, 1992; Reisen, 1995; Ruybal et al., 2016) or pathogen strains (Kilpatrick et al., 2008), but this variation was not systematically associated with their thermal environments of origin. Accordingly, the potential for thermal adaption in mosquitoes and their pathogens remains an open question. Regardless, more data may improve the accuracy of all of the models, even those without missing data.

10) Please discuss the limitations of using data collected at constant temperatures to infer transmission in a context of fluctuating temperatures in the field.

We expanded our discussion of the effects of varying temperature in the Discussion section:

“Accounting for the effects of temperature variation (Bernhardt et al., 2018; Lambrechts et al., 2011; Paaijmans et al., 2010) is an important next step for using these types of models to accurately predict transmission. In nature, mosquitoes and pathogens experience daily temperature variation that can dramatically alter performance compared to constant temperatures with the same mean temperature (Lambrechts et al., 2011; Paaijmans et al., 2010). Rate summation is the most common method for predicting performance in variable temperatures based on experimental data at constant temperatures (Bernhardt et al., 2018; Lambrechts et al., 2011). This approach is ideal because mean temperature and daily temperature variation vary somewhat independently over space and time, and measuring vector and pathogen performance at sufficient combinations of both is logistically difficult. However, its accuracy for predicting mosquito and pathogen traits or mosquito-borne disease transmission has not been rigorously evaluated.”

11) Figure 1 does not include all mosquito vectors that can potentially transmit these viruses. Please either include all vectors for the listed viruses, or indicate why only these specific vectors were selected.

We believe that providing an exhaustive list of potential vectors for all six viruses is beyond the scope of this paper for the following reasons. First, determining what is a vector is not straightforward. There are three criteria that are typically reported—field isolation, lab infection, and lab transmission—and it is not obvious what criteria or combination of criteria to use. Second, assuming we use the most inclusive criteria, the number of species quickly gets very large for many diseases. For instance, according to Braack et al., 2018, there 48 mosquito species that fit at least one criterion for Rift Valley Fever (including *Ae. aegypti* and *An. gambiae*, which are typically considered vectors of dengue fever and malaria, respectively). Exhaustively reporting the vectors for all six diseases with adequate context could form the bulk of a whole publication by itself (and indeed, often does, e.g., for Braack et al., 2018). Third, research effort and approaches are not uniform across pathogens (e.g., most West Nile virus vector research in North America focuses on quantifying known vectors rather than on identifying new ones), so reporting all suspected vectors will give a biased picture of host range among the different viruses. Fourth, to be an “important vector” there need to be reasonably high mosquito densities overlapping with human populations, and this aspect is rarely reported directly alongside the other three criteria for potential vector status. Given these issues, we relied on other studies (cited in the figure caption) that identified the most important vector species for each disease.

Our goals for Figure 1 were (1) to communicate that viruses are transmitted by multiple vectors and vice versa, (2) highlight the most important vectors for each virus, and (3) represent infection data availability for this subset of vectors. The figure caption now directly states these main points and that our figure is not an exhaustive list of vectors, referring readers to the appropriate sources. Additionally, based on additional reading motivated by this reviewer comment, we revised Figure 1 to add *Cx. modestus*, an important vector of WNV in Europe.

12) The reviewers question the modeling of adult lifespan as a linear decreasing function, given that there is almost certainly a minimal temperature where lifespan will be zero. They suggest that a modified flipped reverse Briere function (Briere, Gehman, Hall, Byers) based on freeze tolerance of mosquitos might be more realistic than the current function. Another suggestion was to use data on mud crab lifespan over temperature as another source of data given the similarities, as there is some precedence for lifespan optima being lower in marine crabs (Gehman, Hall and Byers, 2018). Please consider either fitting a modified Briere instead, or discuss the limitation of the linear assumption and how this may have affected results.

We agree that there is indeed a minimal temperature where lifespan will be zero, probably just below 0ºC, based on observational data that *Cx. pipiens* successfully overwinters at near zero and possibly sub-zero temperatures for up to 4 months (120 days) (Vinogradova, 2000). We considered several options, including a reverse Briere function, in our initial model fitting choices. We opted to be conservative such that lifespan was not a major driver of the temperature-dependence of *R_0_* at temperatures where it was not measured. Using a reverse Briere function with a *T_0_* at 0ºC would have assumed very high lifespan at temperatures just above 0ºC, where lifespan was not actually measured. By contrast, our approach conservatively assumes that lifespan plateaus across a wide range of temperatures ranging from 0ºC to14–16ºC. Because other traits drive relative *R_0_* to 0 well above 0ºC, it is unlikely that this decision affects the accuracy of lower limit of *R_0_*, our main interest here (at least for *Cx. pipiens –* less is known about overwintering for *Cx. tarsalis* and especially for *Cx. quinquefasciatus*). However, it does limit the utility of using these thermal response functions for other applications, e.g., using them to predict actual survival at temperatures below the coldest observation.

For these reasons, we elected to keep the linear fits for this manuscript while clarifying our methods in the model description (subsection “Model overview”) and expanding the Discussion section paragraph about lifespan method.

“Given the lack of rigorous trait data, we cannot be certain of the shape of the thermal response of lifespan below 14ºC, although it is almost certainly unimodal, especially at extreme temperatures expected to be fatal even for diapausing mosquitoes (i.e., below 0ºC). Our decision to assume lifespan (*lf*) plateaued at temperatures below the observed data was based on vector natural history (Vinogradova, 2000) and intended to be conservative. This approach ensured that lifespan was not a major driver of the temperature-dependence of *R_0_* at temperatures where it was not measured and that *R_0_* was instead constrained at reasonable temperatures by other traits. Accordingly, our functions for lifespan (*lf*) do not represent the real quantitative thermal responses below the coldest observations, which limits their utility for other applications, such as predicting survival at cold temperatures and lower thermal limits on survival.”

13) Please provide model code to be assessed by the reviewers, as we cannot publish it without having peer-reviewed it.

The code is now provided for review, available via GitHub: https://github.com/mshocket/Six-Viruses-Temp

14) Table 1 WEEV: is there any evidence of infection in the US? Is the statement that the CDC doesn't report the disease indicating that there are no known cases in the US? Are there known cases elsewhere?

WEEV infections do occur in the US and it has been a National Notifiable disease since at least 2005. We do not know why the CDC does not currently publish WEEV data on their website as they do for EEEV and SLEV. Upon further searching, we found a journal article (Ronca et al., 2016) that cites a CDC website updated in 2010 as a source for 640 reported cases of WEEV in the US from 1964 to 2010. It also notes that cases have decreased in recent years. We now include these cases numbers in Table 1 and the additional citation in the caption.

15) Appendix 1—table 1: Please redefine b, c, bc, b*c in the table legend.16) Because there are many different variables analyzed in the context of this paper for the R0 formula, it would help the reader if variables were always referred to by both their full names and abbreviations every time they are mentioned in the text.17) Please provide proper X and Y labels for Appendix 1—figure 2418) Please divide the first sentence of the Introduction into two sentences to improve clarity.

We thank the reviewers for increasing the readability of our manuscript and have made the above changes.

19) The definition of "intermediate environmental temperatures" in the title is unclear. Please rephrase the title with more specific terms.

We revised the title: “Transmission of West Nile and five other temperate mosquito-borne viruses peaks at temperatures from 23–26ºC.”